# POSITIONAL ATTENTION: OUT-OF-DISTRIBUTION GENERALIZATION AND EXPRESSIVITY FOR NEURAL ALGORITHMIC REASONING

## ABSTRACT

There has been a growing interest in the ability of neural networks to solve algorithmic tasks, such as arithmetic, summary statistics, and sorting. While state-of-the-art models like Transformers have demonstrated good generalization performance on in-distribution tasks, their out-of-distribution (OOD) performance is poor when trained end-to-end. In this paper, we focus on value generalization, a common instance of OOD generalization where the test distribution has the same input sequence length as the training distribution, but the value ranges in the training and test distributions do not necessarily overlap. We propose that using fixed positional encodings to determine attention weights – referred to as positional attention – enhances empirical OOD performance while maintaining expressivity. We support our claim about expressivity by proving that Transformers with positional attention can simulate parallel algorithms.

## 1 INTRODUCTION

Transformers (Vaswani et al., 2017) are versatile models used in various applications, including vision (Yuan et al., 2021; Khan et al., 2022; Dehghani et al., 2023) and natural language processing (Wei et al., 2022b; Touvron et al., 2023). Their effectiveness in complex tasks is particularly notable in Large Language Models (LLMs) (Wang et al., 2018; Hendrycks et al., 2021), where they excel at generating coherent text and understanding context. This strong performance has led to an increased interest in understanding the Transformer architecture as a computational model capable of executing instructions and solving algorithmic reasoning problems.

In this context, Pérez et al. (2021); Wei et al. (2022a) show that Transformers are Turing Complete, and Giannou et al. (2023); Back De Luca & Fountoulakis (2024); Yang et al. (2024) demonstrate that Transformers can effectively encode instructions to solve linear algebra and graphs problems. Additionally, it has been shown that Transformers can perform reasoning tasks using far fewer layers than the number of reasoning steps (Liu et al., 2023), indicating a connection between Transformers and parallel algorithms. To this end, Sanford et al. (2024) further demonstrates that Transformers can simulate the Massively Parallel Computation (MPC) model (Andoni et al., 2018), which is based on the MapReduce framework for large-scale data processing (Dean & Ghemawat, 2008).

Complementing this theoretical framework, empirical studies have demonstrated the capabilities of Transformers, among other models, in executing algorithms (Veličković & Blundell, 2021). Notable applications include basic arithmetic (Lee et al., 2024), sorting (Tay et al., 2020; Yan et al., 2020), dynamic programming (Dudzik & Veličković, 2022; Ibarz et al., 2022b), and graph problems (Veličković et al., 2022; Cappart et al., 2023).

Despite promising empirical results, these approaches rely on additional supervision, such as intermediate labels of existing algorithms (Veličković et al., 2022) or self-supervised learning techniques to incorporate algorithmic information (Rodionov & Prokhorenkova, 2023). While extra supervision helps guide the model toward solutions with some degree of OOD generalization, it is not only more expensive to train but also requires knowledge of the underlying algorithmic solution to the problem. Furthermore, this additional supervision can limit the model's ability to derive alternative solutions independently, reducing their functionality to simply simulating a predetermined algorithm. On the other hand, training models like Transformers end-to-end, without additional supervision, on algo-

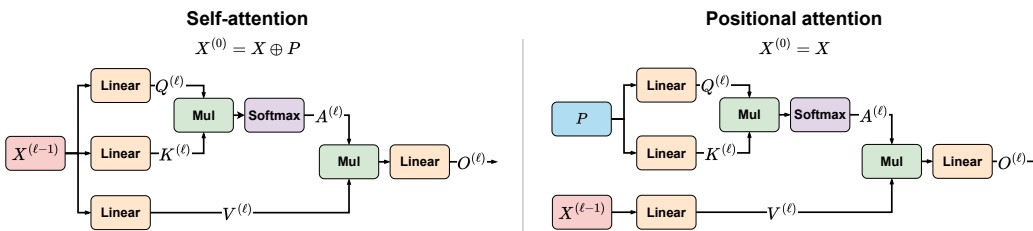

Figure 1: Diagram comparing the operations of self-attention in Transformers with positional attention. The figure illustrates a single attention head, but in multi-head attention, multiple sets of queries, keys, and values are processed in parallel and then combined. In Transformers, the model's input, $X^{(0)}$, is a combination of input values $X$ and positional encodings $P$. In positional attention, however, these components are processed separately. At layer $\ell$, the query ($Q^{(\ell)}$) and key ($K^{(\ell)}$) are derived solely from the positional encodings $P$, where $P$ remains fixed across layers. These are multiplied (denoted by Mul) and passed through a softmax function to produce the attention matrix $A^{(\ell)}$. As in self-attention, the value $V^{(\ell)}$ in positional attention is computed from the previous layer's input, $X^{(\ell-1)}$. The attention matrix $A^{(\ell)}$ and the value $V^{(\ell)}$ are then multiplied to form the weighted representation, which is linearly transformed into the output $O^{(\ell)}$. This output is passed to a Multilayer Perceptron (MLP) for further processing, as detailed in Section 4.

rithmic tasks often leads to overfitting. In such cases, models tend to rely on properties of the training distribution, such as the range of values or input length, resulting in poor OOD performance.

It has been shown both theoretically and empirically that aligning the model's architecture with the algorithm improves generalization (Xu et al., 2020). While it is well-known that Transformers can align with parallel computational models (Sanford et al., 2024), we take this connection a step further by showing that this alignment can be further refined through the use of positional attention. Positional attention differs from standard self-attention in that the attention weights are computed solely using fixed positional encodings that remain the same across all layers.

Regarding OOD generalization, we focus on settings where the input lengths are the same, but the values in the test set have a different or larger magnitude than those in the training set. We call this type of OOD generalization *value generalization* and provide a formal definition in Section 3. Value generalization is particularly important since, when learning to solve an algorithmic task, the model is expected to be able to perform such a task on a range of numbers that it might not have seen during training. This should also serve as an indication that the model learns to solve the problem.

**Our contributions:** We examine Transformers with positional attention (*positional Transformers*) from the perspective of expressivity and OOD value generalization in algorithmic reasoning tasks.

1. (OOD generalization) We empirically demonstrate an average $1000\times$ improvement (ranging from $400\times$ to $3000\times$) in OOD value generalization for positional Transformers compared to traditional Transformers during end-to-end training on various algorithmic tasks (Section 6).

2. (Expressivity) We prove that positional Transformers can simulate any algorithm defined in a parallel computation model, defined in Section 5, which we call Parallel Computation with Oracle Communication (PCOC).

## 2 RELATED WORK

**Empirical:** Several studies focus on the empirical aspects of training neural networks to execute algorithms, achieving promising results in OOD generalization. Notable examples include Yan et al. (2020); Ibarz et al. (2022a); Diao & Loynd (2023); Bevilacqua et al. (2023); Engelmayer et al. (2023); Rodionov & Prokhorenkova (2023). However, all these works rely on some form of additional supervision during training, with some differing in the type of supervision employed. Specifically, Rodionov & Prokhorenkova (2023) leverages problem-specific information within a self-supervised training framework, whereas the other studies utilize intermediate labels to guide

the learning process. For instance, Engelmayer et al. (2023) demonstrates that using intermediate labels derived from parallel algorithms leads to improved performance for parallelizable tasks.

From the perspective of OOD generalization, most works focus on length generalization (Veličković et al., 2022; Minder et al., 2023), i.e., testing on longer inputs, with a few exceptions addressing graphs with different connectivity distributions (Georgiev et al., 2023) as well as graphs with varying sizes, edge weights, and connectivity patterns (Tang et al., 2020). In the context of value generalization, (Klindt, 2023) shows how simple neural network models fail to learn to sum two numbers in a way that generalizes to larger values. In length generalization, some studies employ digitization schemes, such as binary numbers or tokenization (Kaiser & Sutskever, 2016; Lee et al., 2024; Shen et al., 2023; Ruoss et al., 2023), which can also imply value generalization. However, these operations are typically restricted to processing only two numbers, and the values involved are all integers. In contrast, our work operates on real numbers and supports processing multiple elements simultaneously instead of just two.

While some research, such as Kazemnejad et al. (2023), investigates the role of different positional encodings in length generalization, to the best of our knowledge, no existing work examines the use of positional attention within the context of neural algorithmic reasoning. The closest formulation to the positional Transformer architecture presented in Section 4 is the position-based attention proposed by Schmidt & Di Gangi (2023), but it is used in the context of neural machine translation.

**Theoretical:** From a theoretical perspective, the most closely related work to ours is Sanford et al. (2024), which presents simulation results for Transformers within the Massively Parallel Computation (MPC) model. This approach uses local computations over the input data to determine the destinations for communication between machines. In contrast, our method encodes this communication information directly within the network parameters, eliminating the need for destinations to depend on the input. Consequently, we adopt a different parallel computational model.

Other relevant studies demonstrate the expressive power of neural networks through simulation results. For instance, Siegelmann & Sontag (1995) establishes the Turing completeness of recurrent neural networks (RNNs), while Hertrich & Skutella (2023) presents specific RNN constructions that solve the shortest paths problem and provide approximate solutions to the Knapsack problem. Additionally, other simulation results focused on Transformers have shown their Turing completeness (Pérez et al., 2021; Wei et al., 2022a) as well as demonstrated constructive solutions to linear algebra and graph-related problems (Giannou et al., 2023; Back De Luca & Fountoulakis, 2024; Yang et al., 2024). In our work, we are also motivated by the concept of algorithmic alignment (Xu et al., 2020), demonstrating that further aligning the Transformer architecture (Vaswani et al., 2017) with parallel algorithms can lead to better empirical performance.

## 3 PRELIMINARIES AND NOTATION

We denote by $\mathbb{N} = \{1, 2, 3, \dots\}$ the set of natural numbers. We use $[n]$ to refer to the set $\{1, 2, \dots, n\}$ for $n \in \mathbb{N}$. For a set $S$ we denote by $\mathcal{P}(S)$ its power set (i.e. the set containing all subsets of $S$).

**Out-of-distribution generalization.** Out-of-distribution (OOD) generalization broadly refers to the ability of a supervised learning model to "perform well" when evaluated on data that are drawn from a distribution that is different from the one used to generate the training data. Two quantitative measures of OOD generalization are given below.

**Definition 1** (OOD risk). Let $\mathcal{X}$ be the feature space, $\mathcal{Y}$ be the set of labels, and let $h : \mathcal{X} \to \mathcal{Y}$ be the hypothesis returned by a supervised learning algorithm where the training data are sampled from a distribution $\mathcal{D}_{\text{train}}$ on $\mathcal{X} \times \mathcal{Y}$. Let $\mathcal{D}_{\text{test}}$ be a different distribution on $\mathcal{X} \times \mathcal{Y}$. The *out-of-distribution risk* of $h$ with respect to $\mathcal{D}_{\text{test}}$ is defined as $R_{\mathcal{D}_{\text{test}}}(h) = \mathbb{E}_{(x,y) \sim \mathcal{D}_{\text{test}}}[\ell(h(x), y)]$.

**Definition 2** (Empirical OOD risk). Using the same setting as Definition 1 we define the *Empirical OOD risk* as $R_S(h) = \frac{1}{n} \sum_{i=1}^{n} \ell(h(x_i), y_i)$, where $S = ((x_1, y_1), \dots, (x_n, y_n))$ are drawn i.i.d. from the distribution $\mathcal{D}_{\text{test}}$.

Models that achieve low (empirical) OOD risk are said to *OOD-generalize*. We now define *value generalization*, which is the type of OOD generalization that we are concerned with in this work.

**Definition 3** (Value generalization). We use the term *value generalization* to refer to the following particular case of OOD generalization. The feature space is Euclidean, i.e., $\mathcal{X} \subseteq \mathbb{R}^k$ for some $k \in \mathbb{N}$. There exists a ground-truth labeling function $\hat{h}$ which maps every $x \in \mathcal{X}$ to its true label $y = \hat{h}(x) \in \mathcal{Y}$, i.e., $\mathcal{D}_{\text{train}}$ and $\mathcal{D}_{\text{test}}$ are completely characterized by their marginalization onto $\mathcal{X}$, denoted by $\mathcal{D}_{\text{train}}(\mathcal{X})$ and $\mathcal{D}_{\text{test}}(\mathcal{X})$, respectively. We say that a model *value-generalizes* from $\mathcal{D}_{\text{train}}$ to $\mathcal{D}_{\text{test}}$ if it achieves low OOD risk and $\text{supp}(\mathcal{D}_{\text{test}}(\mathcal{X})) \backslash \text{supp}(\mathcal{D}_{\text{train}}(\mathcal{X})) \neq \emptyset$.[1]

Note that the quantities used to measure OOD generalization (and value generalization in particular) do not assume anything regarding the overlap between the training and test distributions. In the context of learning, this can lead to artificially low OOD risk when there is significant overlap between $\mathcal{D}_{\text{train}}$ and $\mathcal{D}_{\text{test}}$. Therefore, the interesting cases are those where samples from $\mathcal{D}_{\text{test}}$ are unlikely to have been sampled from $\mathcal{D}_{\text{train}}$. In the context of value generalization, this translates to $\mathbb{P}_{x \sim \mathcal{D}_{\text{test}}(\mathcal{X})}(x \in \text{supp}(\mathcal{D}_{\text{train}}(\mathcal{X})))$ being low. That is, the probability that a test sample lies in the domain of the training distribution should be small. In our experiments, this probability is sufficiently small (see Appendix D for details), so a low test error indicates "true" value generalization.

In the context of neural algorithmic reasoning, good value generalization (with minimal overlap between test and training distributions) provides a strong indication that a model has learned to execute an algorithm. This is explained by the fact that an algorithm consists of a fixed sequence of instructions that does not change when the input values change.

## 4   THE POSITIONAL TRANSFORMER ARCHITECTURE

We now define the *positional Transformer*, as an adaptation of the Transformer model (Vaswani et al., 2017). The motivation for positional Transformers stems from the observation that in many real-world parallel algorithms, communication between machines is independent of the specific data being processed. Analogously, since the mixture of data is handled by attention, we hypothesize that two factors contribute to poor scale generalization in Transformers: (i) the inclusion of input values in the computation of the attention weights and (ii) the use of positional encodings in the input matrix. To address this, we decouple input values from attention weight computation and remove positional encodings from the input matrix, leading to positional attention. In practice, we validate our empirically validate our hypothesis, demonstrating that positional Transformers achieve better scale generalization within specific algorithmic reasoning tasks.

For an input $\mathbf{X} \in \mathbb{R}^{n \times d_X}$, we define the $\ell^{\text{th}}$ layer of our architecture as follows:

$$\mathbf{F}^{(\ell)}(\mathbf{X}) = \Phi^{(\ell)}\left(\left(\bigoplus_{h=1}^{H} A^{(\ell,h)}\mathbf{X}W_V^{(\ell,h)}\right)W_O^{(\ell)} \oplus X\right). \tag{1}$$

The input is processed by $H$ attention heads, each associated with an attention weight matrix $A^{(\ell,h)} \in (0,1)^{n \times n}$ and a value matrix $W_V^{(\ell,h)} \in \mathbb{R}^{d_X \times d_V}$. Here, $\ell$ denotes the layer index and $h$ the head index, allowing a specific attention head within a layer to be identified as $(\ell, h)$. The outputs of these attention heads are concatenated and then transformed by an output matrix $W_O^{\ell} \in \mathbb{R}^{H \cdot d_V \times d_O}$. This result is concatenated with a residual connection of the input $X$ and then passed through a multilayer perceptron (MLP), represented as $\Phi^{(\ell)} : \mathbb{R}^{d_O + d_X} \to \mathbb{R}^{d_{\text{out}}}$.

Unlike traditional approaches, we utilize *positional attention*, where attention weights are learned solely using positional encodings $P$, which are constant across all layers. This distinction is also illustrated in Figure 1.

$$A^{(\ell,h)} = \text{softmax}\left(\left(PW_Q^{(\ell,h)}\right) \cdot \left(PW_K^{(\ell,h)}\right)^{\top}\right). \tag{2}$$

We utilize node positional encodings defined by a matrix $P \in \mathbb{R}^{n \times d_P}$ and whose attention weights are computed similarly to traditional attention, using query and key matrices $W_Q^{(\ell,h)}, W_K^{(\ell,h)} \in$

---

[1]The support of a distribution $\mathcal{D}$, denoted by $\text{supp}(\mathcal{D})$, is the set of all points whose every open neighborhood $\mathcal{N}$ has the property that $\mathbb{P}_{x \sim \mathcal{D}}(x \in \mathcal{N}) > 0$. Informally, this is the set of all points over which the probability density (under some regularity conditions) or probability mass is nonzero. In this paper, we sometimes abuse the terminology and refer to $\text{supp}(\mathcal{D})$ as the domain of $\mathcal{D}$.

$\mathbb{R}^{d_P \times d_m}$, where $d_m$ is the embedding dimension. The encodings are fixed across layers, as indicated by the absence of an $\ell$ index for $P$.

Theoretically, we evaluate whether these changes reduce positional Transformers' expressive power. We demonstrate that positional Transformers can simulate parallel algorithms under the Parallel Computation with Oracle Communication (PCOC) model, defined in the next section.

## 5 Expressivity of Positional Transformers

We prove that our architecture can simulate algorithms within a parallel computational model, which we refer to as Parallel Computation with Oracle Communication (PCOC). We first describe the main features of PCOC, followed by its definition, and then discuss its limitations. It is important to note that the simulation result is theoretical. In practice, the model could converge to a parameter setting that does not correspond to an interpretable algorithm. However, the simulation result is significant as it demonstrates the minimal capabilities of our architecture in theory. Such theoretical approaches have been employed in previous works. For example, see Sanford et al. (2024); Loukas (2020).

### 5.1 Parallel Computation with Oracle Communication (PCOC)

The PCOC model consists of two steps at each round. The first step is communication, where machines send and receive data from other machines. The communication pattern can change at every round. The second step is computation, where all machines perform some local computation on data stored in their local memory.

**Oracle communication.** For each length $n$ and input data, we assume the existence of an oracle that provides the destination for each machine and message at each round. The oracle executes a communication pattern agnostic to the data being processed. This contrasts with other parallel models, such as the Massively Parallel Computation (MPC) model (Andoni et al., 2018), where it is assumed that the processed data can influence communication patterns. At first glance, introducing such an oracle might not seem particularly useful, especially because it is fixed for each input data, where the data can be real-valued, which implies potentially unaccountably many communication oracles for a particular task. However, its importance lies in the fact that for a variety of parallel algorithms, the communication pattern at each round depends only on the length of the input, and it is independent of other input data. For example, in algorithms that compute basic statistics such as the sum or minimum, or tasks like sorting a list of numbers, the communication between machines is determined not by their values but by their positions. This means that if the values at each machine change, the communication pattern established between machines for a given task and input length remains the same. In Appendix A, we illustrate communication patterns for some of these tasks, which are also addressed in the experiments in Section 6.

The observation that, for many algorithms, communication is agnostic to the data also informs the design of our architecture. On the other hand, since the proposed positional attention does not rely on input data, simulating a model that determines destinations based on input data might be inefficient. Such cases reflect algorithms where the communication pattern varies as a function of the input values. In these instances, PCOC may not be the most appropriate model to consider, a point we further explore in the limitations section below.

**Definition 4** (PCOC model). The PCOC model is described by a set of $n$ machines labeled from 1 to $n$, the number of rounds $R$, an integer $s$ and an oracle $\mathsf{RCV} : [R] \times [n] \to [n] \times (\mathcal{P}([s]) \setminus \{\emptyset\})$ (which is fixed for a given $n$ and input data) satisfying the following:

1. Each machine $i$ has a local memory $\mathtt{MEM}_i \in \mathbb{T}^s$ of size $s$, where $\mathbb{T}$ is some abstract data-type. The contents of the memory are indexed from 1 to $s$ and we use the notation $\mathtt{MEM}_i[j]$ to refer to the element at position $j$ on the memory of the $i$-th machine.

2. Each machine performs some local computation on the data it receives and overrides the results to its memory. A single machine can perform different local computations on different rounds and different machines (generally) perform different local computations.

3. When $(r, i)$, where $r \in [R]$ and $i \in [n]$, is passed to the oracle it returns a subset $M$ of the set $[n] \times (\mathcal{P}([s]) \setminus \{\emptyset\})$. The oracle essentially returns a (possibly empty) set of machines

that machine $i$ has to receive some data from in round $r$ along with the exact positions on the memories of those machines to retrieve.

4. The total size of data sent and received by each machine is at most $s$. Size here is measured in terms of the number of "variables" of data-type $\mathbb{T}$.

The protocol is executed in $R$ rounds. At the start, the input is distributed across the $n$ machines. At the beginning of round $r$, each machine $i$ simultaneously queries the oracle with input $(r, i)$ and receives data from the machines it returns. The machines then simultaneously perform their local computations on the received data.

---

**Input:** $\mathtt{Data} = (\mathtt{Data}_1, \ldots, \mathtt{Data}_n)$ distributed across the memories of $n$ machines, labeled in $[n]$. An oracle $\mathsf{RCV}_{n,\mathtt{Data}} : [R] \times [n] \to [n] \times (\mathcal{P}([s]) \setminus \{\emptyset\})$.

1 **For** each round $r = 1, \ldots, R$ **then**
2      Each machine $i$ simultaneously queries $\mathsf{RCV}_{n,\mathtt{Data}}$ with $(r, i)$ as input and receives

      data from the machines and memory positions returned by the oracle.
3      The machines simultaneously perform local computations on the received data

      and write the results in their local memories.

---

### 5.2 Limitations of PCOC

While PCOC offers great flexibility in executing parallel tasks, it should be noted that the oracle communication scheme can be limiting. The computational model in PCOC is subjected to a given oracle. In contrast, in other models, like MPC, the same computational model can execute different communication patterns based on the input data. Although it is possible to simulate this property within PCOC, it requires more rounds, machines, or memory than with a model such as MPC. We further discuss the differences between the two models in Appendix A.

### 5.3 Positional Transformers can simulate PCOC

Having established the PCOC model, we now show that the positional Transformer model in Equation (1) can simulate it. More specifically, our results show that a $R$-round PCOC protocol can be simulated by a positional Transformer with $R$ layers. We first present the corresponding theorem, followed by a proof overview. The details of the proof are presented in Appendix B

**Theorem 1.** *Consider a PCOC instance* P *with $R$ rounds, $N$ machines with local memory $s$, and data type $\mathbb{T} = \mathbb{R}$. Let $\mathcal{M}$ be a model following the architecture in equation 1 with $n = N + 1$ nodes, $R$ layers and $s$ attention heads. Then, for any instance* P *with Borel measurable local functions, there exists a configuration of $\mathcal{M}$ that approximates* P *to any desired degree of accuracy.*

*Proof overview:* The proof starts by demonstrating that a single layer of the positional Transformer can simulate each individual round of PCOC. The constructive proof can be further divided into two main components: communication and computation.

**Communication:** The communication stage leverages the oracle $\mathsf{RCV}_{n,\mathtt{Data}}$ to specify the subsets of machines and local memory positions from which each machine receives information. These subsets can be transformed into binary encodings, which are represented by distinct attention heads, one for each local memory position, for a total of $s$ positional attention heads. This part of the proof relies on the capability of attention to represent binary patterns, which is shown in Appendix B.1. It is important to note that the number of nodes exceeds the number of machines by one, as an additional node is necessary to represent unsent messages. In the computation of attention, no attention matrix can contain a row of all zeros, implying that a machine is not receiving any information. Due to such cases, we introduce an additional sink node to account for information not directed to any machine. Therefore, in this framework, we can show that any communication pattern defined by the oracle can be effectively represented by $s$ attention heads across $N + 1$ nodes.

**Computation:** This stage accounts for the local computations executed by each machine. To this end, we invoke the universal approximation results of multilayer perceptrons (MLPs) (Cybenko, 1989; Hornik et al., 1989) to establish that, in each round, the local computations initiated by each machine can be approximated by $\Phi^\ell$ as detailed in Equation (1). One important consideration is the inclusion of unique node identifiers in the input to ensure the injectivity of the MLP approximation. Even if two input rows may have the same input values, the unique identifiers guarantee that each row corresponds to a distinct local function. Furthermore, the node identifiers must be preserved at every layer to maintain this injectivity. This is achieved by the MLP when processing both the output of the attention heads and the residual input $X$, ensuring that identifiers are consistently retained.

Therefore, by demonstrating that our architecture can approximate any oracle and local functions, we show its ability to simulate any algorithm in PCOC. In practice, finding an oracle and local functions for a specific task can be posed as a learning problem. Our proposed architecture adopts this approach and can learn to execute parallel algorithms using fixed positional encodings in the attention mechanism. As our experiments illustrate, this approach helps mitigate OOD generalization issues. In our experiments we do not assume access to explicit supervision for communication or computation, both are learned indirectly through ground truth, as all models are trained end-to-end.

## 6 EXPERIMENTS

In this section, we evaluate the performance of positional Transformers across various tasks. Specifically, we compare the effectiveness of positional attention against standard self-attention (Vaswani et al., 2017). Both models utilize the architecture defined in Equation (1), with the distinction that in self-attention, the attention weights are computed based on the input $X$:

$$A^{\ell,h}(X) = \text{softmax}\left(\left(XW_Q^{(\ell,h)}\right) \cdot \left(XW_K^{(\ell,h)}\right)^\top\right). \tag{3}$$

In this setting, standard Transformers also incorporate positional encodings concatenated with the input values. In Appendix C, we examine other configurations for standard Transformers (including one using Rotary Positional Embedding (RoPE)) and find no major differences in performance.

Next, we outline the tasks used in this work, followed by a detailed description of the experimental setup. Finally, we present and discuss the results.

**Tasks:** To analyze the performance of positional attention in contrast to self-attention, we train the models on the following tasks:

1. *Cumulative sum*: Given $x \in \mathbb{R}^n$, output $y \in \mathbb{R}^n$ where each element $y_i$ is the sum of the first $i$ elements of $x$, i.e. $y_i = \sum_{j=1}^{i} x_j$.

2. *Cumulative min*: Given $x \in \mathbb{R}^n$, output $y \in \mathbb{R}^n$ where each element $y_i$ is the minimum value among the first $i$ elements of $x$, i.e. $y_i = \min\{x_j \mid 1 \le j \le i\}$.

3. *Cumulative median*: Given $x \in \mathbb{R}^n$, output $y \in \mathbb{R}^n$ where each element $y_i$ is the median of the first $i$ elements of $x$, i.e. $y_i = \text{median}\{x_j \mid 1 \le j \le i\}$.

4. *Sorting*: Given $x \in \mathbb{R}^n$, output $\text{sort}(x)$, a vector containing the entries of $x$ sorted in ascending order.

5. *Cumulative maximum sum subarray*: given $x \in \mathbb{R}^n$, output $y \in \mathbb{R}^n$ where each element $y_i$ is the sum of elements of a maximum sum subarray of the first $i$ elements of $x$, i.e. $y_i = \max_{1 \le j \le k \le i}\left(\sum_{l=j}^{k} x_l\right)$.

The tasks selected were chosen to ensure a balanced representation of varying levels of complexity. Furthermore, we adopt cumulative versions of algorithms when feasible for several reasons: they naturally provide $n$ to $n$ training settings, they are more challenging than non-cumulative versions, and the non-cumulative versions for tasks such as summing and taking the minimum have trivial one-layer constructions for a fixed input size $n$.

**Experimental setting:** All tasks employ the same model configuration. The model uses the structure of Equation (1), augmented with encoding and decoding layers, which are linear operators.

We compare the standard Transformer, which utilizes the attention mechanism in Equation (3), and the positional Transformer, which employs the attention defined in Equation (2). Both variants share the same number of layers and dimensional configurations, with any specific differences explicitly noted. In all configurations, the total number of layers is set to $\lceil \log_2 n \rceil + 1$, where $n$ denotes the maximum input length, and each layer uses 2 attention heads. Along with each input sequence, we also append an empty scratchpad entry. This extra entry does not count toward the total number of layers and is not used to compute the loss. It is included solely to aid in the computation of the tasks. For the function $\Phi^{(\ell)}$, we employ a 2-layer MLP with ReLU activation functions. The embedding dimension of the encoder and the hidden dimensions of the MLP are both set to $64$.

We use one-hot encoded vectors of dimension $n$ for positional encodings, where the non-zero entry corresponds to the node position. Consequently, the embedding dimensions of $W_Q$ and $W_K$ are set to $n$. A key difference between the models is that standard Transformers concatenate positional encodings to the input, whereas positional Transformers supply positional information exclusively through the matrix $P$. Therefore, in positional Transformers, input values are solely encoded in the input, and positional information is exclusively encoded in the positional encoding matrix.

Both models are trained end-to-end using the squared loss between the predicted and target vectors of size $n$, with no intermediate supervision. We train models with Adam, starting with a learning rate of $5 \cdot 10^{-4}$ and a learning rate scheduler for a total of 2000 epochs.

Our training data consists of samples from the range $[-2, 2]$. To ensure diversity in the data, for each input sample, we first select lower and upper bounds $\gamma_l$ and $\gamma_u$ uniformly in $[-2, 2]$, and then for each of the $n$ elements of the input sample, we select its value uniformly from the interval $[\gamma_l, \gamma_u]$. We employ a similar sampling strategy for testing but extend the value range to $[-2c, 2c]$, where $c > 1$ is the OOD scale factor. Additionally, during the test sampling process, we apply a rejection step to ensure that either $\gamma_l < -2$ or $\gamma_u > 2$, while maintaining $-2c \leq \gamma_l \leq \gamma_u \leq 2c$. This ensures that, with high probability, a test sample does not lie in the domain of the training data. Our sampling strategy for the test data does not guarantee that every test sample will be outside the domain of the training distribution. However, as we show in Appendix D, the probability of generating a test sample that lies inside the domain of the training distribution is at most $O(1/nc^2)$. In fact, it turns out that the vast majority of the test instances in our test data do not lie in the domain of the training distribution.[2] This implies that our test results reflect the "true" value generalization performance of both architectures.

We evaluate our architecture under two different regimes: fixed input length $n$ and variable input lengths ranging from 1 to $n$. We present our results in two subsections corresponding to each regime. We first present the results for variable input lengths, as they represent our general goal. Due to the high resource demands of running variable-length experiments, we resort to a fixed input length setting to provide a more detailed analysis of value generalization as a function of other factors, such as the number of samples and sequence length. The plots presented show the median over ten runs, with shaded areas representing the 10th and 90th percentile. For more detailed analyses, including experiments on a relational task with mixed-type inputs, we refer the reader to Appendix C.

## 6.1 VARIABLE LENGTH INPUTS

In this section, we present value generalization results for models operating on variable-length inputs. This setting aims to verify the models' ability to generalize across different scales while maintaining the flexibility to handle inputs of varying lengths.

**Value Generalization:** In this experiment, we evaluate the models' ability to process sequences of varying lengths up to a maximum size of $n = 8$. Specifically, the model is required to perform tasks on input sequences with lengths ranging from 1 to 8. We train models with 500,000 samples and ensure that all input lengths are equally represented. We then evaluate the OOD loss across different scale factors $c \in \{1, 2, \ldots, 10\}$. Note that when $c = 1$, the setting actually corresponds to in-distribution generalization. The losses reported are calculated using 3,000 samples for each scale. As shown in Figure 5, positional Transformers consistently outperform standard Transformers

---

[2]For example, when $n = 8$ and $c = 3$, which we use in several experiments, the probability of generating a test sample that lies inside the domain of the training distribution is less than 0.05. For details, we refer the reader to Appendix D and Figure 29.

across all scales and tasks. Additionally, our architecture maintains robust OOD performance even in tasks where the output can exceed the input magnitude (e.g., sum and maximum sum subarray).

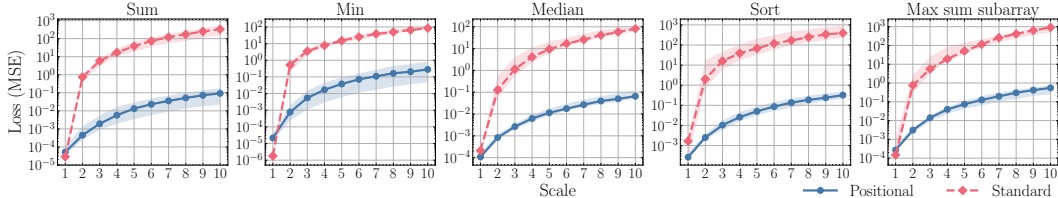

Figure 2: OOD loss (measured as mean squared error, MSE) for standard Transformers (red) and positional Transformers (blue) across all five tasks for *variable* lengths (up to $n = 8$). The x-axis represents the OOD scale factor. The solid line and shaded area denote the median and the region between the 10[th] and 90[th] percentiles over ten trials, respectively.

## 6.2 Fixed length inputs

In this section, we present a more in-depth analysis of value generalization as a function of additional factors such as sample size and input length. Due to the resource demands of variable-length experiments, we present results obtained by training with a single fixed input length.

**Sample Size vs. Value Generalization:** In this setting, we fix the input length $n = 8$ and examine value generalization for $c = 3$, which is three times the training range, i.e., $[-6, 6]$. We then analyze OOD loss as a function of the number of training samples, ranging from $5,000$ to $50,000$. Figure 3 shows that for all tasks, the OOD loss of positional Transformers steadily decreases with an increasing number of samples, whereas the performance of standard Transformers remains roughly constant. Additionally, Appendix C provides training and validation error results, demonstrating that standard Transformers not only converge but also generalize well in-distribution. To rule out potential overfitting due to model complexity, Appendix C includes further analyses showing that standard Transformers with reduced depth also fail to value-generalize.

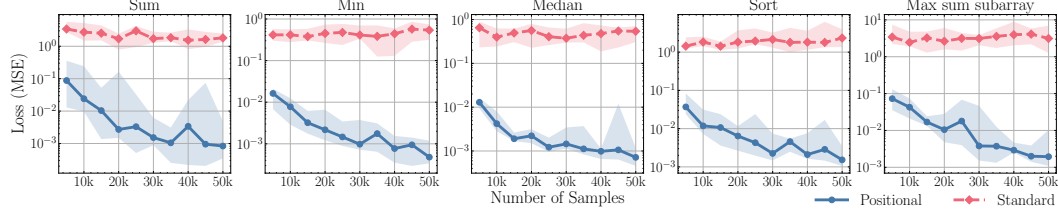

Figure 3: OOD loss across all five tasks for standard Transformers (red) and positional Transformers (blue) as a function of the number of training samples (indicated on the x-axis). Models are trained on the range $[-2, 2]$ with varying training set sizes and tested on $[-6, 6]$ with 1,000 samples.

**Length vs. Value generalization:** This experiment validates that our results hold for multiple values of $n$. We train models for each fixed length $n \in \{2, 4, 8, 16, 32\}$ using 30,000 samples across all settings. The model depth varies with the input length $n$, with the number of layers set to $\lceil \log_2 n \rceil + 1$. Similar to Figure 3, we report the OOD loss for values three times larger than the training range, using 1,000 test samples. As illustrated in Figure 4, positional Transformers exhibit significantly lower OOD loss compared to standard Transformers across various sequence lengths. Naturally, for a fixed number of samples, the OOD loss slightly increases as the sequence length grows, indicating a need for more samples for longer sequences.

**Value generalization:** In this experiment, we use a similar setting to Section 6.1, but we fix the input length to $n = 8$ and train the models with 30,000 samples. We then evaluate the OOD loss across the different scale factors, each calculated using 1,000 samples. As shown in Figure 5, even in the fixed-length regime, standard Transformers struggle to value generalize, while positional Transformers have a much more stable performance across scales.

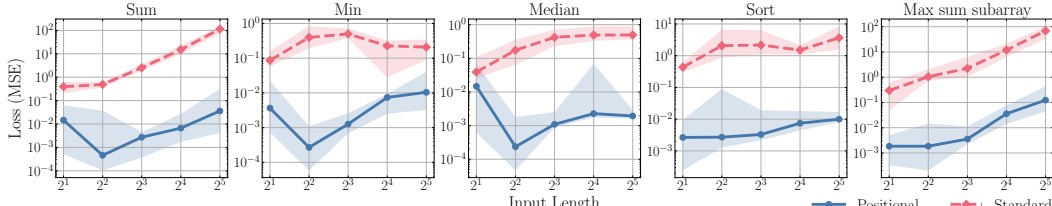

Figure 4: OOD loss for standard Transformers (red) and positional Transformers (blue) across different input lengths. The x-axis is the fixed input length on which the model was trained. Models are trained on the range $[-2, 2]$ with 30,000 samples and tested on $[-6, 6]$ with 1,000 samples.

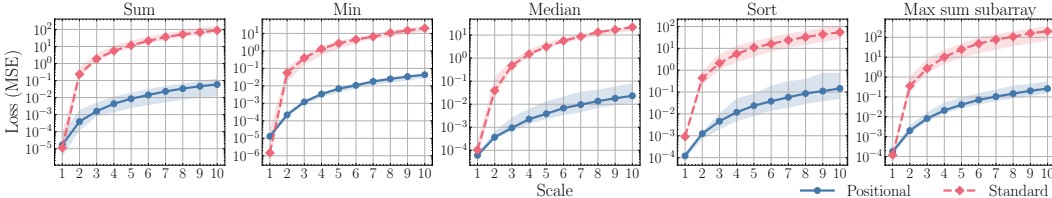

Figure 5: OOD loss for standard Transformers (red) and positional Transformers (blue) across all five tasks and length $n = 8$. The x-axis represents the OOD scale factor. The solid line and shaded area denote the median and the region between the $10^{\text{th}}$ and $90^{\text{th}}$ percentiles for 10 trials, respectively.

## 7 LIMITATIONS AND FUTURE WORK

Our present work shows strong evidence that the positional attention mechanism is a better alternative to standard attention in the context of neural algorithmic reasoning. However, more research is needed to uncover the true potential of this mechanism. We identify three main future research directions which we believe are important.

1. **OOD generalization theory:** It is very often the case that existing OOD generalization bounds are not tight (see Appendix F for an extended discussion). For specific tasks, there is often a gap between what theory says about the worst-case performance and what one observes empirically. This highlights the need for a more fine-grained analysis that will be able to capture the difference in OOD generalization capabilities among different architectures.

2. **Length generalization capability:** Our current proposal uses fixed positional encodings, making it difficult to test it on bigger length inputs. Designing positional encodings that can work with arbitrary input lengths will allow us to explore the length generalization capabilities of positional attention.

3. **Complementary tasks:** Testing the positional attention mechanism on complementary tasks, such as graph algorithms, requires special treatment. In particular, graph algorithms require that the model effectively process graph connectivity rather than merely treating it as input for the data matrix. Adopting positional attention in various architectures that use graph attention is an exciting future work.

A final potential research direction is understanding why the standard Transformer fails to OOD-generalize even when trained on (seemingly) simple tasks. In Appendix E we discuss some potential sources of issues that might be the underlying causes of poor OOD generalization in standard self-attention. For example, a potential problem seems to be related to the stability of self-attention weights against OOD data. In particular, the weights can be very sensitive to the scale of input values. We observe a dramatic change in attention weights as soon as we give the model the same input but scaled so that the values lie outside the domain of the training data. However, more research is needed before we can conclusively answer this question.

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

# APPENDIX

## A    SUPPLEMENTARY RESOURCES ON PARALLEL ALGORITHMS

### A.1    MASSIVELY PARALLEL COMPUTATION (MPC)

The MPC model is a computational model of the MapReduce framework (Dean & Ghemawat, 2008), widely used for computations over massive datasets. It defines a parallel computing model that jointly executes a function across multiple machines, each constrained by limited memory capacity. The MPC model is capable of representing various parallel algorithms (Im et al., 2023) and is more powerful than other established parallel models, such as parallel random-access machine (PRAM) (Andoni et al., 2018).

For completeness, we provide a simplified version of the definition of the MPC protocol by Andoni et al. (2018), which makes the connection to our PCOC model more apparent.

**Definition 5** (MPC protocol, Def. I.1 (Andoni et al., 2018), simplified)**.** Let $s$ be a parameter. There are $p \geq 1$ machines (processors), each with local memory of size $s$. The input is distributed on the local memory of some of the machines. The computation proceeds in rounds. In each round, each machine computes the data in its local memory and sends messages to other machines at the end of the round. The total size of messages sent or received by a machine in a round is bounded by $s$. In the next round, each machine only holds the received messages in its local memory. At the end of the computation, the output is in the memory of some machines.

#### A.1.1    RELATION BETWEEN PCOC AND MPC

**Why we do not use the MPC model.** We describe two main differences between MPC and PCOC, which justify introducing the latter. First, the communication within the MPC model may depend on the processed data. Our agnostic model removes this dependency by decoupling communication and computation. In particular, we assume that information required for communication (i.e. message destinations) is provided by an oracle, which, in practice, can be realized as a learning problem. Second, the original MPC definition contains assumptions about the relations between the memory size $s$, the input size, and the number of machines $p$, as well as an assumption about the number of machines to which the input gets distributed. Although these assumptions make sense when implementing algorithms for MPC in practice, and we could consider such assumptions for our simulation results, it is unclear what additional value they provide within the context of neural algorithmic reasoning. Thus, they are not part of the definition of the PCOC model.

**PCOC can simulate MPC.** For a given task, length $n$ and input Data, it is easy to observe that PCOC can simulate an algorithm on the MPC model that does not utilize the memory and processor restrictions mentioned above. In such cases, PCOC allows two different simulation approaches for an MPC protocol.

First, assuming the existence of an oracle $\mathsf{RCV}_{n,\mathtt{Data}}$ which has information about the communication at each round of a specific $R$-round MPC algorithm as well as the destinations of each element in the memories of the machines at each round, a PCOC algorithm on $n$ machines and $R$ rounds with the aforementioned oracle $\mathsf{RCV}_{n,\mathtt{Data}}$ can simulate an $R$ round MPC protocol on $n$ machines. At each round, the oracle essentially routes all data according to the underlying MPC algorithm's

**(Cumulative) sum/min.**          **Sorting (Odd-Even sort)**

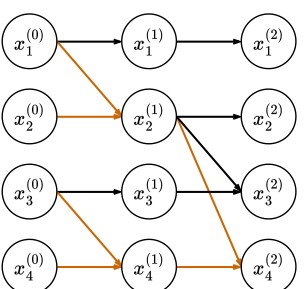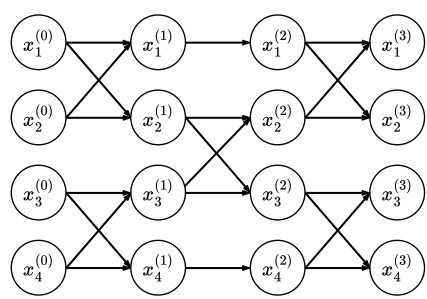

Figure 6: Illustration of the computational graph for algorithms such as (cumulative) sum/minimum (on the left) and sorting (on the right) for $n = 4$. Circles indicate the machines, each indexed by the subscript. The superscript indicates each round of the algorithm. At round 0, machine $i$ holds the $i$-th element of the input. The cumulative version of the sum/minimum algorithm includes all arrows (black and orange), while the non-cumulative version is represented only by the orange arrows.

requirements. The parameter $s$ for PCOC coincides with the one from the MPC model since no more data are routed among machines during the execution of the PCOC algorithm.

Alternatively, PCOC can simulate MPC with a fixed oracle at each round, though it is significantly less efficient. This is feasible even if the destinations are encoded within the data (despite not being used for routing). In this scenario, the oracle sends all relevant data to all machines, allowing each local function to determine which memory slots should be used in the local computation based on the destinations. However, this method is considerably more expensive as each machine requires significantly more memory than the simulated MPC protocol. Moreover, the local functions become more complex, as they must conditionally execute computations based on the destination of each memory slot.

### A.2 ILLUSTRATION OF PARALLEL ALGORITHMS

In this section, we further expand on the discussion of Section 5, stating that communication in several parallel algorithms depends only on the identification of the machines rather than their current values. To illustrate this, we provide some concrete examples.

These tasks are examples of those presented in Section 6. Note that these illustrations do not indicate the computational graphs derived by our architecture, as there are multiple ways to achieve the same goal, and they do not necessarily involve the neat graph structures shown in Figure 6. For a more in-depth analysis of the results obtained by our architecture, we refer the reader to Appendix C.

In these computational graphs, we represent each machine by a circle, distinguished by a subscript (from 1 to 4, since $n = 4$). Furthermore, we use superscripts to denote the rounds of the algorithm, with superscript 0 representing the initial stage where no computation or communication is performed. Note that no specific values are provided in these examples. This indicates that the correct results can be achieved by following the computational graph for any set of values. In the subsequent rounds, each machine receives information from other machines (including itself) and performs some computation. For each algorithm in Figure 6, we will briefly describe the computations involved, noting that this is not the main focus of the paper and serves only as motivating examples.

For the computation of the minimum and the summing function, each machine applies the minimum (or sum) operator to all incoming values from the previous round. By iteratively applying this operator over multiple rounds, machine 4 ultimately obtains the global minimum value (or total sum) across all $n$ entries, while the other machines hold cumulative minima (or sums) up to their positions. For the non-cumulative versions of these algorithms, the local computations are the same as the cumulative versions, and the communication paths are denoted in orange and form a binary tree pattern.

For sorting, the graph on the right of Figure 6 represents the odd-even transposition sort algorithm (Habermann, 1972). This algorithm works by alternating communication between adjacent machines, starting with the odd-indexed pairs (machines with indices 1 and 2, 3 and 4, etc.), then switching to even-indexed pairs (machines with indices 2 and 3, in this example). In stages where two machines communicate, the machine with the lower index picks the minimum value among the two, while the machine with the higher index picks the maximum. The procedure runs for a total of $n-1$ rounds, after which the values are sorted in ascending order.

## B  Expressivity results

### B.1  Hardmax patterns using positional attention

In this section, we show that the positional attention architecture in Equation (2) can approximate any unique hardmax pattern, a concept we define later in this section. We begin by stating the definition of the row-wise hardmax transformation for a $p \times q$ matrix $\mathbf{X}$ from Section 3:

$$\mathrm{hardmax}(\mathbf{X})_{i,j} = \begin{cases} 1 & \text{if } \mathbf{X}_{i,j} = \max_{k \in [q]} X_{i,k} \\ 0 & \text{otherwise} \end{cases} \quad \text{for } i \in [p], j \in [q], \tag{4}$$

where we implicitly extend the definition for vectors in $\mathbb{R}^n$ by viewing them as $1 \times n$ matrices.

We use the term *hardmax pattern* to refer to any matrix in the image of hardmax (i.e. a binary matrix with at least one non-zero element in every row). Furthermore, we use the term *unique hardmax pattern* to refer to hardmax patterns with exactly one non-zero element in every row. Unique hardmax patterns occur when the input matrix has a unique maximum value in every row.

We further define key concepts that will be used for the more formal re-statement of Lemma 1. Let the input have $n$ rows, and the binary positional encodings be defined by the positional encoding matrix $P = I_n$, therefore having $d_P = n$. Finally, let $T$ be a positive scalar that represents a temperature parameter that controls the approximation of softmax.

**Lemma 1.** *For any given $n \times n$ unique hardmax pattern $\bar{A}$, there exists a configuration of node positional attention parameters in Equation* (2) *and a temperature parameter $T$ such that the resulting softmax pattern $A$ approximates $\bar{A}$ to any desired level of accuracy. Formally, for any unique hardmax pattern $\bar{A}$ and any $\varepsilon > 0$, there exists some $T = T(\varepsilon) > 0$ such that the inequality $|\bar{A}_{i,j} - A_{i,j}| \le \varepsilon$ holds for all $i, j \in [n]$.*

*Proof.* Without loss of generality, we may assume $\varepsilon < 1$. We start by setting the node positional attention parameters to be $W_K = I_n$ and $W_Q = T(2\bar{A} - 1)$, where, $T > 0$ is our temperature scalar parameter and $\bar{A}$ is the target pattern. Since, in this construction, node positional encodings are set to be the identity, the inner-product $PW_Q W_K^\top P^\top$ reduces to $W_Q$, where each entry $(i,j)$ is $T$ if $\bar{A}_{i,j} = 1$, and $-T$ otherwise.

This inner product is passed to the softmax operator, resulting in the attention matrix $A$. For each $i, j \in [n]$ we separately analyze the following two cases:

1. Case $\bar{A}_{i,j} = 1$: In that case, the only non-zero element on the $i$-th row of $\bar{A}$ is $\bar{A}_{i,j}$, so we can express the difference as

$$\begin{aligned}
\bar{A}_{i,j} - A_{i,j} &= 1 - \frac{\exp((W_Q)_{i,i})}{\sum_{k=1}^n \exp((W_Q)_{i,k})} = 1 - \frac{\exp(T)}{\exp(T) + \sum_{k \neq j} \exp(-T)} \\
&\le 1 - \frac{\exp(T)}{\exp(T) + n\exp(-T)} = 1 - \frac{1}{1 + \exp(\ln n - 2T)} \\
&= \frac{\exp(\ln n - 2T)}{1 + \exp(\ln n - 2T)} \le \exp(\ln n - 2T)
\end{aligned}$$

2. Case $\bar{A}_{i,j} = 0$: Let $j_0 \neq j$ be the unique index for which $\bar{A}_{i,j_0} = 1$, and we can express the difference as:

$$A_{i,j} - \bar{A}_{i,j} = \frac{\exp((W_Q)_{i,j})}{\sum_{k=1}^{n} \exp((W_Q)_{i,k})} = \frac{\exp(-T)}{\exp(T) + \sum_{k \neq j_0} \exp(-T)}$$

$$= \frac{1}{\exp(2T) + n - 1} \leq \frac{1}{\exp(2T - \ln n) + 1} \leq \exp(\ln n - 2T)$$

In any case, we have that $|\bar{A}_{i,j} - A_{i,j}| \leq \exp(\ln n - 2T)$. Therefore, by taking $T \geq \frac{1}{2} \ln(n/\varepsilon)$, we have $|\bar{A}_{i,j} - A_{i,j}| \leq \varepsilon$. □

### B.2 POSITIONAL TRANSFORMERS SIMULATE PCOC

We begin this section by outlining the key concepts utilized in the routing protocol employed in our constructions. First, we describe the general structure of the input matrix $\mathbf{X}$.

#### B.2.1 ENCODING

**Input matrix:** In alignment with the PCOC model, the input matrix $\mathbf{X}$ represents $N$ machines, where each machine is denoted by a row in the matrix, and its local memory, $\text{MEM}_i \in \mathbb{T}^s$, is represented by the corresponding columns. The maximum size of data that any machine can send or receive is $s$ bits, with each bit corresponding to a column in $\mathbf{X}$.

However, the actual number of rows and columns in $\mathbf{X}$ differs from the number of machines and the local memory size for two reasons:

1. **Sink node**: A dummy node is introduced to facilitate all possible communication patterns in PCOC using positional attention. This is necessary because PCOC allows for the possibility of information not being sent to any receiving machine. This scenario is incompatible with the softmax formulation, which requires at least one non-zero entry. The dummy node serves as a sink, collecting all messages that do not have a destination. Consequently, the number of rows in $\mathbf{X}$ is $n = N + 1$.

2. **Unique node identifier:** Each machine also requires a unique identifier to enable element-wise local computations. To achieve this, we encode a unique scalar for each node in the last column of $\mathbf{X}$, resulting in a feature dimension of $d_X = s + 1$ for the input matrix.

As discussed in Section 5, in PCOC, routing is set by an oracle that decides how packets of data should be routed at each round. Under this framework, routing must be performed to prevent multiple data from being sent to the same destination. Since our construction relies on matrix operations, this leads to the following assumption:

**Assumption 1.** (No-collision). For any layer $\ell \in [L]$, no two different machines $i_1, i_2 \in [N]$ should route data to the same destination $i_3 \in [N]$ for the same column $j \in [s]$, where each column represents a bit of local memory across the nodes.

Note that this assumption does not limit the generality of our PCOC model. It only defines how data should be stored in the memory of each receiving machine, and any valid PCOC routing has a corresponding no-collision configuration of bits that realizes it due to the restriction on the total size of received data. As demonstrated in the constructive proof, this directly influences the sparsity pattern generated by each attention head.

**Positional encodings:** As previously mentioned, although the connectivity at each layer may vary, the positional encodings remain consistent across all layers. Our architecture simulates MPC using node positional encodings with dimension $d_P = n$ by setting $P = I_n$, with each positional encoding functioning as a standard basis vector.

#### B.2.2 SIMULATION RESULTS

We now demonstrate that, with the established encoding, the architecture provided in Section 4 can simulate the execution of any PCOC instance. Each round of such a PCOC instance can be decomposed into two stages: communication and computation. Our objective is to provide existential results for both stages.

In the communication stage, routing assigns destination machines for each message. In our architecture, this assignment is analogously captured by the attention weights, which determine whether a message should be received by a node using binary values.

The no-collision assumption ensures that all routing patterns can be represented by unique hardmax patterns. As expressed in Lemma 1, since any unique hardmax pattern can be approximated by our attention layer using softmax, for simplicity, the subsequent proofs use hardmax instead of softmax. With all details now established, we re-state our main simulation result:

**Theorem 1.** *Consider a PCOC instance* P *with $R$ rounds, $N$ machines with local memory $s$, and data type $\mathbb{T} = \mathbb{R}$. Let $\mathcal{M}$ be a model following the architecture in equation 1 with $n = N + 1$ nodes, $R$ layers and $s$ attention heads. Then, for any instance* P *with Borel measurable local functions, there exists a configuration of $\mathcal{M}$ that approximates* P *to any desired degree of accuracy.*

*Proof.* Despite the desired degree of accuracy being influenced by the number of rounds performed, it suffices to show that one layer of our architecture can simulate one round of PCOC. The same constructive arguments can be extended to more rounds, ensuring the overall degree of approximation is respected. To this end, we begin the proof with the communication stage.

**Communication:** In PCOC, communication is encoded as routing patterns determined by the oracle. At round $\ell \in [R]$, we denote by $H^{(\ell)} = \{((i, j), K) \mid i, j \in [N], K \in \mathcal{P}([s])\}$ the set of valid routing patterns provided by the oracle. This set specifies that the data at positions $K$ in the local memory of machine $i$ must be sent to machine $j$ at the same position. A valid routing pattern requires that no collisions occur (i.e., no two triplets in $H^{(\ell)}$ should have the same destination $j$ and memory position $k \in K$). We further denote by $H_z^{(\ell)} = \{((i, j), z) \mid ((i, j), K) \in H^{(\ell)}, z \in K\}$ the subset of routing patterns corresponding to position $z$ in local memory.

The first part of our result constructively demonstrates that positional attention can reproduce any valid routing set by the oracle that adheres to the PCOC model. We construct $s$ attention heads indexed by $h \in [s]$, which handle routing for the corresponding subset $H_h$.

For clarity in the construction phase, we introduce an augmented set to simplify notation. We begin by extracting the set of source nodes for each set $H_z^{(\ell)}$, denoted as $I_z^{(\ell)} = \{i \mid ((i, j), z) \in H_z^{(\ell)}, j \in [N], z \in [s]\}$. Next, we create a complement set $\bar{H}_z^{(\ell)}$, which routes all unused sources (i.e., those not in $I_z^{(\ell)}$) to the sink node labeled $n = N + 1$. We denote this complement set by $\bar{H}_z^{(\ell)} = \{(i, n, z) \mid i \in [n] \setminus I_z^{(\ell)}\}$. Finally, we define the union of these sets as $\hat{H}_z^{(\ell)} = H_z^{(\ell)} \cup \bar{H}_z^{(\ell)}$.

The attention parameters are then set as follows:

$$\left(W_K^{(\ell,h)}\right)_{i,j} = \begin{cases} 1 & \text{if } i = j, \\ 0 & \text{otherwise,} \end{cases} \quad \text{and} \quad \left(W_Q^{(\ell,h)}\right)_{i,j} = \begin{cases} 1 & \text{if } (j, i, h) \in \hat{H}_h^{(\ell)} \\ 0 & \text{otherwise,} \end{cases}$$

In this construction, we first observe that both the node positional encodings and the key matrix are identity matrices, reducing the inner product in attention to be solely defined by the query matrix. The query matrix is then designed to encode the source node $i$ as a standard basis vector in the row corresponding to the destination node $j$. This effectively represents the routing set $\hat{H}_h^{(\ell)}$ as a binary matrix, which is also preserved after applying hardmax. Additionally, the no-collision assumption, combined with the sink node strategy, ensures exactly one non-zero entry in the first $N$ rows of the attention weights matrix for each attention head $h$.

For the value and output transformation, we set all value matrices $W_V^{(\ell,h)}$ to be the identity $I_{d_X}$ and define the output matrix $W_O^{(\ell)} \in \mathbb{R}^{(H \cdot d_X) \times (d_X - 1)}$ as follows:

$$\left(W_O^{(\ell)}\right)_{i,j} = \begin{cases} 1 & \text{if } i = k + (h-1)s, \ j = h \\ 0 & \text{otherwise.} \end{cases}$$

Here, the output matrix $W_O^{(\ell)}$ ensures that only the correct memory position receives the information and places it in the corresponding column. Note that since the outputs of the attention heads are concatenated before being processed by $W_O^{(\ell)}$, the values along the rows of the output matrix also depend on the attention head.

We now focus on the computation stage for the second part of the proof.

**Computation:** At round $\ell \in [R]$ of a PCOC model, let $[\phi_i^{(\ell,z)}]_{i=1}^n$ be the local functions applied by each machine $i \in [N]$ and let $\phi_n^{(\ell,z)}$ correspond to the function of the augmented sink node, which effectively erases all data received. Each function $\phi_i^{(\ell,z)}$ operates on received data in each machine's local memory, which corresponds to the output of the attention layer, denoted by $z$ and outputs a vector of the same dimension $s$, that is, $\phi_i^{(\ell,z)} : \mathbb{R}^s \to \mathbb{R}^s$.

Furthermore, let $\phi^{(\ell,x)} : \mathbb{R}^{d_X} \to \mathbb{R}^{d_X}$ be a function common to all nodes, which operates solely on the residual connection $x$. This function outputs a vector where all entries are zero except the last entry. The value in this last entry corresponds to the unique node identifier extracted from the residual input $x$.

We aim to approximate both $\phi_i^{(\ell,z)}$ and $\phi^{(\ell,x)}$ using neural networks. To this end, we define the combined function $\phi_i^{(\ell)} : \mathbb{R}^{d_X+s} \to \mathbb{R}^{d_X}$ by:

$$\phi_i^{(\ell)}(z_i \oplus x_i) := \phi^{(\ell,x)}(x_i) + \phi_i^{(\ell,z)}(z_i) \oplus 0, \tag{5}$$

where $z_i \oplus x_i$ denotes the concatenation of output of $z_i \in \mathbb{R}^s$ and $x_i \in \mathbb{R}^{d_X}$. We further augment the output of $\phi_i^{(\ell,z)}$ with a zero scalar to match the dimension $d_X = s + 1$.

Let $[\hat{\phi}_i^{(\ell)}]_{i=1}^n$ correspond to multilayer perceptrons (MLPs), each applied to each input $z_i \oplus x_i$. By invoking universal approximation results such as those by Cybenko (1989); Hornik et al. (1989), we assert that as long as the local functions $\phi_i^{(\ell)}$ are Borel measurable, there exist neural networks $\hat{\phi}_i^{(\ell)}$ that can approximate the functions $\phi_i^{(\ell)}$ to any desired degree of accuracy. Additionally, note that the function $\phi_n^{(\ell,z)}$ of the sink node, as well as the function $\phi^{(\ell,x)}$ that operates on the residual connection, are both linear and therefore Borel measurable.

The final step in this argument is to relate these approximations to the proposed architecture in Section 4. Specifically, we use the MLP $\Phi^{(\ell)}$ in Equation (1) and leverage the aforementioned universality results to approximate all the element-wise functions $\phi_i^{(\ell)}$.

A crucial aspect of this step is the need for the input of each machine to be uniquely identifiable. This ensures that a single model can injectively encode multiple functions. Intuitively, it guarantees that each approximation of the local function can identify that it is processing the right row. The unique identification of each machine is guaranteed by the scalar encodings of every node, which, regardless of the contents in local memory, ensure that the input rows are unique. Therefore, the function that $\Phi^{(\ell)}$ has to approximate is a piecewise Borel function with each branch being one of the $\phi_i^{(\ell)}$, based on the unique machine identifier. Such function is Borel measurable, and so the universal approximation results of Hornik et al. (1989) hold, guaranteeing the existence of the desired MLP $\Phi^{(\ell)}$.

This demonstrates that our neural network architecture can emulate the computations performed by the local functions $\phi_i^{(\ell,z)}$ acting on the output of the attention layer (with their outputs zero-padded to match the required dimension) and the function $\phi^{(\ell,x)}$ acting on the residual connection, even though they act on distinct parts of the input.

Therefore, we establish that our proposed architecture can approximate the computations in each round of the PCOC model. $\qquad\square$

An important observation is that the computational model and expressive results for the proposed architecture are specific to a fixed input length $n$. Furthermore, one could extend such results to a model with communication and local computations that also consider the input length as an input. For local computations, proof in Theorem 1 can also cover such cases, provided that the information about the length is also encoded in the input. For communication, we present the following remark.

**Remark 1.** *For any collection of unique hardmax patterns $\{\bar{A}^{(k)}\}_{k=1}^n$, where $\bar{A}^{(k)}$ is $k \times k$, there exists a configuration of node positional attention parameters in Equation* (2) *and a temperature*

parameter $T$ such that the resulting softmax patterns $\{A^{(k)}\}_{k=1}^n$ approximate each $\bar{A}^{(k)}$ to any desired level of accuracy. Formally, for any collection of unique hardmax patterns $\{\bar{A}^{(k)}\}_{k=1}^n$ and any $\varepsilon > 0$, there exists a temperature parameter $T = T(\varepsilon) > 0$ and corresponding attention parameters such that for all $i, j \in [n]$ and for all $k \in [n]$, the following inequality holds: $\left| \bar{A}_{i,j}^{(k)} - A_{i,j}^{(k)} \right| \le \varepsilon$.

The proof of this remark relies on a slight modification of the proof of Lemma 1. However, to cover all possible patterns, the embedding dimension of positional encodings should also encode the input length and be of the order of $O(n^3)$. Although this embedding dimension is theoretically large, in practice, one does not need as many dimensions for positional encodings, as demonstrated in the variable length experiments in Section 6.

### B.3 Softmax patterns using positional attention

We conclude the discussion on expressivity by showing a final, standalone, result, namely that the positional attention architecture in Equation (2) can represent any softmax pattern. We begin by stating the definition of the row-wise softmax transformation for a matrix $\mathbf{X} \in \mathbb{R}^{p \times q}$:

$$\text{softmax}(\mathbf{X})_{i,j} = \frac{\exp(\mathbf{X}_{i,j})}{\sum_{k=1}^q \exp(\mathbf{X}_{i,k})} \quad \text{for } i \in [p], j \in [q] \tag{6}$$

As with hardmax, the definition is implicitly extended to vectors in $\mathbb{R}^n$ by viewing them as $1 \times n$ matrices. The image of the softmax function is the set of row-stochastic matrices with entries in $(0, 1)$. Indeed, it is easy to see that when softmax is applied to a matrix, the resulting matrix satisfies the above property. On the other hand, for a matrix $\mathbf{B} = (b_{ij})_{i \in [p], j \in [q]}$ with $b_{ij} \in (0, 1)$ and $\sum_{j \in [q]} b_{ij} = 1$ for all $i \in [p]$ we have $\text{softmax}(\tilde{\mathbf{X}}) = B$ where $\tilde{\mathbf{X}}_{i,j} = \ln(b_{ij})$. We use the term *softmax pattern* to refer to any matrix in the image of softmax.

Consider attention weights $A^{(\ell,h)}$ that are defined by positional encodings in equation 2. Let $B \in (0,1)^{n \times n}$ be a softmax pattern. We would like to find parameters $W_Q^{(\ell,h)}$ and $W_K^{(\ell,h)}$ that induce $B$, that is $A^{(\ell,h)} = B$. From the properties of softmax described above, it suffices to solve the matrix equation $(PW_Q^{(\ell,h)}) \cdot (PW_K^{(\ell,h)})^\top = \tilde{B}$ where $\tilde{B}_{ij} = \ln(B_{ij})$. This equation always has a solution when $d_P = n$ and $P$ is invertible. We summarize the above observation in the following expressivity remark:

**Remark 2** (Positional attention is expressive). *Positional attention can realize all softmax patterns at every layer provided that $d_P = n$ and $P$ is invertible. This is not necessarily true in the case of standard attention where, in subsequent layers, positional encodings are modified and, therefore, not guaranteed to be linearly independent.*

## C Experiments

This section presents detailed results for the experiments reported in Section 6. All experiments in this section are performed on lists of fixed length. Briefly, we examine the following:

- The capability of more compact standard Transformers to achieve value generalization.
- The capability of a standard Transformer with Rotary Positional Embedding (RoPE) to achieve value generalization.
- The OOD performance of standard and positional Transformers using different types of positional encodings.
- How the number of training samples affects value generalization.
- The OOD performance of standard and positional Transformers for various fixed input lengths.
- The OOD performance of standard and positional Transformers when evaluated on a relational task with mixed-type inputs.

- The OOD performance of standard and positional Transformers when evaluated on the previous relational task with additional noise.

Finally, we present two new transformations that label each prediction as "correct" or "incorrect" and use the labels to calculate the OOD accuracies of both standard and positional Transformer for all tasks.

Unless otherwise specified, all configurations are consistent with those described in Section 6.2. Both training and testing lists are generated using the sampling strategy discussed in Section 6.

## C.1  VALUE GENERALIZATION OF COMPACT TRANSFORMERS

This section presents additional value generalization results for simpler models, aiming to rule out potential overfitting caused by the excessive complexity of the standard Transformer. We examine two configuration variants: one with $\log n + 1$ layers (4 layers) and another with a single layer. The plots also illustrate the outcomes for different hidden dimensions in the MLP. We report the value generalization results for the cumulative sum (Figure 7) and cumulative minimum (Figure 8) tasks. As observed, in both cases, the OOD performance deteriorates as the network size decreases. Although single-layer networks exhibit slightly better performance, they remain inferior to the performance of positional attention reported in the main paper.

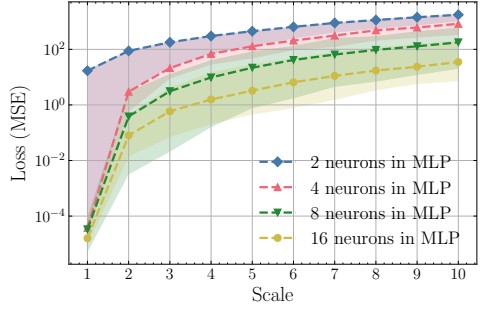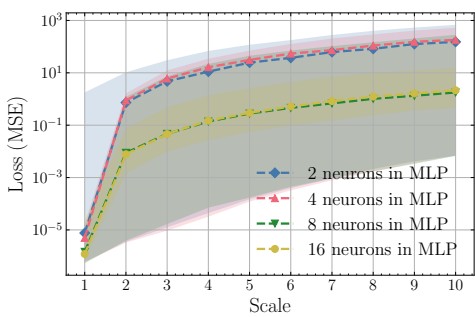

Figure 7: OOD loss for various standard Transformer models on the cumulative sum task with fixed length ($n = 8$). The left plot displays results for models with 4 layers, while the right plot shows results for single-layer models, both featuring varying hidden dimensions in the MLPs. The x-axis represents the out-of-distribution scale factor, indicating the distance from the training distribution. The solid lines and shaded areas denote the median and the regions between the 10[th] and 90[th] percentiles across ten trials, respectively.

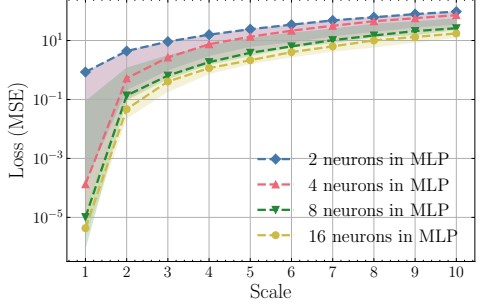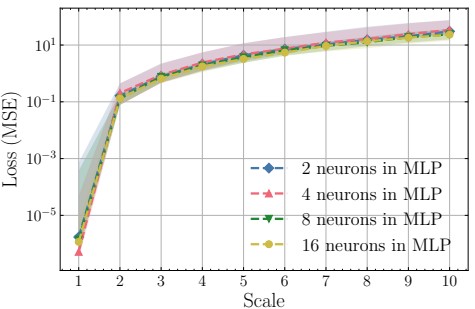

Figure 8: OOD loss for various standard Transformer models on the cumulative minimum task with fixed length ($n = 8$). The left plot displays results for models with 4 layers, while the right plot shows results for single-layer models, both featuring varying hidden dimensions in the MLPs. The x-axis represents the out-of-distribution scale factor, indicating the distance from the training distribution. The solid lines and shaded areas denote the median and the regions between the 10[th] and 90[th] percentiles across ten trials, respectively.

## C.2 VALUE GENERALIZATION FOR TRANSFORMERS WITH ROTARY POSITIONAL EMBEDDING (RoPE)

We compare Positional Transformers with standard Transformers using Rotary Positional Embedding (RoPE) (Su et al., 2024), a widely adopted technique in natural language processing contexts, which has also been applied to algorithmic tasks (Bounsi et al., 2024). Even though RoPE manages to decrease the OOD test loss, this improvement is not enough to claim value generalization. Our architecture still performs significantly better in every task. The results for this experiment are presented in Figure 9.

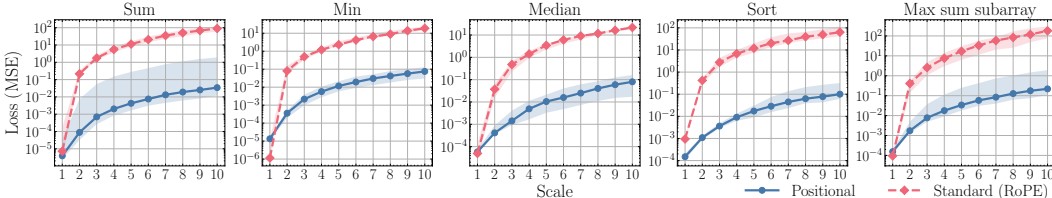

Figure 9: OOD loss (measured as mean squared error, MSE) for standard Transformers with RoPE (red) and positional Transformers (blue) across all five tasks for fixed length ($n = 8$). The x-axis represents the OOD scale factor, indicating the distance from the training distribution. The solid line and shaded area denote the median and the region between the $10^{th}$ and $90^{th}$ percentiles over ten trials, respectively.

## C.3 VALUE GENERALIZATION FOR TRANSFORMERS WITH ALTERNATIVE POSITIONAL ENCODINGS

We compare the performance of Positional Transformers and standard Transformers using alternative positional encodings, specifically binary and sinusoidal encodings Vaswani et al. (2017). For binary positional encodings, we use $\lceil \log_2 n \rceil$ dimensions, where each entry represents the binary encoding of its index in $\lceil \log_2 n \rceil$ bits, with zeros encoded as $-1$. The result for binary encodings is shown in Figure 10. For sinusoidal positional encodings, we follow the strategy outlined in Vaswani et al. (2017), with the encoding dimension set to $\lceil n/2 \rceil$. The result for sinusoidal encodings is shown in Figure 11. From an expressivity perspective, while these encodings are less expressive than one-hot positional encodings, they maintain consistent out-of-distribution (OOD) performance across all ranges. Furthermore, Positional Transformers outperform standard Transformers in every task tested.

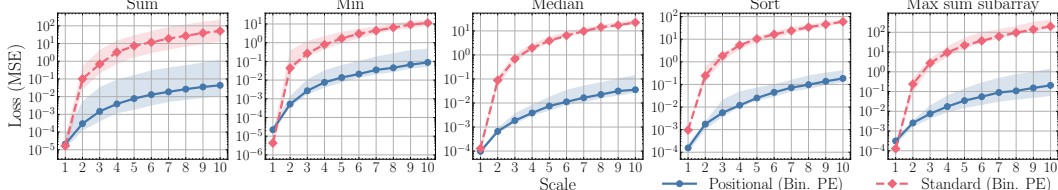

Figure 10: OOD loss (measured as mean squared error, MSE) for standard Transformers (red) and positional Transformers (blue) using binary positional encodings across all five tasks for fixed length ($n = 8$). The x-axis represents the OOD scale factor, indicating the distance from the training distribution. The solid line and shaded area denote the median and the region between the $10^{th}$ and $90^{th}$ percentiles over ten trials, respectively.

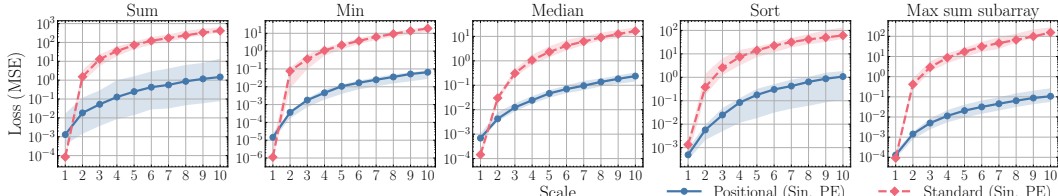

Figure 11: OOD loss (measured as mean squared error, MSE) for standard Transformers (red) and positional Transformers (blue) using sinusoidal positional encodings across all five tasks for fixed length ($n = 8$). The x-axis represents the OOD scale factor, indicating the distance from the training distribution. The solid line and shaded area denote the median and the region between the 10th and 90th percentiles over ten trials, respectively.

### C.4 SAMPLE SIZE VS. VALUE GENERALIZATION EXPERIMENTS

In this section, we provide detailed results showcasing the training, validation, and OOD test performance for each of the five tasks as a function of the number of training samples used. From the results, we can draw two conclusions about the behavior of the models as the number of samples increases. First, both modes achieve better in-distribution performance. Second, only the positional Transformer achieves better OOD performance. The results for this experiment are presented in Figures 12 to 16.

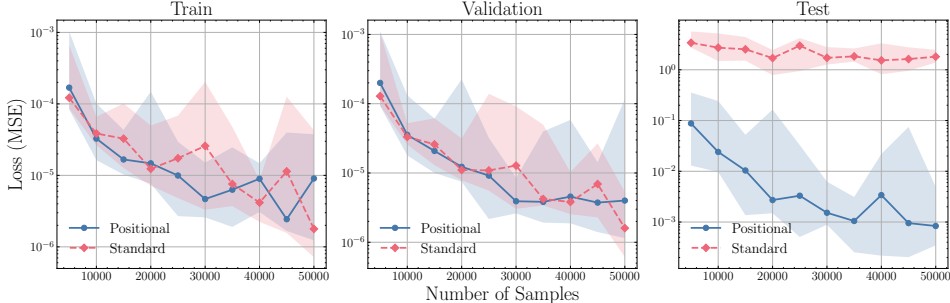

Figure 12: Training, validation, and test performance for the summing task are shown for standard Transformers (red) and positional Transformers (blue) as a function of the number of training samples (indicated on the x-axis). Models are trained on the range $[-2, 2]$ with varying training set sizes. Validation is performed on the same domain, and testing is conducted on an extended domain, $[-6, 6]$, each using 1,000 samples.

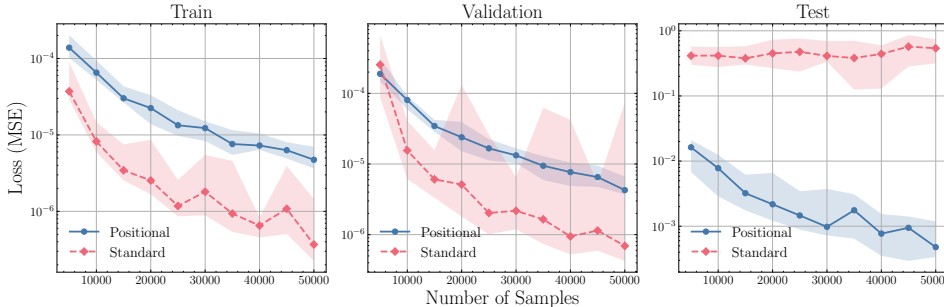

Figure 13: Training, validation, and test performance for the minimum task are shown for standard Transformers (red) and positional Transformers (blue) as a function of the number of training samples (indicated on the x-axis). Models are trained on the range $[-2, 2]$ with varying training set sizes. Validation is performed on the same domain, and testing is conducted on an extended domain, $[-6, 6]$, each using 1,000 samples.

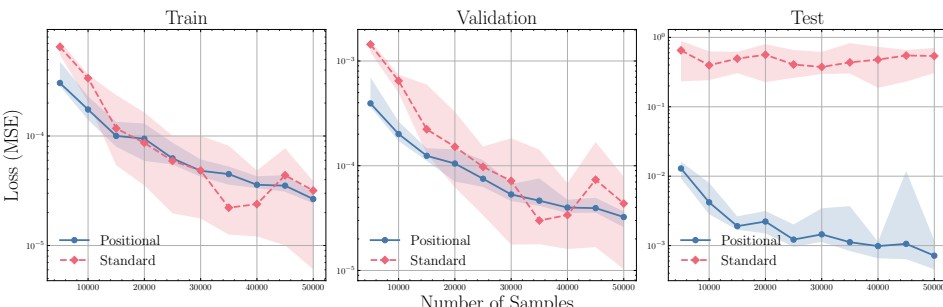

Figure 14: Training, validation, and test performance for the median task are shown for standard Transformers (red) and positional Transformers (blue) as a function of the number of training samples (indicated on the x-axis). Models are trained on the range $[-2, 2]$ with varying training set sizes. Validation is performed on the same domain, and testing is conducted on an extended domain, $[-6, 6]$, each using 1,000 samples.

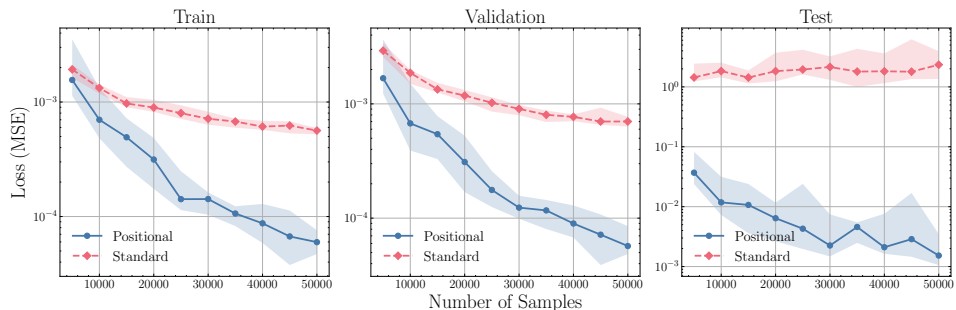

Figure 15: Training, validation, and test performance for the sorting task are shown for standard Transformers (red) and positional Transformers (blue) as a function of the number of training samples (indicated on the x-axis). Models are trained on the range $[-2, 2]$ with varying training set sizes. Validation is performed on the same domain, and testing is conducted on an extended domain, $[-6, 6]$, each using 1,000 samples.

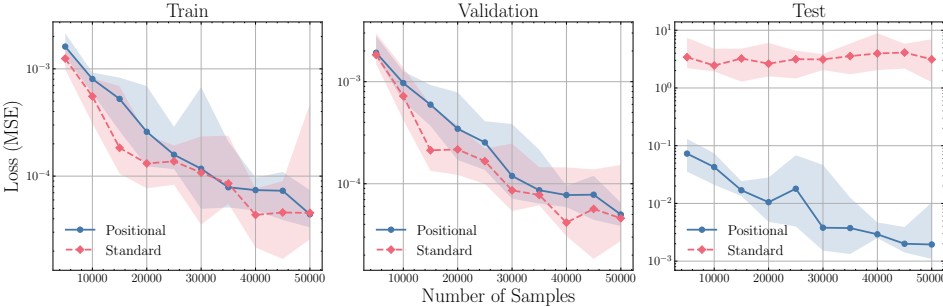

Figure 16: Training, validation, and test performance for the maximum sum subarray task are shown for standard Transformers (red) and positional Transformers (blue) as a function of the number of training samples (indicated on the x-axis). Models are trained on the range $[-2, 2]$ with varying training set sizes. Validation is performed on the same domain, and testing is conducted on an extended domain, $[-6, 6]$, each using 1,000 samples.

## C.5 INPUT LENGTH VS. VALUE GENERALIZATION EXPERIMENTS

In this section, we validate the robust performance of our architecture across increasing input lengths. We present detailed results showing the training, validation, and OOD test performance

for each of the five tasks as a function of input length. For each task, both models were trained on lists with fixed input lengths of 2, 4, 8, 16, and 32. As input length increases, both models' in-distribution and out-of-distribution performance decreases. However, the positional Transformer maintains good OOD performance even for inputs of length $n = 32$, whereas the standard Transformer's OOD performance remains unsatisfactory even for inputs of length $n = 2$. The results for this experiments are presented in Figures 17 to 21.

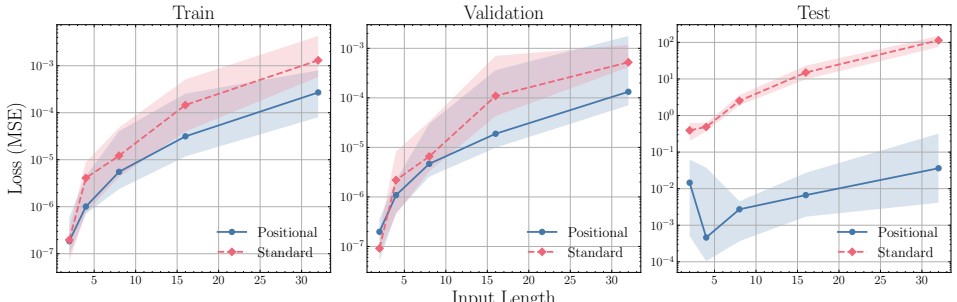

Figure 17: Training, validation, and test performance for the summing task are shown for standard Transformers (red) and positional Transformers (blue) across different input lengths. The x-axis indicates the fixed input length on which the model was trained. Models are trained on the range $[-2, 2]$ with 30,000 samples, validated on the same domain, and tested on an extended domain, $[-6, 6]$, each with 1,000 samples.

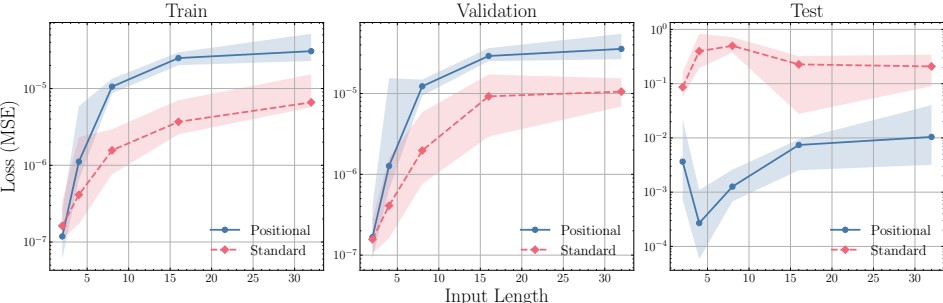

Figure 18: Training, validation, and test performance for the minimum task are shown for standard Transformers (red) and positional Transformers (blue) across different input lengths. The x-axis indicates the fixed input length on which the model was trained. Models are trained on the range $[-2, 2]$ with 30,000 samples, validated on the same domain, and tested on an extended domain, $[-6, 6]$, each with 1,000 samples.

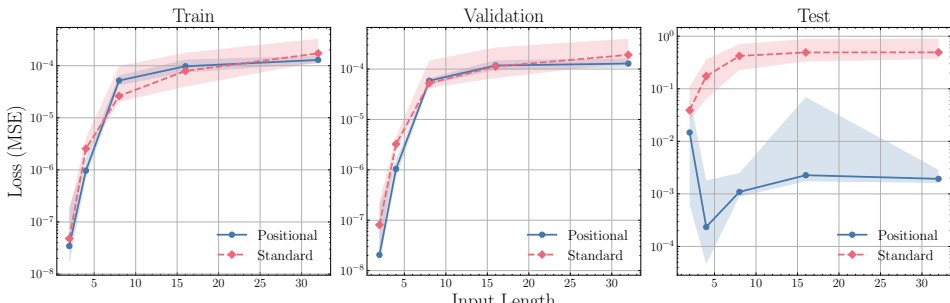

Figure 19: Training, validation, and test performance for the median task are shown for standard Transformers (red) and positional Transformers (blue) across different input lengths. The x-axis indicates the fixed input length on which the model was trained. Models are trained on the range $[-2, 2]$ with 30,000 samples, validated on the same domain, and tested on an extended domain, $[-6, 6]$, each with 1,000 samples.

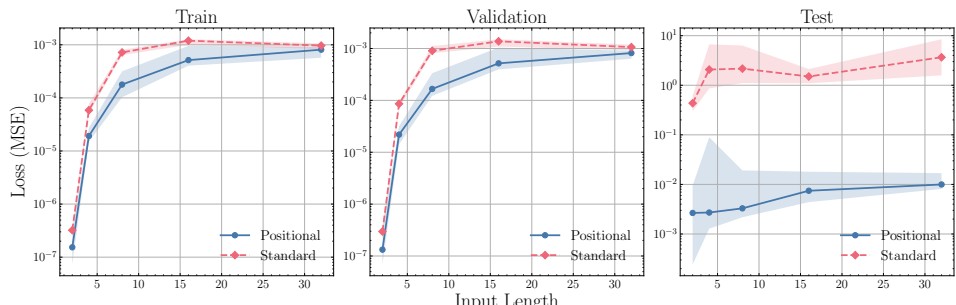

Figure 20: Training, validation, and test performance for the sorting task are shown for standard Transformers (red) and positional Transformers (blue) across different input lengths. The x-axis indicates the fixed input length on which the model was trained. Models are trained on the range $[-2, 2]$ with 30,000 samples, validated on the same domain, and tested on an extended domain, $[-6, 6]$, each with 1,000 samples.

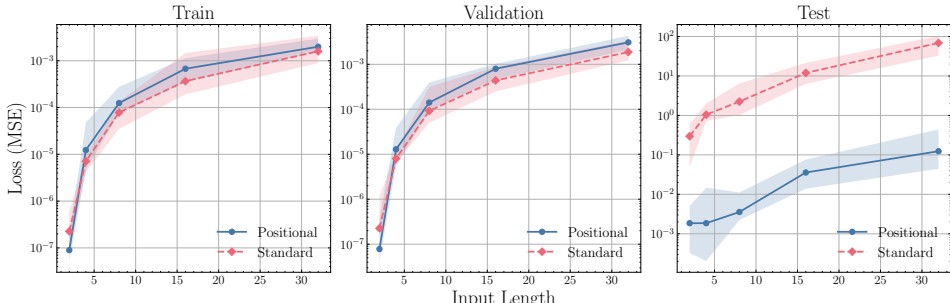

Figure 21: Training, validation, and test performance for the maximum sum subarray task are shown for standard Transformers (red) and positional Transformers (blue) across different input lengths. The x-axis indicates the fixed input length on which the model was trained. Models are trained on the range $[-2, 2]$ with 30,000 samples, validated on the same domain, and tested on an extended domain, $[-6, 6]$, each with 1,000 samples.

## C.6 A RELATIONAL TASK WITH MIXED-TYPE INPUTS

In this section, we evaluate the OOD performance of standard and positional Transformers on a relational task that involves numerical and textual data. We begin by presenting an example of a training and a testing sample for this task and then proceed with a formal description.

Training sample:

```
Input = ['Cat2', 3.45, 'Cat5', 1.23, 'Cat7', 0.65, 'Cat8', 2.23, 'Cat11',
4.10, 'Cat13', 1.10, 'Cat14', 0.10, 'Cat20', 2.75, 'Find min of Cat5, Cat8,
Cat11 and Cat20']
Output = 1.23
```

Test sample:

```
Input = ['Cat23', 7.28, 'Cat24', 33.5, 'Cat28', 9.17, 'Cat30', 55.90,
'Cat31', 23.70, 'Cat33', 12.47, 'Cat34', 8.45, 'Cat40', 1.50, 'Find min of
Cat28, Cat31, Cat33 and Cat40']
Output = 1.50
```

Note that the categories and the range of values change between the training and test data. Additionally, we experiment with the following tasks: minimum (min), sum, and a multi-tasking task that combines minimum (min) and maximum (max). For multi-tasking, we choose minimum and maximum since they are opposites.

This task requires pattern matching, since the categories between the train and test data are different. It requires conditional reasoning, since the queries are about a sub-set of the categories in the samples. Finally, it requires algorithmic reasoning to compute the output.

**Task description:** Let $n, k \in \mathbb{N}$ with $n \geq k$. The input to the model is a list consisting of text and numerical values of the following form:

$$['Cati_1', v_{i_1}, 'Cati_2', v_{i_2}, \ldots, 'Cati_n', v_{i_n}, 'Find \{type\} of Catj_1, Catj_2, \ldots, Catj_{k-1} and Catj_k']^3$$

where $i_1, \ldots, i_n \in \mathbb{N}, v_{i_1}, \ldots, v_{i_n} \in \mathbb{R}_{\geq 0}, j_1, \ldots, j_k \in \{i_1, \ldots, i_n\}$, and $\{type\}$ is one of $'min', 'max',$ $'sum'$. Notice that this setting allows for one model to potentially process multiple query types. In fact, we present one such experiment later. The answer to the query is either the minimum, maximum, or sum of the set $\{v_{j_1}, \ldots, v_{j_k}\}$ depending on $\{type\}$. As a real-world use case, the categories could correspond to the various types of expenses of a company, and the model could be used by data analysts to perform common aggregation tasks.

**OOD generalization:** For this task, we measure OOD generalization in two ways (simultaneously):

1. We test using category identifiers (i.e $i_1, i_2, \ldots, i_n$) that the models haven't encountered during training, and

2. The range of values $v_1, v_2, \ldots, v_n$ used for testing is larger than the range used for training.

**Experimental setting:** We fix $n = 8$ and $k = 4$. Both standard and positional Transformers consist of 3 layers, two attention heads per layer, and an embedding dimension of 32. We employ 2-layer MLPs with hidden dimension 32. All characters of the non-numeric parts of the input are tokenized and passed through an embedding layer, while the numeric ones are passed through a linear layer. The category identifiers $i_1, i_2, \ldots, i_8$ are sampled randomly from $\{1, 2, \ldots, 20\}$ for training and $\{21, 22, \ldots, 40\}$ for testing. The query category identifiers are then sampled randomly from $\{i_1, i_2, \ldots, i_8\}$. The values $v_{i_1}, \ldots, v_{i_8}$ are sampled using the technique of Section 6 from $[0, 5]$ for training and from $[0, 5c]$ where $c \in \{1, 2, \ldots, 10\}$ is the scaling factor for testing. We also apply the rejection step of Section 6 when generating testing samples. Both models are trained using Adam with a sample size of 50.000, a batch size of 1024, and an initial learning rate of $5 \cdot 10^{-4}$. The training runs for a total of 150 epochs, and a learning rate scheduler is employed. We present the median mean squared error (MSE) OOD loss as well as the median Mean Absolute Percentage Error (MAPE) OOD loss across five runs for three different variations of the task:

- Both models are trained and tested exclusively on prompts where $\{type\}$ is $'min'$.
- Both models are trained and tested exclusively on prompts where $\{type\}$ is $'sum'$.

---

[3]The ellipses are not part of the prompt

- Both models are trained and tested exclusively on prompts where {`type`} is either '`min`' or '`max`'. When generating a training or testing sample, {`type`} is chosen at random. We refer to this experiment as "multitasking" since it allows a single trained model to process different query types. In fact, the choice of minimum and maximum as the two query types is, in some sense, an "extreme" case, given the "opposite" nature of minimum and maximum operations.

The median MSE and median MAPE losses for the variations described above are presented in Figure 22 and Figure 23, respectively. We observe that, even in this complex relational task with mixed-type inputs, the positional Transformer still significantly outperforms the standard Transformer (and more so in the harder multitasking regime), demonstrating the potential utility of our application for certain real-world applications.

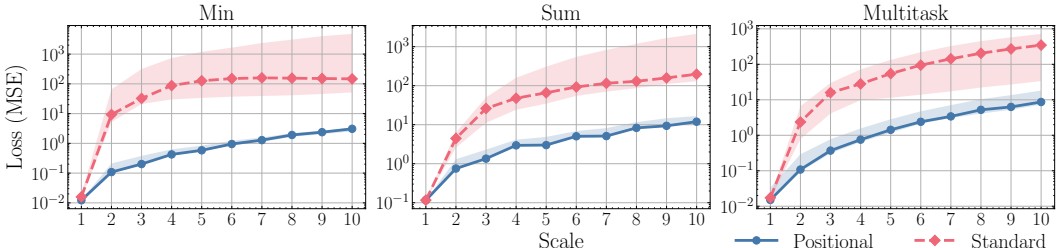

Figure 22: OOD loss (measured as mean squared error, MSE) for standard Transformers (red) and positional Transformers (blue) across all three variations. The x-axis represents the OOD scale factor. The solid line and shaded area denote the median and the region between the 10th and 90th percentiles over five trials, respectively.

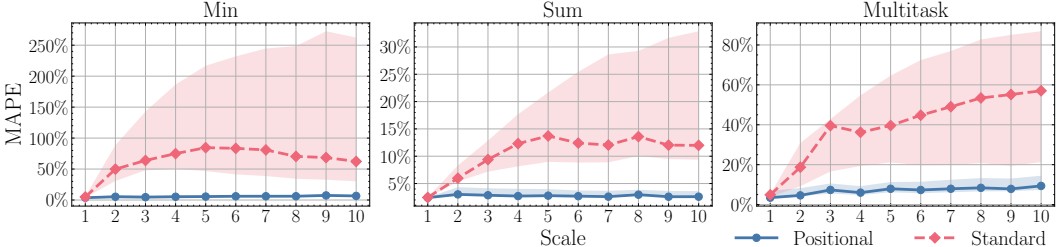

Figure 23: OOD loss (measured as mean absolute percentage error, MAPE) for standard Transformers (red) and positional Transformers (blue) across all three variations. The x-axis represents the OOD scale factor. The solid line and shaded area denote the median and the region between the 10th and 90th percentiles over five trials, respectively.

## C.7 A RELATIONAL TASK WITH MIXED-TYPE INPUTS AND IRRELEVANT STRUCTURE

In this section, we revisit the task of Appendix C.6 and evaluate the OOD performance of standard and positional Transformers in the presence of irrelevant categories in the input injected at random positions. The random categories in the train set are different than the ones in the test set. The model needs to figure out what is a useful category, and then perform pattern matching, conditional and computational reasoning. We begin by presenting an example of a training and a testing sample for this task and then proceed with a formal description.

Training sample:

```
Input = ['Cat2', 3.45, 'Cat+7', 'Cat5', 1.23, 'Cat7', 0.65, 'Cat8', 2.23,
'Cat11', 'Cat-8', 4.10, 'Cat13', 1.10, 'Cat14', 0.10, 'Cat20', 2.75, 'Find
min of Cat5, Cat8, Cat11 and Cat20']
Output = 1.23
```

Test sample:

```
Input = ['Cat23', 7.28, 'Cat24', 33.5, 'Cat28', 9.17, 'Cat30', 55.90,
'Cat31', 'Cat*24', 23.70, 'Cat33', 12.47, 'Cat34', 8.45, 'Cat_40', 'Cat40',
```

```
1.50, 'Find min of Cat28, Cat31, Cat33 and Cat40']
Output = 1.50
```

Note that in this setting, the categories, the range of values, as well as the type of irrelevant structure changes between the training and test data. Additionally, we experiment with the following tasks (same as the ones in Appendix C.6): minimum (min), sum, and a multi-tasking task that combines minimum (min) and maximum (max). As before, for multi-tasking, we choose minimum and maximum since they are opposites.

This task requires distinguishing relevant from irrelevant categories, pattern matching, since the categories and type of irrelevant structure between the train and test data are different. It requires conditional reasoning, since the queries are about a sub-set of the categories in the samples. Finally, it requires algorithmic reasoning to compute the output.

**Task description:** Let $n, k, m \in \mathbb{N}$ with $n \geq k$ and $n \geq m$. The input to the model is a list similar to that of Appendix C.6, augmented by adding $m$ more alphanumeric elements at random positions in the list that we call irrelevant structure. The irrelevant structure is generated by concatenating the string `'Cat'` with one of the characters in $\{$`'+'`, `'-'`, `'_'`, `'*'`$\}$ and a category identifier. The query part of the input, as well as the answer, remains consistent with Appendix C.6. As before, this augmented setting allows for one model to potentially process multiple query types and we present one such experiment later. As a real-world use case, this setting vaguely corresponds to the case where corrupted data is part of the input.

**OOD generalization:** For this task, we measure OOD generalization in three ways (simultaneously):

1. We test using category identifiers (i.e $i_1, i_2, \ldots, i_n$) that the models haven't encountered during training, and

2. The range of values $v_1, v_2, \ldots, v_n$ used for testing is larger than the range used for training.

3. The type of irrelevant structure used for testing is different than the type used for training. We detail this difference below.

**Experimental setting:** We fix $n = 8$, $k = 4$ and $m = 2$. Training and testing samples are generated as in Appendix C.6, with the difference being the presence of irrelevant structure in the inputs. Specifically, when sampling a training list we form irrelevant structure by concatenating the string `'Cat'` with either one of `'+'` or `'-'` (chosen at random) and a category identifier that is present in the actual input. For testing, irrelevant structure is formed similarly by concatenating the string `'Cat'` with either one of `'_'` or `'*'` (chosen at random) and a category identifier that is present in the actual (test) input. In both cases, irrelevant structure is injected into the actual list at random positions. We experiment on the same three variations as in Appendix C.6. Namely,

1. One where training and testing prompts consist exclusively of prompts where $\{$`type`$\}$ is `'min'`.

2. One where training and testing prompts consist exclusively of prompts where $\{$`type`$\}$ is `'max'`.

3. One where training and testing prompts consist exclusively of prompts where $\{$`type`$\}$ is either `'min'` or `'max'`. We refer to this experiment as "multitasking" since it allows a single trained model to process different query types. In fact, the choice of minimum and maximum as the two query types is, in some sense, an "extreme" case, given the "opposite" nature of minimum and maximum operations.

We experiment using three different types of architectures. We report the median MSE and median MAPE losses for the variations described above. The types of architectures tested are as follows:

1. The models for the first setting are exactly the same as the ones in Appendix C.6. In particular, all characters of the non-numeric parts of the input are tokenized and passed through an embedding layer, while the numeric ones are passed through a linear. The results for this setting are presented in Figure 24. In this setting, the performance of the positional Transformer is much better than that of the Transformer, which overfits the data, for all tasks. This version of the positional Transformer is the best-performing model among all other architectures and settings which are described below.

2. For the second setting, we tokenized everything in the input of both models. In particular, the numeric and non-numeric parts of the input are tokenized in the same way (character by character and digit by digit, respectively). For the standard Transformer, we closely follow the typical procedure of summing the embedded input and the vector of positional encodings. The results for this setting are presented in Figure 25. Tokenization of the numerical values resulted in poor performance for both architectures. This is also a good indication, that tokenization of the numbers is not a good approach for the algorithmic tasks which we consider in this paper. This is also confirmed by our experiment in the next setting, where we show that a fine-tuned GPT2 model also has considerably worse performance than positional Transformer with numerical representation of numbers.

3. Finally, for the third setting we take our experiment one step further and fine-tune GPT2-large from Hugging Face Radford et al. (2019) on this task. For all three variations (min, max, and multitasking), we fix the precision of the input numbers to 4 digits (2 digits for the whole part and 2 digits after the decimal point) and the precision of the output to 4 digits for min and max (2 digits for the whole part and 2 digits after the decimal point) and 5 digits for sum (3 digits for the whole part and 2 digits after the decimal point). This covers all possible numbers that can be sampled. We used byte pair encoding for tokenization and fine-tuned using 50.000 training samples. Model training ran until the validation MSE loss dropped below a task-specific threshold (0.05 for min and multitask and 1 for sum). Since this setting essentially reduces to next-token prediction, there were cases where the model's output did not correspond to a real number. As this did not occur frequently enough to be considered a problem, we report the median OOD losses (MSE and MAPE) ignoring samples for which no numeric value could be extracted from the model's output. The results are presented in Figure 26. We note that the OOD performance of the fine-tuned GPT2 model is worse than the positional Transformer with numerical representation of numbers in Figure 24, but it was better than that of the standard Transformer in other settings for the min and multitasking variations.

Overall, it is important to note that only the positional Transformer which maintained the numerical representation of numbers achieved good OOD performance. For all other architectures and settings (including the fine-tuned GPT2-large model), the OOD performance was considerably worse.

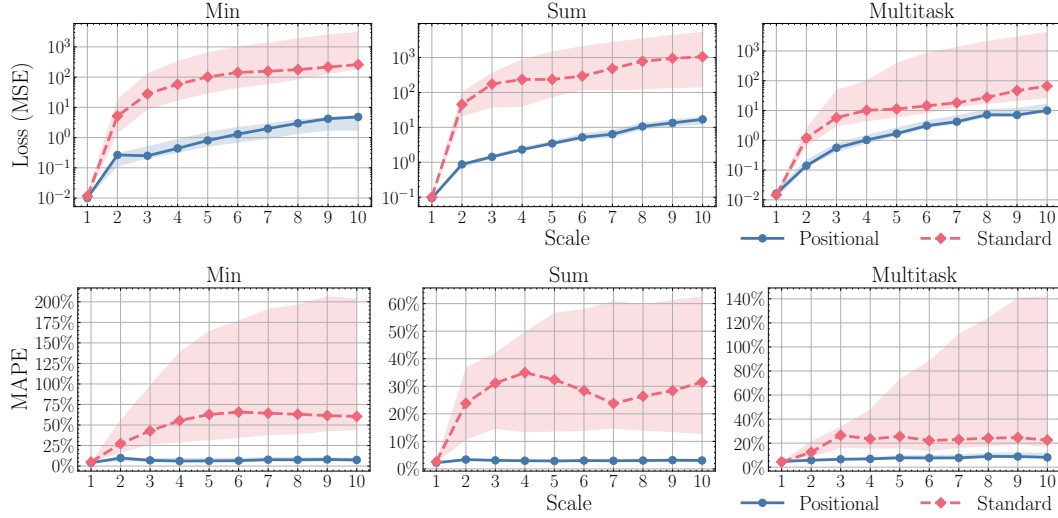

Figure 24: This experiment corresponds to the first setting above. OOD losses (measured as mean squared error, MSE, and mean absolute percentage error, MAPE) for standard Transformers (red) and positional Transformers (blue) across all three variations for the first setting. The x-axis represents the OOD scale factor. The solid line and shaded area denote the median and the region between the 10th and 90th percentiles over five trials, respectively.

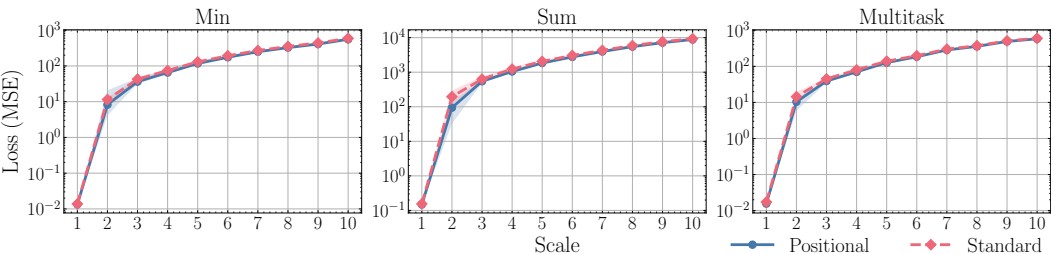

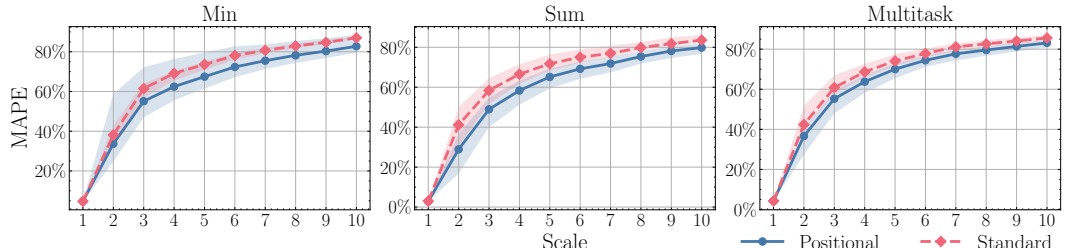

Figure 25: This experiment corresponds to the second setting above. OOD losses (measured as mean squared error, MSE, and mean absolute percentage error, MAPE) for standard Transformers (red) and positional Transformers (blue) across all three variations for the second setting. The x-axis represents the OOD scale factor. The solid line and shaded area denote the median and the region between the 10th and 90th percentiles over five trials, respectively.

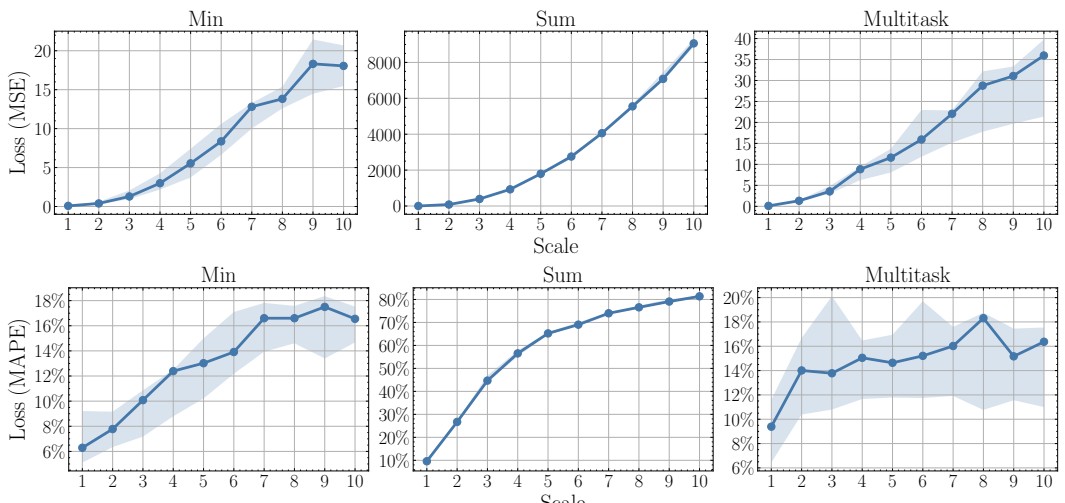

Figure 26: This experiment corresponds to the third setting above. All plots show OOD performance for the fine-tuned GPT2 model (measured as mean squared error, MSE, and mean absolute percentage error, MAPE). The x-axis represents the OOD scale factor. The solid line and shaded area denote the median and the region between the 10th and 90th percentiles over five trials, respectively.

## C.8 ACCURACY MEASURES

Given the regressive nature of the tasks considered in this work, the notion of model accuracy is not properly defined. However, we propose the following transformation strategies that assign binary labels ("correct"/"incorrect") to the models' outputs allowing us to measure their accuracy (with respect to these transformations):

- Rounding transformation: We evaluate the model on lists containing integers (while training is still done using real numbers) and round the model's output to the nearest integer (or nearest 0.5 for the median task). A prediction is considered "correct" if the rounded and ground truth lists are the same. The value generalization accuracies for all tasks in the main paper using this transformation strategy are presented in Figure 27.

- Closeness transformation: We evaluate the model on lists of real numbers, considering a prediction "correct" if each entry in the predicted list is within an absolute precision of 0.05 and a relative precision of 5% compared to the corresponding entry of the ground truth list. The value generalization accuracies for all tasks in the main paper using this transformation strategy are presented in Figure 28.

It is important to note that the above metrics are quite unforgiving, as even a single element in the predicted list failing to meet the corresponding criterion results in the entire list being classified as "incorrect". It is therefore expected that increasing the OOD scale factor causes the model's accuracy to decrease rapidly. Nevertheless, the positional Transformer significantly outperforms the standard Transformer in terms of accuracy in all five tasks.

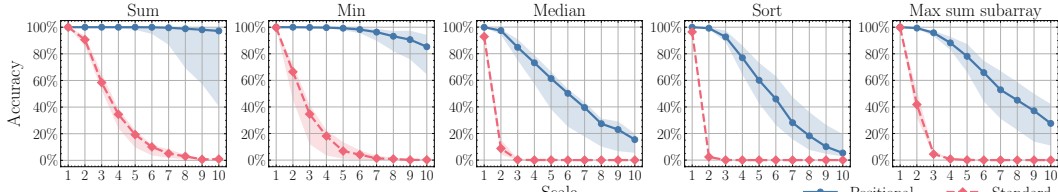

Figure 27: OOD accuracy when using "rounding transformation" across all five tasks for standard Transformers (red) and positional Transformers (blue) as a function of the scale factor. The solid line and shaded area denote the median and the region between the $10^{\text{th}}$ and $90^{\text{th}}$ percentiles over ten trials, respectively.

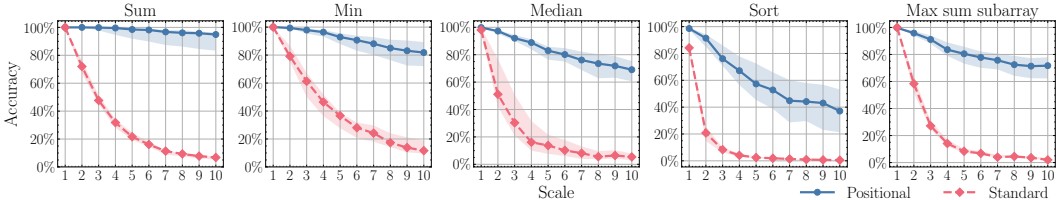

Figure 28: OOD accuracy when using "closeness transformation" across all five tasks for standard Transformers (red) and positional Transformers (blue) as a function of the scale factor. The solid line and shaded area denote the median and the region between the $10^{\text{th}}$ and $90^{\text{th}}$ percentiles over ten trials, respectively.

## D PROBABILITY OF GENERATING OOD TEST DATA IN OUR EMPIRICAL SETTING

Recall that we sample the training and test data in the following way. The training data consists of i.i.d samples whose values are drawn from the range $[-2, 2]$. To ensure diversity, for each training sample, we first select lower and upper bounds $\gamma_l$ and $\gamma_u$ uniformly in $[-2, 2]$, and then for each of the $n$ elements of the training sample, we select its value uniformly from the interval $[\gamma_l, \gamma_u]$. We employ a similar sampling strategy for testing but extend the value range to $[-2c, 2c]$, where $c > 1$ is the OOD scale factor. Additionally, during the test sampling process, we apply a rejection step to ensure that either $\gamma_l < -2$ or $\gamma_u > 2$, while maintaining $-2c \leq \gamma_l \leq \gamma_u \leq 2c$.

We will compute the probability that a randomly sampled test instance $x \in \mathbb{R}^n$ lies in the domain of the training distribution, i.e., we will compute $\mathbb{P}_{x \sim \mathcal{D}_{\text{test}}(\mathcal{X})}(x \in [-2, 2]^n)$. In particular, we will show that this probability is proportional to $1/nc^2$. Consequently, in our experiments, the majority of the test data lie outside of the domain of the training distribution.

Without loss of generality, let us assume that the training data are sampled within the interval $[-1, 1]$ and the test data are sampled within the interval $[-c, c]$, where $c$ is the OOD scale factor. Note that this does not affect the probability that we want to compute. In the test sampling process, when we sample two uniform numbers $\gamma_\ell$ and $\gamma_u$ from $[-c, c]$, exactly one of the following 3 disjoint events can happen.

- *Event A*. Exactly one of $\gamma_\ell$ and $\gamma_u$ lies in $[-1, 1]$. This happens with probability $2(\frac{c-1}{c})(\frac{1}{c})$.

- *Event B*. Neither $\gamma_\ell$ nor $\gamma_u$ is inside the interval $[-1, 1]$. This happens with probability $(\frac{c-1}{c})^2$.

- *Event C*. Both $\gamma_\ell$ and $\gamma_u$ are inside the interval $[-1, 1]$. This happens with probability $\frac{1}{c^2}$.

Our rejection step rejects the samples generated under Event C. Therefore, in our setting when we sample a pair of $\gamma_\ell$ and $\gamma_u$ in order to generate a single instance of the test list, we have that

$$\mathbb{P}(\text{Event A}|\text{Rejecting Event C}) = \frac{2(c-1)}{c^2-1}, \quad \mathbb{P}(\text{Event B}|\text{Rejecting Event C}) = \frac{(c-1)^2}{c^2-1}. \tag{7}$$

We analyze the probability of generating OOD test data under each event. First, let us suppose that Event A happens when we sample $\gamma_\ell$ and $\gamma_u$. This means that either $\gamma_\ell \in [-c, -1)$ or $\gamma_u \in (1, c]$ (but not both). More precisely, the following two sub-events partition Event A:

- *Event A.1.* $\gamma_\ell \in [-1, 1]$ and $\gamma_u \in (1, c]$. Given that Event A happens, this sub-event happens with probability 1/2.
- *Event A.2.* $\gamma_\ell \in [-c, -1)$ and $\gamma_u \in [-1, 1]$. Given that Event A happens, this sub-event happens with probability 1/2.

By symmetry of the probability distributions, the probability that we wish to compute remains the same under both of the above sub-events. Therefore, let us focus on Event A.1. Suppose that Event A.1 happens, i.e., $\gamma_\ell \in [-1, 1]$ and $\gamma_u \in (1, c]$. Conditioning on this event, we know that $\gamma_\ell$ is uniform on $[-1, 1]$ and $\gamma_u$ is uniform on $(1, c)$. The probability density functions for $\gamma_\ell$ and $\gamma_u$ are

$$f_{\gamma_\ell}(s) = \frac{1}{2} \cdot \mathbb{I}(-1 \le s \le 1), \quad f_{\gamma_u}(t) = \frac{1}{c-1} \cdot \mathbb{I}(1 < t \le c).$$

Let $X \in \mathbb{R}^n$ be a random vector whose $i$th coordinate $X_i$ is independently and uniformly sampled from the interval $[\gamma_\ell, \gamma_u]$, then the conditional probability density function for $X$ given $\gamma_\ell, \gamma_u$ is

$$f_X(x_1, x_2, \ldots, x_n | \gamma_\ell = s, \gamma_u = t) = \left(\frac{1}{t-s}\right)^n \cdot \mathbb{I}(s \le x_i \le t, \forall i).$$

Therefore, the joint density function is

$$f_{X, \gamma_\ell, \gamma_u}(x_1, \ldots, x_n, s, t) = \frac{1}{2} \frac{1}{(c-1)} \left(\frac{1}{t-s}\right)^n \cdot \mathbb{I}(-1 \le s \le 1, 1 < t \le c, s \le x_i \le t, \forall i).$$

It follows that

$$\begin{aligned}
p_{\text{A,in}} &:= \mathbb{P}_{X, \gamma_\ell, \gamma_u}\left(X_i \in [-1, 1], \forall i \in [n] \Big| \text{Event A}\right) \\
&= \int_{s \in [-1,1]} \int_{t \in (1,c]} \int_{x \in [s,1]^n} \frac{1}{2} \frac{1}{(c-1)} \left(\frac{1}{t-s}\right)^n dx\, dt\, ds \\
&= \frac{1}{2} \frac{1}{(c-1)} \int_{s \in [-1,1]} \int_{t \in (1,c]} \left(\frac{1-s}{t-s}\right)^n dt\, ds \\
&= \frac{1}{2} \frac{1}{(c-1)} \frac{1}{(n-1)} \int_{s \in [-1,1]} (1-s) \left(1 - \left(\frac{1-s}{c-s}\right)^{n-1}\right) ds \\
&= \frac{1}{2} \frac{1}{(c-1)} \frac{1}{(n-1)} \left[\int_{s \in [-1,0]} (1-s) \left(1 - \left(\frac{1-s}{c-s}\right)^{n-1}\right) ds \right.\\
&\qquad\qquad\qquad\qquad\qquad \left. + \int_{s \in [0,1]} (1-s) \left(1 - \left(\frac{1-s}{c-s}\right)^{n-1}\right) ds \right] \\
&\le \frac{1}{2} \frac{1}{(c-1)} \frac{1}{(n-1)} \left[\int_{s \in [-1,0]} (1-s) \left(1 - \left(\frac{1}{c}\right)^{n-1}\right) ds + \int_{s \in [0,1]} (1-s) ds \right] \\
&= \frac{3\left(1 - 1/c^{n-1}\right) + 1}{4(n-1)(c-1)}. \tag{8}
\end{aligned}$$

This is the probability that, under Event A, a randomly generated test sample lies within the domain of the training distribution. Again, recall that by scaling down the domain of the test distribution to $[-c, c]$ accordingly, we have assumed that the domain of the training distribution is $[-1, 1]^n$ without loss of generality.

Now suppose that Event B happens. In this case both $\gamma_\ell$ and $\gamma_u$ are uniformly distributed over $[-c, -1) \cup (1, c]$. The following two sub-events partition Event B:

- *Event B.1.* Either both $\gamma_\ell, \gamma_u > 1$ or both $\gamma_\ell, \gamma_u < -1$. Given that Event B happens, this sub-event happens with probability 1/2.

- *Event B.2.* $\gamma_\ell < -1$ and $\gamma_u > 1$. Given that Event B happens, this sub-event happens with probability $1/2$.

Let $X \in \mathbb{R}^n$ be a random vector whose $i$th coordinate $X_i$ is independently and uniformly sampled from the interval $[\gamma_\ell, \gamma_u]$. Note that under Event B.1, one always has that $X \notin [-1, 1]^n$, i.e.,

$$p_{\text{B.1,in}} := \mathbb{P}_{X, \gamma_\ell, \gamma_u} \left( X_i \in [-1, 1], \forall i \in [n] \Big| \text{Event B.1} \right) = 0. \tag{9}$$

Therefore let us consider Event B.2. Conditioning on this event, we know that $\gamma_\ell$ is uniform on $[-c, -1)$ and $\gamma_u$ is uniform on $(1, c]$. The joint density function (conditional on Event B.2) for $X, \gamma_\ell, \gamma_u$ is

$$f_{X, \gamma_\ell, \gamma_u}(x_1, \ldots, x_n, s, t) = \frac{1}{(c-1)^2} \left( \frac{1}{t-s} \right)^n \cdot \mathbb{I}(-c \leq s \leq 1, 1 < t \leq c, s \leq x_i \leq t, \forall i).$$

Therefore, we have that

$$p_{\text{B.2,in}} := \mathbb{P}_{X, \gamma_\ell, \gamma_u} \left( X_i \in [-1, 1], \forall i \in [n] \Big| \text{Event B.2} \right)$$

$$= \int_{s \in [-c, 1)} \int_{t \in (1, c]} \int_{x \in [s, 1]^n} \frac{1}{(c-1)^2} \left( \frac{1}{t-s} \right)^n dx \, dt \, ds$$

$$= \frac{1}{(c-1)^2} \int_{s \in [-c, 1)} \int_{t \in (1, c]} 2^n \left( \frac{1}{t-s} \right)^n dt \, ds$$

$$= \frac{2^n}{(c-1)^2 (n-1)} \int_{s \in [-c, 1)} \left( \frac{1}{(1-s)^{n-1}} - \frac{1}{(c-s)^{n-1}} \right) ds$$

$$= \begin{cases} \dfrac{4 - 8(\frac{2}{1+c})^{n-2} + 4(\frac{1}{c})^{n-2}}{(c-1)^2 (n-1)(n-2)}, & \text{if } n \geq 3, \\[2ex] \dfrac{2 \log(c+1) - \log c - 2 \log 2}{(c-1)^2}, & \text{if } n = 2. \end{cases} \tag{10}$$

Combining Equation (7), Equation (8), Equation (9), Equation (10), we get that, if $n \geq 3$,

$$p_{\text{in}} := \mathbb{P}_{x \sim \mathcal{D}_{\text{test}}(\mathcal{X})}(x \in [-1, 1]^n) \leq \frac{3(1 - 1/c^{n-1}) + 1}{2(c^2 - 1)(n-1)} + \frac{2 - 4(\frac{2}{1+c})^{n-2} + 2(\frac{1}{c})^{n-2}}{(n-1)(n-2)(c^2 - 1)} \tag{11}$$

$$\leq \frac{2}{(c^2 - 1)(n-1)} + \frac{2}{(n-1)(n-2)(c^2 - 1)}$$

$$= O\left( \frac{1}{nc^2} \right)$$

and if $n = 2$,

$$p_{\text{in}} \leq \frac{3(1 - 1/c) + 1 + 2 \log(c+1) - \log c - 2 \log 2}{2(c^2 - 1)} \leq \frac{3(1 - 1/c) + 9/8}{2(c^2 - 1)}. \tag{12}$$

In the above, $p_{\text{in}}$ is the probability that a randomly sampled test list $x \in \mathbb{R}^n$ has all its elements lie within $[-1, 1]$, that is, the probability that $x$ lies within the domain of the training distribution. This probability is at most $O(1/nc^2)$. Suppose that we generate $N$ test instances, then a straightforward application of the multiplicative Chernoff bound yields that with probability at least $N^{-C}$ for some constant $C > 0$, at most $O(\frac{N}{nc^2})$ samples will lie in the domain of the training distribution.

For $n \in \{2, 4, 8, 16, 32\}$ and $c \in \{1, 2, \ldots, 10\}$ which we consider in our experiments, Figure 29 shows a contour plot of the probability upper bound Equation (11) and Equation (12). This probability is sufficiently small such that the majority of test instances in our test data does not belong to the domain of the training distribution.

For small $n$ and $c$, to determine the fraction of sampled test instances that will be within the domain of the training distribution, it is more informative to directly invoke the additive Chernoff bound with $p_{\text{in}}$. Let $N$ denote the total number of test instances that we sample, and further let $N_{\text{in}}$ denote the number of sampled instances that lie in the domain of the training distribution. Then by the additive Chernoff bound we have that

$$\mathbb{P}(N_{\text{in}} \geq N(p_{\text{in}} + \epsilon)) \leq \exp(-2N\epsilon^2). \tag{13}$$

For example, suppose that we sample $N = 1000$ test instances from the test distribution. Suppose that we generate the test data using list length $n = 2$ and OOD scale factor $c = 2$. Then in this case $p_{\text{in}} \leq 0.4375$. Take $\epsilon = 0.0625$. Then (13) says that with probability at least 0.9995, at least $N/2$ samples do not lie in the domain of the training distribution. For another example with slightly larger $n$ and $c$, suppose that we generate the test data using list length $n = 8$ and OOD scale factor $c = 10$. Then $p_{\text{in}} \leq 0.0034$. Take $\epsilon = 0.0466$. Then (13) says that with probability at least 0.98, more than 95% of test instances do not lie in the domain of the train distribution.

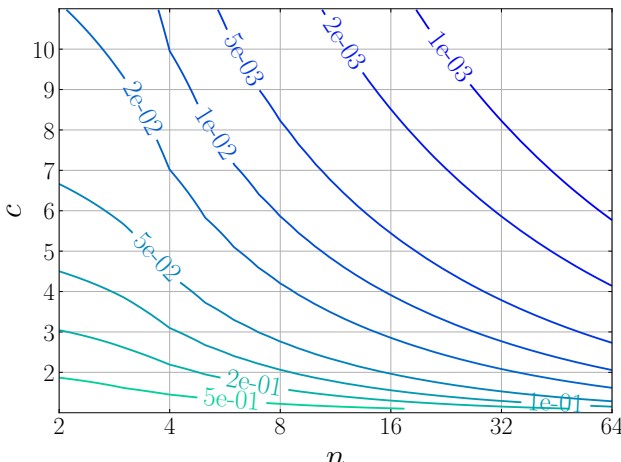

Figure 29: Contour plot of $\mathbb{P}_{x \sim \mathcal{D}_{\text{test}}(\mathcal{X})}(x \in \text{supp}(\mathcal{D}_{\text{train}}(\mathcal{X})))$, i.e., the probability (upper bound in Equation (11) and Equation (12)) that a randomly sampled test instance $x \in \mathbb{R}^n$ lies in the domain of the training distribution.

# E    POTENTIAL REASONS FOR FAILURE IN SELF-ATTENTION

In this section, we discuss potential reasons for the shortcomings of standard Transformers in algorithmic tasks. While it is more straightforward to elicit reasons for the success of positional attention – motivated by the *algorithmic alignment* (Xu et al., 2020) between positional Transformers and parallel computational models – it is considerably more challenging to pinpoint the causes of failure in standard Transformers.

Firstly, Transformers can simulate parallel algorithms, as demonstrated by Sanford et al. (2024). Intuitively, a single layer of self-attention should be more powerful than positional attention, as it leverages attention beyond positional encodings and allows for a more flexible structure in response to input variations. However, as discussed in Section 5, executing parallel algorithms does not require using anything beyond positional information in attention.

Assuming that standard Transformers should adopt positional information to effectively execute parallel algorithms, the operations required by standard Transformers become increasingly difficult than positional Transformers for two main reasons:

1. Self-attention layers must learn to ignore input values and exploit positional information.

2. Transformer layers must preserve positional encodings for subsequent layers.

Namely, these desirable properties of positional Transformers present two significant challenges for standard Transformers. The first challenge arises naturally from the differences between standard and positional attention mechanisms. The second challenge highlights the compositional structure of attention layers, which can be detrimental during training. Specifically, the operations performed by each attention and MLP layer can degrade the inputs of subsequent layers.

This issue is further emphasized in Remark 2, where we state that while positional Transformers can represent any softmax pattern at any layer, standard Transformers may fail to do so due to potential degradation of the attention inputs. Although residual connections can mitigate this issue by preserving input information, they must ensure that no overlaps hinder the use of positional encodings in subsequent layers. Moreover, this problem compounds across layers, making training more difficult as errors in earlier layers adversely affect subsequent computations.

Nevertheless, these remain speculative reasons for the observed failure of standard Transformers. Determining the exact causes and the difficulty of achieving the two aforementioned goals through training requires a thorough analysis of the training dynamics, which is inherently challenging. Future in-depth work within the mechanistic interpretability framework (Nanda et al., 2023) can potentially shed light on these issues by inspecting network parameters at convergence, thereby uncovering the underlying reasons for the failure of standard Transformers.

Along this direction, we present some empirical evidence, in Figure 30 and Figure 31, that self-attention layers in a trained Transformer model can be highly sensitive to input values. In particular, the attention weights

change dramatically when the input values of the test data do not necessarily lie in the domain of the training data. This suggests that self-attention potentially overfits the training data and, therefore, offers a plausible explanation for why the standard Transformers exhibit such poor value generalization in our experiments.

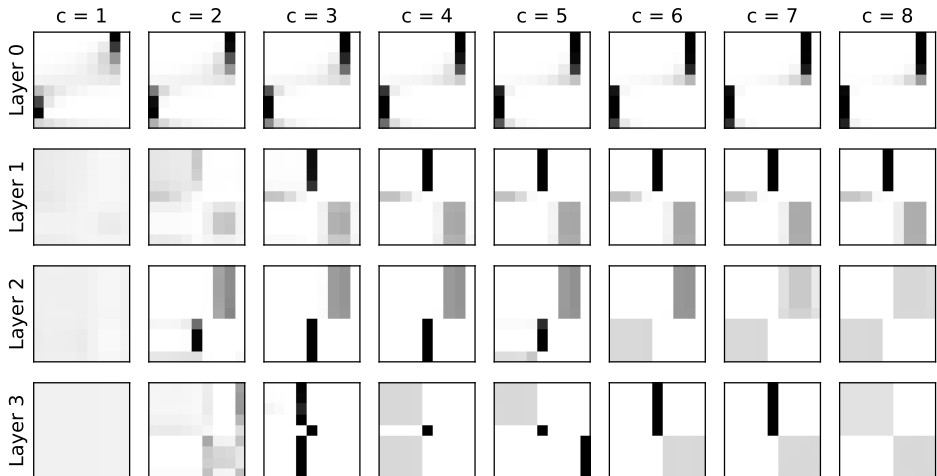

Figure 30: Visualization of learned attention weights in the standard Transformer model trained to solve the sorting task in our experiments. The input list to the model is $cX$ where $X = [1.75, 1.25, 0.75, 0.25, -0.25, -0.75, -1.25, -1.75]$ and $c = 1, 2, \ldots, 8$ is a scaling factor. The model is trained on data whose input values range from -2 to 2. Therefore $c = 1$ gives in-distribution data and larger $c$ yields OOD data. For each layer in the architecture, we plot 1 of the 2 attention heads for illustration purposes. The trend for the other head is similar. Observe that the attention weights change dramatically as we increase the scaling factor of input values, with deeper layers suffering from more radical changes in the attention pattern under even a small change in the scale (e.g. going from $c = 1$ to $c = 2$). This behavior potentially explains why the standard Transformer model performs poorly on OOD test data in our experiments.

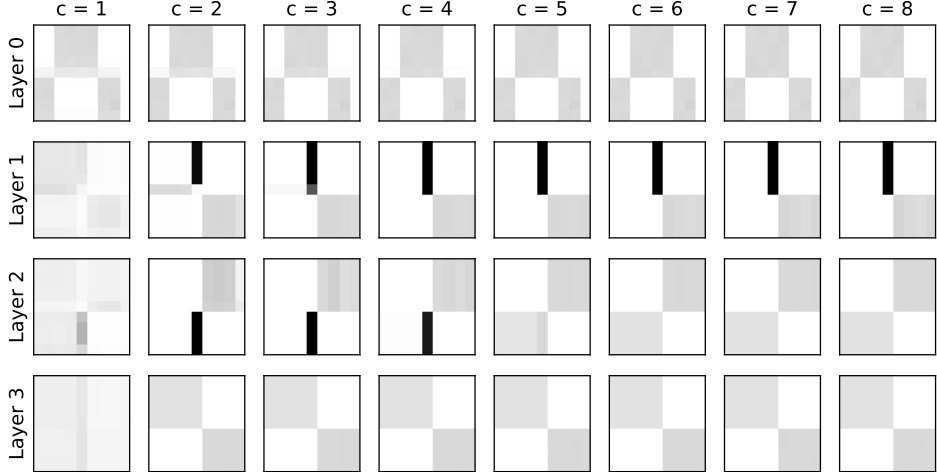

Figure 31: Another visualization of learned attention weights in the standard Transformer model. We use the same setting as described in Figure 30, except that the input list is $cX$ where $X = [2, 2, -2, -2, -2, -2, 2, 2]$. Again, we observe that the attention weights are highly sensitive to the scaling factor $c$, especially those at deeper layers.

# F   Informal Discussion on Out-of-distribution Generalization Bounds and future work

The topic of OOD generalization has been studied extensively from a theoretical perspective. Researchers often focus on bounding the risk on the test distribution using a mixture of terms that go to zero as the number of samples increases, plus a divergence term that does not necessarily go to zero depending on the distributions and the hypothesis class. For an extensive survey, we refer the reader to Redko et al. (2022). Although, in general, such bounds offer valuable intuition, they might not be tight for our particular setting. In particular, we examine a popular type of bound found in Mansour et al. (2009) which can be large even if the difference in the support of the train and test distributions is the smallest it can be. Note that there are more types of bounds in Redko et al. (2022) than the one found in Mansour et al. (2009). Although we have not conducted an in-depth analysis of all cases, we note that all of them depend in a worst-case on the hypothesis class. We believe that to improve upon such generic bounds, one must consider the dynamics of the training procedure for deep positional attention architectures. We find this topic extremely interesting, but we leave it for future work, as it is highly non-trivial.

In what follows, we use a popular example of one of these bounds Mansour et al. (2009) and illustrate why it is not tight for a simple task of interest. Briefly, the main issue with this particular OOD bound is that it depends in a worst-case manner on the hypothesis class. We demonstrate this issue using the task of computing the minimum over an array of length $n$. We assume that $n$ is even. For simplicity, we do not work with the cumulative version of the minimum problem, as we did in the main paper. Therefore, the ground truth is simply the minimum of the input array. $\mathcal{D}_{\text{train}}$ is the train distribution over arrays of length $n$, where each component of the array is sampled independently and uniformly at random from integers in the range 0 to $L_{\mathcal{D}_{\text{train}}}$, where $L_{\mathcal{D}_{\text{train}}}$ is a constant. $\mathcal{D}_{\text{test}}$ is the test distribution over arrays of length $n$, where each component of the array is sampled independently and uniformly at random from integers in the range 0 to $L_{\mathcal{D}_{\text{train}}} + z$, where $z \geq 0$ is a constant integer. We use an equal number of samples from the train and test distributions, denoted by $m$. The loss function is $\ell(h(x), y) = |h(x) - y|$, where $h$ is a hypothesis, $x$ is an input array of length $n$, and $y = \min(x)$, which is the minimum function over the array $x$.

The hypothesis class $H$ is the architecture in equation 1 with $\log_2 n$ layers, 2 heads per layer, and $W_O$ and $W_V$ as the identity matrices for all layers. For positional vectors in general position with dimension $n$, Lemma 1 implies that there exist key and query matrices of size $n \times n$ that can represent any attention pattern at each layer. The MLP at each layer consists of 2 layers with a hidden dimension equal to 4. We use the ReLU activation function in all MLPs. This allows the MLP to represent the minimum and maximum functions on two input values exactly. This is because the minimum and maximum functions can be written using ReLUs and linear operations:

$$\min(x_1, x_2) = \frac{1}{2}(\text{ReLU}(x_1 + x_2) - \text{ReLU}(-x_1 - x_2) - \text{ReLU}(x_1 - x_2) - \text{ReLU}(x_2 - x_1))$$

and

$$\max(x_1, x_2) = \frac{1}{2}(\text{ReLU}(x_1 + x_2) - \text{ReLU}(-x_1 - x_2) + \text{ReLU}(x_1 - x_2) + \text{ReLU}(x_2 - x_1)).$$

Note that the MLP's ability to represent the minimum function for two inputs exactly is also the reason it can represent the maximum function. In other words, the exact representation of the minimum function comes with the consequence that the MLP can also represent the maximum function for two inputs. This observation is crucial later when we show that an existing popular bound from Mansour et al. (2009) is not tight for this particular task.

Furthermore, we assume that $|h(x)|$ is constant, which further implies that the magnitude of the loss is bounded above by a constant. Observe that this hypothesis class can represent a binary tree reduction algorithm for the minimum and maximum functions. This is possible because positional attention can represent any attention pattern, and the MLPs can represent the minimum and maximum functions for two inputs exactly. Specifically, the first layer of the positional attention architecture can represent the connections between layers 0 (leaf nodes) and 1 in the binary tree computational graph, the second layer can represent the connections between layers 1 and 2, and so on, up to the $\log_2 n$-th layer. The MLPs are used locally at each node of the binary tree to compute the minimum between two input values. Therefore, the minimum and maximum functions over an array of $n$ elements are in the hypothesis class.

Let us now discuss one of the most popular OOD generalization bound results for regression. We will use the third case of Theorem 8 in Mansour et al. (2009), which states the following.

**Theorem 2** (Theorem 8 in Mansour et al. (2009), repurposed for the minimum function task). *Assume that the loss function $\ell$ is symmetric, it obeys the triangle inequality, and it is bounded above by a constant. If the minimum function is in the hypothesis class $H$, then, for any hypothesis $h \in H$, the following holds:*

$$R_{\mathcal{D}_{test}}(h) \leq R_{\mathcal{D}_{train}}(h) + disc_\ell(\mathcal{D}_{test}, \mathcal{D}_{train})$$

*where*

$$disc_\ell(\mathcal{D}_{test}, \mathcal{D}_{train}) = \max_{h,h' \in H} \left| R_{\mathcal{D}_{test}}(h, h') - R_{\mathcal{D}_{train}}(h, h') \right|$$

*and*

$$R_{\mathcal{D}_{test}}(h, h') = \mathbb{E}_{(x,y) \sim \mathcal{D}_{test}}[\ell(h(x), h'(x))].$$

*and $R_{\mathcal{D}_{train}}(h, h')$ is defined similarly.*

The above theorem states that the difference in risk between the test and train distributions is bounded only by the discrepancy term between the two distributions. It is important to note that the discrepancy term depends on the hypothesis class and measures the worst-case difference in the train and test risks within that class. The fact that the discrepancy considers the worst-case difference is why this bound is not tight for our task of computing the minimum function.

Let us now focus on the discrepancy term. Corollary 7 in Mansour et al. (2009) states that

$$\mathrm{disc}_\ell(\mathcal{D}_{\text{test}}, \mathcal{D}_{\text{train}}) \leq \mathrm{disc}_\ell(\hat{\mathcal{D}}_{\text{test}}, \hat{\mathcal{D}}_{\text{train}}) + 4(\hat{\mathcal{R}}_{\mathcal{S}_{\text{test}}}(H) + \hat{\mathcal{R}}_{\mathcal{S}_{\text{train}}}(H)) + \mathcal{O}\left(\sqrt{\frac{1}{m}}\right),$$

where $\hat{\mathcal{D}}_{\text{test}}$ and $\hat{\mathcal{D}}_{\text{train}}$ are the empirical versions of the test and train distributions, repsectively. We will use the common assumption that these empirical distributions are uniform over the samples. $\mathcal{S}_{\text{test}}$ and $\mathcal{S}_{\text{train}}$ are the sample sets for the train and test distributions, respectively. Moreover, $\hat{\mathcal{R}}_{\mathcal{S}_{\text{test}}}(H)$ and $\hat{\mathcal{R}}_{\mathcal{S}_{\text{train}}}(H)$ are the empirical Rademacher complexities for the train and test distributions, respectively. It is well-known that the part of the bound corresponding to the empirical Rademacher complexities goes to zero as the number of samples increases. The same holds for the square-root term in the bound. Therefore, the only term left to understand is the discrepancy between the empirical distributions. Let us try to understand how this term behaves. Its definition is:

$$\mathrm{disc}_\ell(\hat{\mathcal{D}}_{\text{test}}, \hat{\mathcal{D}}_{\text{train}}) = \max_{h,h' \in H} \left| \frac{1}{m} \sum_{x \in \mathcal{S}_{\text{test}}} \ell(h(x), h'(x)) - \frac{1}{m} \sum_{x \in \mathcal{S}_{\text{train}}} \ell(h(x), h'(x)) \right|.$$

A lower bound of $\mathrm{disc}_\ell(\hat{\mathcal{D}}_{\text{test}}, \hat{\mathcal{D}}_{\text{train}})$ is given by setting $h$ to be the minimum function and $h'$ to the maximum function:

$$\mathrm{disc}_\ell(\hat{\mathcal{D}}_{\text{test}}, \hat{\mathcal{D}}_{\text{train}}) \geq \left| \frac{1}{m} \sum_{x \in \mathcal{S}_{\text{test}}} (\max(x) - \min(x)) - \frac{1}{m} \sum_{x \in \mathcal{S}_{\text{train}}} (\max(x) - \min(x)) \right|$$

We claim that for polynomial number of samples $m$, e.g., $m = n^c$, where $c$ is a positive integer, there exists $n_0$, such that for all $n \geq n_0$ we have that $\min(x) = 0$ and $\max(x) = L_{\mathcal{D}_{\text{train}}}$ for all $x \in \mathcal{S}_{\text{train}}$, and $\min(x) = 0$ and $\max(x) = L_{\mathcal{D}_{\text{train}}} + z$ for all $x \in \mathcal{S}_{\text{test}}$ with probability at least $0.8$. The proof of this claim is trivial and we provide it below. For now, let us discuss the implications of this claim. We have that $\mathrm{disc}_\ell(\hat{\mathcal{D}}_{\text{test}}, \hat{\mathcal{D}}_{\text{train}}) \geq L_{\mathcal{D}_{\text{train}}} + z - L_{\mathcal{D}_{\text{train}}} = z$ with probability at least $0.8$. Therefore, even if $z$ is the smallest it can be such that there is a difference between the train and test distributions in this particular setting, i.e., $z = 1$, then the empirical discrepancy is going to be at least 1. This means that the upper bound of Theorem 2 is at least one, and it is not going to zero as the number of samples increases. This is because the discrepancy definition considers the worst-case scenario without considering the training procedure. In practice, the training procedure may help to discover a hypothesis that is close to the minimum function since the minimum function is part of the hypothesis class. If the hypothesis discovered by the training procedure is close enough to the minimum function, the OOD generalization error may be much smaller than 1. Therefore, depending on the learning task and the hypothesis class, the bound in Theorem 2 can be loose.

Consider the following example, $L_{\mathcal{D}_{\text{train}}} = 3$ and $z = 1$. Therefore, the largest value in the test distribution is 4. For $m$ and $n$ as noted above, the bound implies that the loss might be up to 1 for any hypothesis $h$. This further implies that for any hypothesis $h$ the relative error might be up to $25\%$ with probability at least $0.8$, despite the fact that the hypothesis class includes the true function and the training procedure could converge to a good approximation of it.

Let us prove the above probability claim. For $x \in \mathcal{S}_{\text{train}}$ we have

$$\mathbb{P}(x_i \neq 0 \text{ for all } i \in [n]) = \prod_{j=1}^{n} \mathbb{P}(x_j \neq 0)$$

$$= \prod_{j=1}^{n} \mathbb{P}(x_j \in \{1, 2, \ldots, L_{\mathcal{D}_{\text{train}}}\})$$

$$= \prod_{j=1}^{n} \frac{L_{\mathcal{D}_{\text{train}}}}{L_{\mathcal{D}_{\text{train}}} + 1}$$

$$= \left( \frac{L_{\mathcal{D}_{\text{train}}}}{L_{\mathcal{D}_{\text{train}}} + 1} \right)^{n}.$$

Similarly, we have that

$$\mathbb{P}(x_i \neq L_{\mathcal{D}_{\text{train}}} \text{ for all } i \in [n]) = \left( \frac{L_{\mathcal{D}_{\text{train}}}}{L_{\mathcal{D}_{\text{train}}} + 1} \right)^{n}.$$

Furthermore, we have that

$$\mathbb{P}(\exists\, i, j \in [n] : x_i = 0 \text{ and } x_j = L_{\mathcal{D}_{\text{train}}}) = 1 - \mathbb{P}(x_i \neq 0 \,\forall\, i \in [n] \text{ or } x_i \neq L_{\mathcal{D}_{\text{train}}} \,\forall\, i \in [n])$$

$$\geq 1 - \mathbb{P}(x_i \neq 0 \,\forall\, i \in [n]) - \mathbb{P}(x_i \neq L_{\mathcal{D}_{\text{train}}} \,\forall\, i \in [n])$$

$$= 1 - 2 \left( \frac{L_{\mathcal{D}_{\text{train}}}}{L_{\mathcal{D}_{\text{train}}} + 1} \right)^{n}.$$

Therefore, we conclude that

$$\mathbb{P}(\text{all samples } x \in \mathcal{S}_{\text{train}} \text{ have at least one } 0 \text{ or } L_{\mathcal{D}_{\text{train}}}) \geq \left( 1 - 2 \left( \frac{L_{\mathcal{D}_{\text{train}}}}{L_{\mathcal{D}_{\text{train}}} + 1} \right)^{n} \right)^{m}.$$

and, similarly, we conclude that

$$\mathbb{P}(\text{all samples } x \in \mathcal{S}_{\text{test}} \text{ have at least one } 0 \text{ or } L_{\mathcal{D}_{\text{train}}} + z) \geq \left( 1 - 2 \left( \frac{L_{\mathcal{D}_{\text{train}}} + z}{L_{\mathcal{D}_{\text{train}}} + z + 1} \right)^{n} \right)^{m}.$$

For a polynomial number of samples $m$, e.g., $m = n^c$, where $c$ is a positive integer, there exists some $n_0 \in \mathbb{N}$, such that for all $n \geq n_0$ we have that the latter two probabilities are at least $0.9$ (since for $m = n^c$ both lower bounds tend to $1$ as $n$ tends to infinity). Therefore, our claim about the minimum and maximum over the sampled arrays holds with probability at least $0.81$.

# G ABLATION STUDY ON VARIOUS ARCHITECTURAL DESIGN CHOICES

We carried out additional experiments to explore variations in the input data format and the placement of positional encodings. This leads to the following 7 architectural choices (including 2 from the main paper and 5 additional variations):

1. Standard Transformers: Input numbers and positional encodings are fed to MLPs, value, query, and key matrices.

2. Standard Transformers with positional input (but no positional encodings): Input numbers are placed in one-hot positions, that is, the input is $\text{Diag}(X)$ where $X$ is the list of input numbers we give to other architectures. No additional positional encodings are used.

3. Positional Transformers: Input numbers are fed to the MLPs and value matrix; positional encodings are fed to query and key matrices.

4. Misaligned Positional Transformers: Input numbers are fed to the MLPs and value matrix; positional encodings are fed to the MLPs, value, query, and key matrices. That is, compared with Positional Transformers, we add positional encodings to the input.

5. Input-regularized Standard Transformers: Input numbers are fed to the MLPs, value, query, and key matrices; positional encodings are only used in query and key matrices.

6. No Positional Encodings: input numbers are fed to the MLPs, value, query, and key matrices. No positional encodings are used.

7. Using RoPE Only: Input numbers are fed to MLPs, value, query, and key matrices, removing absolute positional encodings, and using only RoPE in standard transformers.

We test the architectures for the cumulative sum task and the sorting task as representative tasks, for fixed input length $n = 8$. We keep the train/valid/test setup the same as before. The results are shown in Figure 32 and Figure 33. We note that our proposed Positional Transformer architecture has all of the following 3 important factors that enabled OOD value generalization: (1) use positional encodings, (2) do not use positional encodings in the value matrix of the attention, (3) use only fixed positional encodings in the key and query matrices. This design principle aligns with algorithms that are typically used to solve algorithmic tasks.

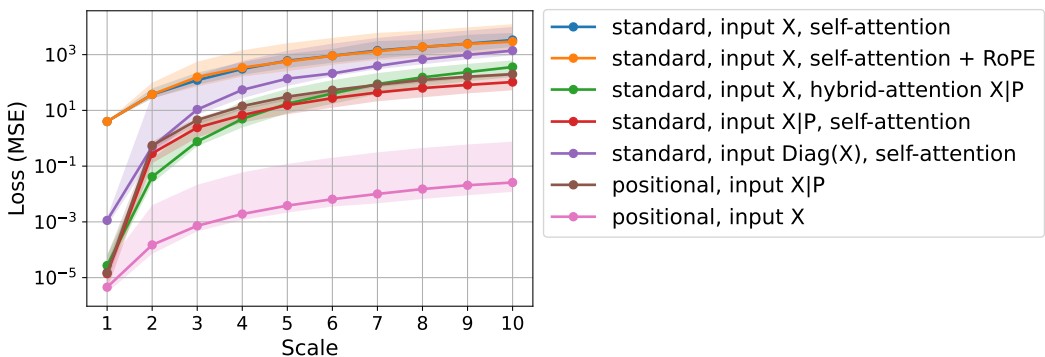

Figure 32: OOD loss for the cumulative sum task under various architectural choices. We fix input length $n = 8$. The x-axis represents the OOD scale factor. The solid line and shaded area denote the median and the region between the $10^{th}$ and $90^{th}$ percentiles for 10 trials, respectively.

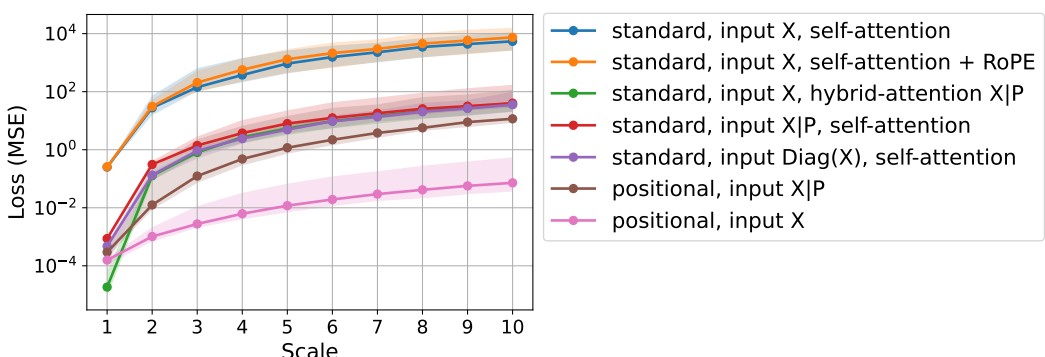

Figure 33: OOD loss for the sorting task under various architectural choices. We fix input length $n = 8$. The x-axis represents the OOD scale factor. The solid line and shaded area denote the median and the region between the $10^{th}$ and $90^{th}$ percentiles for 10 trials, respectively.

## H  EMPIRICAL RESULTS USING GRAPH NEURAL NETWORKS (GNNS)

Graph Neural Networks are a popular choice for solving algorithmic tasks on graphs. We tested the performance of Graph Convolutional Network (GCN) and Graph Attention Network (GAT) in solving the cumulative sum and sorting tasks. Since the tasks tested have no underlying native graph, we tested these models on complete and star graphs. Notably, the original GAT architecture on a complete graph is similar to a standard transformer but differs in that the value, key, and query weights are shared in GAT. In Table 1 we present results reporting the median MSE loss. As the results indicate, neither GCN nor GAT works very well even for in-distribution data (OOD scaling factor = 1), let alone achieving OOD generalization. We believe this is because these tasks are inherently unsuitable for standard message-passing architectures.

Table 1: OOD loss of Graph Convolutional Network (GCN) and Graph Attention Network (GAT)

| Task | Graph | Architecture | OOD scale factor | | | | | |
| | | | 1 | 2 | 3 | 4 | 5 | 10 |
|---|---|---|---|---|---|---|---|---|
| Cumulative Sum | Complete | GCN | 3.9e+0 | 2.1e+1 | 4.3e+1 | 7.4e+1 | 1.2e+2 | 4.8e+2 |
| | | GAT | 3.7e+0 | 2.0e+1 | 4.3e+1 | 7.5e+1 | 1.2e+2 | 5.4e+2 |
| | Star | GCN | 4.0e+0 | 2.0e+1 | 4.2e+1 | 7.4e+1 | 1.2e+2 | 4.6e+2 |
| | | GAT | 3.8e+0 | 2.0e+1 | 4.1e+1 | 7.5e+1 | 1.2e+2 | 5.0e+2 |
| Sorting | Complete | GCN | 1.9e-1 | 9.8e-1 | 2.2e+0 | 3.9e+0 | 6.1e+0 | 2.7e+1 |
| | | GAT | 1.9e-1 | 9.6e-1 | 2.1e+0 | 3.7e+0 | 5.6e+0 | 2.4e+1 |
| | Star | GCN | 1.9e-1 | 1.0e+0 | 2.1e+0 | 3.7e+0 | 6.1e+0 | 2.6e+1 |
| | | GAT | 1.9e-1 | 9.9e-1 | 2.0e+0 | 3.5e+0 | 5.6e+0 | 2.3e+1 |

