# OpenReview forum: "Positional Attention: Out-of-Distribution Generalization and Expressivity for Neural Algorithmic Reasoning"
_ICLR.cc/2025/Conference — Submitted to ICLR 2025_

### Official Review · Reviewer_k9Nh · 2024-10-24

**Soundness:** 2
**Presentation:** 4
**Contribution:** 2
**Rating:** 5
**Confidence:** 4

**Summary:**

The paper focuses on the task of improving the performance of Transformer models on arithmetic tasks. In particular, the authors focus on “value generalisation”, that is “where the test distribution has the same input sequence length as the training distribution, but the value ranges in the training and test distributions do not necessarily overlap”. Stemming from the observation that the steps executed by many basic array algorithms (sum, sort, min, max) are independent of the underlying values, the authors propose to fix the attention matrix of each layer, and make it independent of the provided input.

**Strengths:**

The paper impresses with the following points:

S1. The paper is very-well written, with the exposition, background material, definitions, methods and results being very clearly presented and easy to follow, with extensive details.

S2. The paper offers a thorough theoretical explanation of how the proposed positional transformers can simulate any algorithm falling under the Parallel Computation with Oracle Communication (PCOC) model.

**Weaknesses:**

W1. Section 5 of the paper focuses offers a theoretical explanation of why the proposed positional transformers can simulate any algorithm falling under the Parallel Computation with Oracle Communication (PCOC) model. While it is indeed quite interesting that there is a rich class of computation that can be approximated with fixed attention, it is not clear how this theory ties in with the rest of the paper. Most critically, how does this theory explain the observation that fixed attention result in __better__ empirical performance? Doesn’t the referenced work of Sanford et al. imply that transformers with dynamic attention should also be able to approximate algorithms from the same class?

W2. The paper’s main thesis is that _“positional attention enhances empirical OOD performance”_, however it does not propose a hypothesis explaining as to why that might be the case. I find this conclusion of the paper to be surprising and difficult to believe without further details. The authors construct a _strictly less expressive_ architecture that has _better_ performance. While that can be due to appropriate inductive biases, the classic transformer should be able to learn these tasks too. If the positional embeddings of the classic transformer are learnable and concatenated to the inputs, then isn’t it possible to show that for every positional Transformer, one can construct a classic transformer that is functionally equivalent to it? Furthermore, as the authors also discuss in their related work section, we know that the classic transformer is a universal approximator for continuous and discrete functions, as well as Turing-complete. Hence, the classic transformer should also  be able to solve the tasks considered in the paper. I am also concerned with the authors mentioning that

> In particular, the weights can be very sensitive to the scale of input values. We observe a dramatic change in attention weights as soon as we give the model the same input but scaled so that the values lie outside the domain of the training data.

Perhaps I am missing something, but shouldn’t scaling the input by $\lambda$ scale all elements of the pre-softmax attention matrix by $\lambda^2$. And as scaling the pre-softmax activations is equivalent to adjusting the temperature of the softmax, isn't this difference in attention predictable and expected?

All that makes me think that the reported results might be caused by technical details in the implementation of the experiments, rather than due to positional attention being fundamentally better for some reason.

W3. Improving the ability of a Transformer to solve algorithms in isolation seems unnecessary: the example tasks used in the paper can be solved exactly with just a couple of lines of code. There would be value if the model uses these tasks as a subroutine of more complex tasks, e.g., given a natural text description of purchases, group them by type and compute commulative sums for each group. However, the paper does not study whether the proposed positional encodings work for more general data distributions, e.g., NLP tasks, different formatting of the same tasks, non-PCOC tasks, or mixtures of tasks. My intuition is that the ridgidity of the attention mechanism that improves the performance on the tasks considered in the paper, would be detrimental to the performance on such tasks that require flexibility, nuance, and robustness. And as LLMs are useful precisely for these less-structured tasks —for the structured ones, one is better off writing a Python script— I feel that the paper has failed to propose a modification of the transformer architecture that would result in improved performance in any realistic case. As such, the paper would likely be of limited value to the community.

Overall, despite the clear writing and good presentation, the paper has major conceptual and methodological flaws. The restricted architecture that is proposed could work well for the small task set the paper considers, but is unlikely to work well in practical settings where flexibility, nuance, and robustness matter. Moreover, the paper does not hypothesise or explain why the prosed restricted architecture would perform better than the more general classical transformer that subsumes it. This question is also not addressed by the theoretical analysis of the paper. The leaves the central claim that “positional attention enhances empirical OOD performance” not well-substantiated.

**Questions:**

Minor question:

Q1. Why are you parametrising the fixed attention matrices $A$ by the $W_q$ and $W_k$ matrices instead of directly parameterising $A$ (in Eq. 2)? Is it to reduce the model size? How would your results change if you parametrise $A$ directly?

---

> ### Author Response · Authors · 2024-11-24
> **Response to Reviewer k9Nh (Part 1/5)**
>
> We thank the reviewer for taking the time to read our manuscript and provide insightful feedback. We hope that, if satisfied, the reviewer might consider raising their score. Below, we respond to the reviewer’s concerns in detail, starting with a key point highlighted in the review.
>
> We appreciate the reviewer's interest in having a clear hypothesis in our paper. While we did not explicitly state this as a hypothesis, we did observe and discuss that transformers struggle with value generalization. Additionally, in our explanation of the architecture (e.g., as shown in Figure 1), we highlight the specific components that may contribute to overfitting.
>
> For clarity, we have made this hypothesis more explicit in our manuscript (lines 185-192 in orange) and we restate it below.
>
> We hypothesize that two factors contribute to poor scale generalization in standard Transformers:
> 1.  The inclusion of input values in the computation of the attention weights;
> 2.  The use of positional encodings in the input matrix
>
> are potential sources of overfitting during training. To address this, we decouple input values from attention weight computation and remove positional encodings from the input matrix, leading to positional attention. In practice, we validate our empirically validate our hypothesis, demonstrating that positional Transformers achieve better scale generalization within specific algorithmic reasoning tasks.
>
> As mentioned in our paper at lines 185-186, this hypothesis is inspired by the observation that many parallel algorithms do not rely on values within their communication frameworks. We further discuss how the presence of these components could explain the poor OOD performance of standard transformer models in Appendix E. We also provide an ablation study in Appendix G demonstrating how the removal of these two factors results in better performance.
>
> We really appreciate the reviewer's request for stating a clear hypothesis, as this helps to establish a consistent structure throughout our paper.
>
> > (1) Section 5 of the paper focuses offers a theoretical explanation of why the proposed positional transformers can simulate any algorithm falling under the Parallel Computation with Oracle Communication (PCOC) model. While it is indeed quite interesting that there is a rich class of computation that can be approximated with fixed attention, it is not clear how this theory ties in with the rest of the paper.
>
> Since the reviewer mentions "fixed attention", to clarify and ensure alignment, we reemphasize that the attention weights in our model are not fixed, but are a function of the positional encodings. The attention weights change at each layer. However, it is possible that we have misunderstood the reviewer’s point, and the reviewer meant that the attention is independent of the input values.
>
> The introduction of PCOC is motivated by the need to demonstrate the expressivity of our positional transformers. We do this because our architecture removes important components from the standard transformer architecture, namely:
> 1. Removing input values from the computation of attention weights;
> 2. Removing positional encodings from the input matrix.
>
> Section 5 investigates whether these restrictions do not compromise positional transformers’ ability to simulate parallel algorithms. To provide such expressivity guarantees, we examine how the proposed architecture can simulate some specific computational model. The fundamental differences between our proposed architecture and existing computational models
> require the development of a parallel model that better reflects the positional transformer architecture, hence the introduction of PCOC. Due to these differences between the architectures, we find this theoretical assessment necessary. We kindly encourage the reviewer to consider how the proposed architecture would lack substantiation without a formalized expressivity result.

---

> ### Author Response · Authors · 2024-11-24
> **Response to Reviewer k9Nh  (Part 2/5)**
>
> > (2) Most critically, how does this theory explain the observation that fixed attention result in better empirical performance?
>
> The expressivity results in our paper ensures that positional Transformers are at least theoretically capable of performing the algorithmic tasks of interest in this paper. It is not designed to address the learnability potential of the architecture, which is a separate question discussed in Appendix F. In short, we do not provide formal proofs on this matter, as such results would require precise control over the training process, which, to our knowledge, no existing theoretical framework adequately captures for models such as transformers. Additional gaps in this area are further discussed in Appendix F. We hope the reviewer recognizes that establishing theoretical OOD generalization bounds is beyond the scope of this work and is left as a direction for future research.
>
> That said, as mentioned previously, we propose a hypothesis for why positional Transformers demonstrate better OOD performance for the tasks studied here. This hypothesis is supported by extensive numerical experiments designed to validate it.
>
> > (3) Doesn't the referenced work of Sanford et al. imply that transformers with dynamic attention should also be able to approximate algorithms from the same class?
>
> While the transformer architecture can approximate algorithms within the same class, this does not guarantee that transformers can, through training, discover solutions that perform effectively in the out-of-distribution settings described in our paper. Although two methods may have the same expressivity (as we further elaborate in (5)), this does not imply that they will perform equally well after training. As noted, the components of standa
>
> > (4) The paper’s main thesis is that "positional attention enhances empirical OOD performance", however it does not propose a hypothesis explaining as to why that might be the case
>
> We kindly refer the reviewer to the initial discussion in the author response.
>
> > (5) I find this conclusion of the paper to be surprising and difficult to believe without further details. The authors construct a strictly less expressive architecture that has better performance.
>
> The reviewer's statement that positional transformers are strictly less expressive for simulating parallel algorithms is incorrect. As outlined in our paper, the expressivity of the proposed architecture is demonstrated through its ability to simulate the PCOC model. Furthermore, [1] establishes that computations performed by transformers are achievable by the MPC computational model. In Appendix A, we note that while PCOC uses data-agnostic communication, it can simulate MPC provided sufficient memory is available. This suggests that positional transformers are as expressive as MPC and standard transformers. However, achieving this equivalence for certain tasks may require additional depth or width in the architecture.
>
> In addition, as the reviewer may be aware, we point out that in practice it is typically not the case that the most expressive models have the best performance. For example, in traditional statistical learning, regularization terms are often added to a linear model to actually reduce model complexity, avoid overfitting and consequently improve performance. Even in the context of Large Language Models, depending on the size of the training data, it has been shown that larger, and hence more expressive, models often have worse performance than smaller (and hence strictly less expressive) models with appropriate model sizes (e.g. see Figure 3 in [2]).

---

> ### Author Response · Authors · 2024-11-24
> **Response to Reviewer k9Nh  (Part 3/5)**
>
> > (6) While that can be due to appropriate inductive biases, the classic transformer should be able to learn these tasks too. If the positional embeddings of the classic transformer are learnable and concatenated to the inputs, then isn’t it possible to show that for every positional Transformer, one can construct a classic transformer that is functionally equivalent to it?
>
> We kindly refer the reviewer to answer (3).
>
> > (7) Furthermore, as the authors also discuss in their related work section, we know that the classic transformer is a universal approximator for continuous and discrete functions, as well as Turing-complete. Hence, the classic transformer should also be able to solve the tasks considered in the paper.
>
> We kindly refer the reviewer to answers (5) and (6).
>
> > (8) In particular, the weights can be very sensitive to the scale of input values. We observe a dramatic change in attention weights as soon as we give the model the same input but scaled so that the values lie outside the domain of the training data. Perhaps I am missing something, but shouldn’t scaling the input by $\lambda$ scale all elements of the pre-softmax attention matrix by $\lambda^2$. And as scaling the pre-softmax activations is equivalent to adjusting the temperature of the softmax, isn't this difference in attention predictable and expected?All that makes me think that the reported results might be caused by technical details in the implementation of the experiments, rather than due to positional attention being fundamentally better for some reason.
>
> Due to the bias term in the encoding layer, key and query transformations, scaling the input by $\lambda$ does not scale the elements of the pre-softmax attention matrix by $\lambda^2$. The actual scaling depends on the learned bias terms and thus is not predictable. On the other hand, the reviewer is right in that the elements of the pre-softmax attention matrix will inevitably be affected by the scaling of the input. This is precisely the issue with standard transformers and it partially explains why they suffer from poor OOD performance. An important motivation for us to propose positional transformers is to solve this issue.
>
> Let us clarify that we are not trying to claim that positional attention is fundamentally better. It is never our goal and we, too, do not believe that positional attention is fundamentally better in every aspect. In this paper our goal is to enable good OOD performance of the transformer architecture for algorithmic tasks, and we propose positional attention as a solution to effectively address the instability issue of self-attention described above. Such results are particularly relevant to the community focused on Neural Algorithmic Reasoning [3, 4].

---

> ### Author Response · Authors · 2024-11-24
> **Response to Reviewer k9Nh  (Part 4/5)**
>
> > (9) Improving the ability of a Transformer to solve algorithms in isolation seems unnecessary: the example tasks used in the paper can be solved exactly with just a couple of lines of code. There would be value if the model uses these tasks as a subroutine of more complex tasks, e.g., given a natural text description of purchases, group them by type and compute commulative sums for each group. However, the paper does not study whether the proposed positional encodings work for more general data distributions, e.g., NLP tasks, different formatting of the same tasks, non-PCOC tasks, or mixtures of tasks. My intuition is that the ridgidity of the attention mechanism that improves the performance on the tasks considered in the paper, would be detrimental to the performance on such tasks that require flexibility, nuance, and robustness. And as LLMs are useful precisely for these less-structured tasks —for the structured ones, one is better off writing a Python script— I feel that the paper has failed to propose a modification of the transformer architecture that would result in improved performance in any realistic case. As such, the paper would likely be of limited value to the community.
>
> We appreciate the reviewer’s suggestion. Below, we describe a realistic application which requires complex and flexible reasoning. Briefly, we present the model with an input consisting of a mixture of alphanumeric and numeric elements corresponding to categories (alphanumeric) and prices (numeric). We then ask the model to output either the minimum/maximum/sum of a random subset of categories. We measure value generalization with respect to the category names (using unseen categories when testing) as well as the category prices (testing on larger prices). We present 3 experiments: one where the prompt is exclusively to calculate the minimum, one where the prompt is exclusively to calculate the sum and finally, one where the prompt either asks for minimum or maximum (which we call multitask). This task requires pattern matching, since the categories between the train and test data are different. It requires conditional reasoning, since the queries are about a sub-set of the categories in the samples. Finally, it requires algorithmic reasoning to compute the output.
>
> For details on our experimental setting, please refer to Appendix C.6 of the revised manuscript. The alphanumeric part of the input is tokenized and passed through an embedding layer, while the numeric part is passed through a linear layer. Below, we present the median MSE and MAPE losses for both standard and positional Transformer on all three experiments.
>
>
> **Median MSE for standard transformer:**
> | OOD Scaling factor |1|2|3|4|5|6|7|8|9|10|
> | :-------- | -------: |  -------: |  -------: |  -------: |  -------: |  -------: |  -------: |  -------: |  -------: |  -------: |
> min | 0.015 | 8.702 | 31.278 | 82.631 | 122.833 | 155.723 | 155.550 | 153.188 | 149.889 | 148.727
> sum | 0.128 | 5.329 | 24.528 | 46.977 | 58.523 | 97.080 | 120.254 | 139.406 | 173.005 | 187.943
> multitask | 0.017 | 2.574 | 15.124 | 27.309 | 54.959 | 94.314 | 146.044 | 199.593 | 270.039 | 347.843
>
> ---
>
> **Median MSE for positional transformer:**
> | OOD Scaling factor |1|2|3|4|5|6|7|8|9|10|
> | :-------- | -------: |  -------: |  -------: |  -------: |  -------: |  -------: |  -------: |  -------: |  -------: |  -------: |
> min | 0.012 | 0.090 | 0.207 | 0.383 | 0.580 | 0.913 | 1.376 | 1.986 | 2.733 | 3.121
> sum | 0.095 | 0.730 | 1.532 | 2.018 | 3.397 | 4.932 | 6.407 | 7.273 | 9.425 | 11.779
> multitask | 0.014 | 0.141 | 0.466 | 1.033 | 1.640 | 2.311 | 3.045 | 4.668 | 6.075 | 8.940
>
> ---
>
> **Median MAPE for standard transformer:**
> | OOD Scaling factor |1|2|3|4|5|6|7|8|9|10|
> | :-------- | -------: |  -------: |  -------: |  -------: |  -------: |  -------: |  -------: |  -------: |  -------: |  -------: |
> min | 4.68% | 50.86% | 63.99% | 74.71% | 83.90% | 83.00% | 79.46% | 75.62% | 66.50% | 59.20%
> sum | 2.28% | 6.04% | 9.60% | 12.39% | 13.03% | 13.05% | 12.30% | 12.42% | 11.62% | 11.37%
> multitask | 4.74% | 20.50% | 36.92% | 36.26% | 39.56% | 46.51% | 49.53% | 53.05% | 55.67% | 57.56%
>
> ---
>
> **Median MAPE for positional transformer**
> | OOD Scaling factor |1|2|3|4|5|6|7|8|9|10|
> | :-------- | -------: |  -------: |  -------: |  -------: |  -------: |  -------: |  -------: |  -------: |  -------: |  -------: |
> min | 3.53% | 4.64% | 5.62% | 5.40% | 5.66% | 6.41% | 5.64% | 6.25% | 7.06% | 7.19%
> sum | 2.23% | 3.16% | 2.86% | 2.81% | 2.45% | 2.64% | 2.61% | 2.78% | 2.70% | 2.85%
> multitask | 3.26% | 4.12% | 5.56% | 6.10% | 7.60% | 8.05% | 7.58% | 10.44% | 8.31% | 8.83%
>
> We note that, even in this complex relational task with mixed-type
> inputs, the positional Transformer still significantly outperforms the standard Transformer, demonstrating the potential utility of our application for certain real-world applications.

---

> ### Author Response · Authors · 2024-11-24
> **Response to Reviewer k9Nh  (Part 5/5)**
>
> > (10) Overall, despite the clear writing and good presentation, the paper has major conceptual and methodological flaws. The restricted architecture that is proposed could work well for the small task set the paper considers, but is unlikely to work well in practical settings where flexibility, nuance, and robustness matter. Moreover, the paper does not hypothesise or explain why the prosed restricted architecture would perform better than the more general classical transformer that subsumes it. This question is also not addressed by the theoretical analysis of the paper. The leaves the central claim that “positional attention enhances empirical OOD performance” not well-substantiated.
>
> We appreciate the reviewer’s feedback, but we find this particular comment to be disproportionate, as it does not present evidence of major conceptual or methodological flaws in our work. The authors thoroughly examined the reviewer's comments, and could not identify any specific issues that substantiate the claims of major flaws in the proposed architecture or methodology. Below, we address the specific points raised:
>
> > (10.1) [...] but is unlikely to work well in practical settings where flexibility, nuance, and robustness matter
>
> This statement appears unfair, as we demonstrate a realistic application in (9), which indeed requires flexibility, nuance, and robustness. We hope the reviewer appreciates the effort that was put in addressing their concerns, especially regarding the more realistic settings, which seems to be one of the reviewer's main point of concern.
>
> > (10.2) Moreover, the paper does not hypothesise or explain why the prosed restricted architecture would perform better than the more general classical transformer that subsumes it.  This question is also not addressed by the theoretical analysis of the paper.
>
> While we acknowledge that an explicit hypothesis is not stated in a dedicated section, as mentioned, we did observe and discuss that transformers struggle with value generalization. These factors provide a rationale for why the restricted architecture offers advantages in specific contexts.
>
> Furthermore, as stated in (2), the goal of the theoretical analysis was to address the expressability of the architecture, as some components are removed from the standard transformer's architecture.
>
> > (11) Why are you parametrising the fixed attention matrices  A by the Wq and Wk matrices instead of directly parameterising A (in Eq. 2)? Is it to reduce the model size? How would your results change if you parametrise A directly?
>
> This structure complies with the traditional attention structure and expresses some factorization of an attention matrix A. For fixed-length experiments with the dimension of positional encodings being equal to the sequence length, the two parameterizations (via $W_q$ and $W_k$ or directly using A) are equivalent since P is square. However, in variable-length experiments (Section 6.1), the attention matrix depends on a different P for each length, which breaks this equivalence. For fixed lengths, using positional encodings with number of columns less than n also restricts the attention patterns that can be represented, effectively imposing a low-rank condition on A. Appendix C.3 includes experiments with positional encodings where P is not square.
>
> Once again, we would like to thank the reviewer and invite them to respond with further comments.
>
> ---
> References:\
> [1] Sanford et al. Transformers, parallel computation, and logarithmic depth. ICML 2024\
> [2] Hoffmann et al. Training Compute-Optimal Large Language Models. NeurIPS 2022\
> [3] Bevilacqua et al. Neural Algorithmic Reasoning with Causal Regularisation. ICML 2023\
> [4] Rodionov et al. Neural Algorithmic Reasoning Without Intermediate Supervision. NeurIPS 2023

---

> ### Comment · Reviewer_k9Nh · 2024-11-26
>
> I would like to thank the authors for the extremely detailed and comprehensive response. I will try to respond to as much as I can.
>
> __On the hypothesis:__ Thank you for clarifying your hypothesis. Just to be on the same page, I do agree that you might be indeed seeing improved OOD performance on the tasks you are evaluating on. However, my concerns whether that applies to the Transformer architecture in general remain. One can come up with an architecture that performs really good on a specific task. However, the challenge is to do that while maintaining the performance on all other tasks. Nevertheless, I do see the theoretical value of your work.
>
> __On the attention weights:__ Perhaps I am missing a key detail, but I still fail to see how the attention weights are not fixed for a given input sequence length. Moreover, assuming causal masking is applied, that means that (up to the softmax normalisation) the attention weights of any sequence of length $n$ are a $n\times n$ sub-matrix of the attention weights of any sequence of length $m>n$. The parameterization you use might reduce the number of parameters that need to be stored (and the rank) but it does not change the fact that the mixing of information throughout the model is fixed (given a sequence length).
>
> > While the transformer architecture can approximate algorithms within the same class, this does not guarantee that transformers can, through training, discover solutions that perform effectively in the out-of-distribution settings described in our paper. Although two methods may have the same expressivity (as we further elaborate in (5)), this does not imply that they will perform equally well after training.
>
> While this is correct, the current paper also does not investigate the trainability of the Positional Transformer. Possibly, you can frame this as that you are providing inductive biases that are more aligned with the tasks you are considering.
>
> >  In Appendix A, we note that while PCOC uses data-agnostic communication, it can simulate MPC provided sufficient memory is available. This suggests that positional transformers are as expressive as MPC and standard transformers. However, achieving this equivalence for certain tasks may require additional depth or width in the architecture.
>
> While I haven't read Appendix A in detail, I am concerned by the fact that your proof appears to rely on an external oracle to provide the routing destinations (which feels like the difficult part) and that you are routing all the messages to all the nodes, which seem to mean that 1. your model size needs to grow with the input length, and 2. you are effectively showing that a single processor can simulate a multi-process task. That would further imply that, beyond the input layer, you might not even need the transformer, and possibly could achieve the same with an MLP.
>
> __On the additional experiments:__ While I greatly appreciate the effort that you've invested in designing and evaluating the additional experiments in Appendix C.6, I am afraid they still appear to miss the point. Looking at the examples on lines 1410-1420, they appear to be strictly formatted in a very precise and standardised way. As such, the model need not know the context, as it has every piece of information exactly where it expects it exactly the way it expects it. Which is probably why you constructed the task this way. My assumption is that if you add an extra nuisance token in the beginning, say ["Cat-1", "Cat 1", 1.2...] your performance will go close to random guess. I understand that you are not claiming that this experiment shows universal practical value of your proposed architecture but my concerns that it is unlikely to work well in practical settings where flexibility, nuance, and robustness matter.
>
> I do agree that the theoretical aspect of the paper is interesting. The authors have clearly put a great deal of effort and work in this paper. In light of that, I will increase my score to 5. However, as currently written, and in light of my above comments, it makes too strong claims on practical utility and importance. As such, I would encourage the authors to consider repositioning the paper to focus and expand on its theoretical contributions and resubmit.

---

> ### Author Response · Authors · 2024-11-27
> **Response to Reviewer k9Nh (Part 1/4)**
>
> > On the hypothesis: Thank you for clarifying your hypothesis. Just to be on the same page, I do agree that you might be indeed seeing improved OOD performance on the tasks you are evaluating on. However, my concerns whether that applies to the Transformer architecture in general remain. One can come up with an architecture that performs really good on a specific task. However, the challenge is to do that while maintaining the performance on all other tasks. Nevertheless, I do see the theoretical value of your work.
>
> We thank the reviewer for acknowledging the theoretical value of our work.
> However, we find it hard to understand the reviewer’s concept of generality, and why our work is being entirely judged by “applicability in general”. We are interested in solving algorithmic reasoning tasks, which are of interest to a wide community [1,2,3,4,5].
>
> Nevertheless, despite not being the focus of our paper, the reviewer seems to question that this applicability is not general enough. To satisfy the reviewer's concern, we provided additional experiments that, based on the reviewer's comment, match their conception of flexibility, nuance, and robustness. The reviewer then further argued with additional details, not part of the initial reply, for a more complex task, to which we also successfully address at the end of our response (under experiment details below).
>
> The reviewer further states that our work “makes too strong claims on practical utility and importance”. We ask the reviewer where in the paper such claims have been made, since our work was purposely designed to tackle algorithmic tasks.
>
> Considering the aspects mentioned above, as well as the additional experiments addressing their concerns, we respectfully ask the reviewer to reevaluate our work.
>
> >On the additional experiments: While I greatly appreciate the effort that you've invested in designing and evaluating the additional experiments in Appendix C.6, I am afraid they still appear to miss the point. Looking at the examples on lines 1410-1420, they appear to be strictly formatted in a very precise and standardised way. As such, the model need not know the context, as it has every piece of information exactly where it expects it exactly the way it expects it. Which is probably why you constructed the task this way. My assumption is that if you add an extra   in the beginning, say ["Cat-1", "Cat 1", 1.2...] your performance will go close to random guess. I understand that you are not claiming that this experiment shows universal practical value of your proposed architecture but my concerns that it is unlikely to work well in practical settings where flexibility, nuance, and robustness matter.
>
> We would like to point out that we structured the task this way because it aligns with the example proposed by the reviewer, and we closely followed their suggestion. Nevertheless, since the reviewer raised additional nuance concerns, we have implemented a modification of the new experiment where we inject nuisance strings in the input at random positions within the list. Taking it one step further, we also considered OOD generalization in the type of additional nuisance input. To be specific, we augment the training input list by injecting strings of the form `Cat+{id}` or `Cat-{id}` at random positions (`id` is one of the categories present in the true list chosen at random, `+` or `-` is also chosen randomly). For instance, this should generate the exact example proposed by the reviewer: ["Cat-1", "Cat 1", 1.2...], where “Cat-1” is the nuisance string that should be ignored in the computation. For testing, we inject strings of the form `Cat_{id}` or `Cat*{id}` with the same assumptions as before. Therefore, the train and test nuisance strings are different. Please refer to Appendix C.7 or the revised manuscript for more details. We highlight that even in this more nuanced setting, our model still outperforms the standard Transformer.
>
> [...]

---

> ### Author Response · Authors · 2024-11-27
> **Response to Reviewer k9Nh (Part 2/4)**
>
> **Median MSE for positional Transformer (with irrelevant structure):**
>
>
> | OOD Scaling factor |1|2|3|4|5|6|7|8|9|10|
> | :-------- | -------: |  -------: |  -------: |  -------: |  -------: |  -------: |  -------: |  -------: |  -------: |  -------: |
> min | 0.010 | 0.265 | 0.251 | 0.440 | 0.802 | 1.298 | 1.983 | 2.970 | 4.206 | 4.817
> sum | 0.094 | 0.872 | 1.428 | 2.305 | 3.463 | 5.178 | 6.355 | 10.623 | 13.432 | 16.923
> multitask | 0.016 | 0.140 | 0.561 | 1.026 | 1.679 | 3.096 | 4.211 | 7.230 | 7.070 | 9.997
> ***
>
> **Median MSE for standard Transformer (with irrelevant structure):**
>
> | OOD Scaling factor |1|2|3|4|5|6|7|8|9|10|
> | :-------- | -------: |  -------: |  -------: |  -------: |  -------: |  -------: |  -------: |  -------: |  -------: |  -------: |
> min | 0.016 | 5.125 | 27.519 | 60.279 | 96.470 | 137.361 | 159.923 | 180.919 | 209.009 | 252.358
> sum | 0.109 | 46.082 | 169.309 | 237.623 | 236.518 | 307.131 | 496.052 | 768.734 | 954.127 | 1092.671
> multitask | 0.018 | 1.094 | 5.785 | 11.105 | 12.456 | 14.069 | 20.393 | 31.993 | 48.419 | 67.631
> ***
>
> **Median MAPE for positional Transformer (with irrelevant structure):**
>
> | OOD Scaling factor |1|2|3|4|5|6|7|8|9|10|
> | :-------- | -------: |  -------: |  -------: |  -------: |  -------: |  -------: |  -------: |  -------: |  -------: |  -------: |
> min | 4.06% | 9.81% | 6.95% | 6.10% | 6.30% | 6.57% | 7.76% | 7.63% | 8.04% | 7.48%
> sum | 2.24% | 3.40% | 3.10% | 2.98% | 2.89% | 3.03% | 2.95% | 3.05% | 3.17% | 3.07%
> multitask | 4.76% | 5.79% | 6.60% | 6.91% | 7.78% | 7.64% | 7.75% | 8.94% | 8.85% | 8.15%
> ***
>
> **Median MAPE for standard Transformer (with irrelevant structure):**
>
>
> | OOD Scaling factor |1|2|3|4|5|6|7|8|9|10|
> | :-------- | -------: |  -------: |  -------: |  -------: |  -------: |  -------: |  -------: |  -------: |  -------: |  -------: |
> min | 4.83% | 26.30% | 44.61% | 53.68% | 62.15% | 64.70% | 64.20% | 62.54% | 61.95% | 60.23%
> sum | 2.48% | 23.80% | 30.09% | 35.61% | 32.64% | 27.62% | 24.23% | 26.94% | 27.83% | 30.56%
> multitask | 3.90% | 12.46% | 23.93% | 24.13% | 23.05% | 23.76% | 23.36% | 24.61% | 24.36% | 23.80%
> ***
>
> We hope that our detailed responses and the additional experiments address the reviewer’s concerns. We believe they demonstrate the flexibility, nuance, and robustness of our work, and trust that they will resolve any remaining questions.
>
> >On the attention weights: Perhaps I am missing a key detail, but I still fail to see how the attention weights are not fixed for a given input sequence length. Moreover, assuming causal masking is applied, that means that (up to the softmax normalisation) the attention weights of any sequence of length n are a n×n sub-matrix of the attention weights of any sequence of length m>n. The parameterization you use might reduce the number of parameters that need to be stored (and the rank) but it does not change the fact that the mixing of information throughout the model is fixed (given a sequence length).
>
> Your thinking is correct. Our previous remark re-explaining the architecture was just there to ensure we were on the same page, since other reviews suggested a misunderstanding. The attention weights depend on the positional encodings, and because the positional encodings are fixed, the attention weights are also fixed for a given input length once trained. However, they can still vary across layers and attention heads.

---

> ### Author Response · Authors · 2024-11-27
> **Response to Reviewer k9Nh (Part 3/4)**
>
> >“While the transformer architecture can approximate algorithms within the same class, this does not guarantee that transformers can, through training, discover solutions that perform effectively in the out-of-distribution settings [...]”
> While this is correct, the current paper also does not investigate the trainability of the Positional Transformer. Possibly, you can frame this as that you are providing inductive biases that are more aligned with the tasks you are considering.
>
> We do investigate the trainability of positional Transformers empirically, and these experiments are also aligned with the motivation of our proposed architecture, which is to solve algorithmic reasoning tasks. Maybe the reviewer expected a more technical analysis of this inductive bias, but in our previous response, we stated the reasons why a general PAC-style proof for this out-of-distribution setting is difficult (also discussed in Appendix F with an example).
> This is not a specific limitation of our work, as existing theoretical frameworks do not capture this nuance. For in-distribution generalization, we could establish an improvement in the complexity bound using an analysis similar to [6] and by exploiting the fact that our architecture breaks the dependence between the attention weights and the input values. However, such analysis is only in-distribution and does not generalize to the OOD setting that we are interested in. Alternatively, we could have made limiting and less practical assumptions for the sake of a theoretical result (such as infinite depth/width for NTK-like results), but the impact of such a result would not cover the practical aspects of our experiments. Providing an analysis to thoroughly study out-of-distribution would require new techniques, which due to its complexity, merit a paper of its own.
>
> >In Appendix A, we note that while PCOC uses data-agnostic communication, it can simulate MPC provided sufficient memory is available. This suggests that positional transformers are as expressive as MPC and standard transformers. However, achieving this equivalence for certain tasks may require additional depth or width in the architecture. While I haven't read Appendix A in detail, I am concerned by the fact that your proof appears to rely on an external oracle to provide the routing destinations (which feels like the difficult part) and that you are routing all the messages to all the nodes, which seem to mean that 1. your model size needs to grow with the input length, and 2. you are effectively showing that a single processor can simulate a multi-process task. That would further imply that, beyond the input layer, you might not even need the transformer, and possibly could achieve the same with an MLP.
>
> The focus of the paper is to establish the expressivity of Positional Transformers by simulating PCOC, which can implement various parallel algorithms. Our focus is not establishing efficient simulations across different parallel models. The discussion in Appendix A.1.1 is meant to be a complement of the limitations of PCOC in comparison to MPC, discussed in Section 5.2.
>
> Note that the strategy described in Appendix A is just one way to establish equivalence between PCOC and MPC, in case anyone is interested in such equivalences. While this equivalence is established using a complete network (i.e. sending all data to all machines), there are other ways to establish this equivalence in a less trivial way. For example, one could use communication networks such as Batcher [7] or Benes [8] networks, which are known to be universal, and therefore can realize any desired connection pattern between machines.
> For example, Benes networks establish pairwise communications, instead of “n-to-n” as described in our discussion, and because of this, they utilize significantly less memory.  As a consequence, such networks require more rounds to establish communication (O(log n) for Benes). In this example, no machine has access to all data, so you cannot simply use a single MLP to solve the task. Overall, we think that the value of establishing an equivalence given a different oracle is marginal. However, if the reviewer thinks that this is important, we would be happy to add this observation in the Appendix.

---

> ### Author Response · Authors · 2024-11-27
> **Response to Reviewer k9Nh (Part 4/4)**
>
> References:\
> [1] Yan et al. Neural Execution Engines: Learning to Execute Subroutines. NeurIPS 2020\
> [2] Numeroso et al. Dual Algorithmic Reasoning. ICLR 2023\
> [3] Bevilacqua et al. Neural Algorithmic Reasoning with Causal Regularisation. ICML 2023\
> Rodionov et al. Neural Algorithmic Reasoning Without Intermediate Supervision. NeurIPS 2023\
> [4] Diao and Loynd. Relational Attention: Generalizing Transformers for Graph-Structured Tasks. ICLR 2023\
> [5] Back de Luca and Fountoulakis. Simulation of Graph Algorithms with Looped Transformers. ICML 2024\
> [6] Zhang et al. Inductive Biases and Variable Creation in Self-Attention Mechanisms. ICML 2022\
> [7] Narasimha. The Batcher-Banyan self-routing network: universality and simplification. IEEE Transactions on Communications 1998\
> [8] Lee. A New Benes Network Control Algorithm. IEEE Transactions on Computers 1987

---

> > ### Author Response · Authors · 2024-11-28
> > **Additional Response to Reviewer k9Nh**
> >
> > To further demonstrate the superiority of positional Transformers when it comes to algorithmic tasks, even in this more nuanced setting, we took our experiments one step further and fine-tuned the pre-trained large version of GPT2 (774 million parameters) from Hugging Face. This experiment also failed to beat our much smaller positional Transformer! We hope that this experiment makes it clear that a powerful model, which is capable of general text generation, does not have better performance on the nuanced algorithmic reasoning tasks requested by the reviewer. All results and technical details are provided in the revised Appendix C.7, and we also summarize them in the tables below. We also repost the previously reported results on this task to facilitate comparisons:
> >
> > **Median MSE for positional Transformer (first setting of Appendix C.7, character-only tokenization):**
> >
> > | OOD Scaling factor |1|2|3|4|5|6|7|8|9|10|
> > | :-------- | -------: |  -------: |  -------: |  -------: |  -------: |  -------: |  -------: |  -------: |  -------: |  -------: |
> > min | 0.010 | 0.265 | 0.251 | 0.440 | 0.802 | 1.298 | 1.983 | 2.970 | 4.206 | 4.817
> > sum | 0.094 | 0.872 | 1.428 | 2.305 | 3.463 | 5.178 | 6.355 | 10.623 | 13.432 | 16.923
> > multitask | 0.016 | 0.140 | 0.561 | 1.026 | 1.679 | 3.096 | 4.211 | 7.230 | 7.070 | 9.997
> > ***
> >
> > **Median MSE for standard Transformer (first setting of Appendix C.7, character-only tokenization):**
> >
> > | OOD Scaling factor |1|2|3|4|5|6|7|8|9|10|
> > | :-------- | -------: |  -------: |  -------: |  -------: |  -------: |  -------: |  -------: |  -------: |  -------: |  -------: |
> > min | 0.016 | 5.125 | 27.519 | 60.279 | 96.470 | 137.361 | 159.923 | 180.919 | 209.009 | 252.358
> > sum | 0.109 | 46.082 | 169.309 | 237.623 | 236.518 | 307.131 | 496.052 | 768.734 | 954.127 | 1092.671
> > multitask | 0.018 | 1.094 | 5.785 | 11.105 | 12.456 | 14.069 | 20.393 | 31.993 | 48.419 | 67.631
> > ***
> >
> > **Median MAPE for positional Transformer (first setting of Appendix C.7, character-only tokenization):**
> >
> > | OOD Scaling factor |1|2|3|4|5|6|7|8|9|10|
> > | :-------- | -------: |  -------: |  -------: |  -------: |  -------: |  -------: |  -------: |  -------: |  -------: |  -------: |
> > min | 4.06% | 9.81% | 6.95% | 6.10% | 6.30% | 6.57% | 7.76% | 7.63% | 8.04% | 7.48%
> > sum | 2.24% | 3.40% | 3.10% | 2.98% | 2.89% | 3.03% | 2.95% | 3.05% | 3.17% | 3.07%
> > multitask | 4.76% | 5.79% | 6.60% | 6.91% | 7.78% | 7.64% | 7.75% | 8.94% | 8.85% | 8.15%
> > ***
> >
> > **Median MAPE for standard Transformer (first setting of Appendix C.7, character-only tokenization):**
> >
> >
> > | OOD Scaling factor |1|2|3|4|5|6|7|8|9|10|
> > | :-------- | -------: |  -------: |  -------: |  -------: |  -------: |  -------: |  -------: |  -------: |  -------: |  -------: |
> > min | 4.83% | 26.30% | 44.61% | 53.68% | 62.15% | 64.70% | 64.20% | 62.54% | 61.95% | 60.23%
> > sum | 2.48% | 23.80% | 30.09% | 35.61% | 32.64% | 27.62% | 24.23% | 26.94% | 27.83% | 30.56%
> > multitask | 3.90% | 12.46% | 23.93% | 24.13% | 23.05% | 23.76% | 23.36% | 24.61% | 24.36% | 23.80%
> > ***
> >
> > **Median MSE for fine-tuned GPT2 (third setting of Appendix C.7, fined-tuned GPT2):**
> >
> > | OOD Scaling factor |1|2|3|4|5|6|7|8|9|10|
> > | :-------- | -------: |  -------: |  -------: |  -------: |  -------: |  -------: |  -------: |  -------: |  -------: |  -------: |
> > min | 0.078 | 0.391 | 1.285 | 2.989 | 5.539 | 8.361 | 12.811 | 13.824 | 18.318 | 18.053
> > sum | 1.026 | 81.531 | 392.995 | 928.482 | 1795.709 | 2755.613 | 4056.421 | 5551.319 | 7082.871 | 9060.282
> > multitask | 0.107 | 1.326 | 3.551 | 8.850 | 11.614 | 15.932 | 22.069 | 28.775 | 31.080 | 35.961
> > ***
> >
> > **Median MAPE for fine-tuned GPT2  (third setting of Appendix C.7, fined-tuned GPT2):**
> > | OOD Scaling factor |1|2|3|4|5|6|7|8|9|10|
> > | :-------- | -------: |  -------: |  -------: |  -------: |  -------: |  -------: |  -------: |  -------: |  -------: |  -------: |
> > min | 6.29% | 7.78% | 10.08% | 12.39% | 13.02% | 13.91% | 16.60% | 16.60% | 17.50% | 16.55%
> > sum | 9.60% | 26.71% | 44.69% | 56.53% | 65.23% | 69.04% | 73.99% | 76.58% | 79.09% | 81.36%
> > multitask | 9.39% | 14.01% | 13.78% | 15.05% | 14.64% | 15.21% | 16.02% | 18.31% | 15.17% | 16.37%

---

> > > ### Author Response · Authors · 2024-11-29
> > >
> > > Could the reviewer confirm if our response has addressed the reviewer’s concerns? If the reviewer is satisfied, we would be grateful if they could consider raising the score.

---

### Official Review · Reviewer_77EA · 2024-10-28

**Soundness:** 2
**Presentation:** 3
**Contribution:** 2
**Rating:** 6
**Confidence:** 4

**Summary:**

The paper explores the enhancement of out-of-distribution (OOD) generalization in Transformers for algorithmic tasks by proposing the positional attention mechanism. Unlike standard self-attention, this approach calculates attention weights solely based on fixed positional encodings, maintaining expressivity. Through theoretical and empirical evidence, the authors demonstrate that positional Transformers can simulate parallel algorithms, achieving significant improvements in OOD value generalization over standard Transformers.

**Strengths:**

1. The methodology is well-explained, with diagrams and equations aiding in understanding the architecture and theoretical claims.

2. The paper includes a theoretical analysis that establishes the expressive power of positional Transformers by linking them to the Parallel Computation with the Oracle Communication (PCOC) model.

**Weaknesses:**

1. The model’s OOD generalization appears to primarily rely on two factors: (1) the independence of the attention matrix from the input $X$, and (2) the absence of one-hot encoding for the input. For example, in the cumulative sum task, positional attention only needs to equally weight the first $i$ tokens onto the  $i$-th token, which allows the output to approximately scale by the same factor as the input within a certain range. However, this design introduces notable limitations: first, positional attention is unable to model relative relationships between different inputs $X$, limiting its applicability to tasks that require more complex relative positioning. Second, this model appears suitable primarily for tasks involving numeric values that do not require one-hot encoding, constraining its utility in a broader range of applications.

2. Several minor typos and formatting inconsistencies were observed, such as missing periods on lines 156, 356, and 364. Addressing these issues would improve the manuscript’s professionalism and readability.

**Questions:**

In my view, the proposed architecture resembles a learnable adjacency matrix version of a Graph Neural Network (GNN), even with the explicit incorporation of positional information. Could the authors compare the results with those achieved by a GNN under the same sentence length? It would be interesting to see if the proposed method significantly outperforms GNNs, as such a comparison could highlight the unique advantages of this approach.

---

> ### Author Response · Authors · 2024-11-24
> **Response to Reviewer 77EA (Part 1/4)**
>
> We thank the reviewer for reading our manuscript and for the thoughtful comments. Below, we address the reviewer’s concerns one by one. We also provide additional experiments on complex relational reasoning task that might be of the reviewer's interest later on.
>
> We welcome further questions and hope our responses and experiments meet the reviewer’s expectations to increase their score.
>
> > (1) The model’s OOD generalization appears to primarily rely on two factors: the independence of the attention matrix from the input X, and the absence of one-hot encoding for the input.
>
> In the factors described by the reviewer, we believe that the second factor might have two different interpretations. The reviewer might refer to “one-hot encoding” as some representation related to the input tokenization, extending beyond the numerical data used in our previous experiments. To this end, we refer the reviewer to the results in response (3) which further describe a more complex setting that deals with numerical and textual data. In this setting, our proposed architecture also shows favorable OOD results with respect to both data modalities.
>
> Alternatively, the reviewer might refer to the one-hot positional encodings used in positional attention. To this end, we emphasize that our results do not require one-hot positional encodings. To substantiate this claim, in Appendix C.3, we present experiments utilizing other types of positional encodings (binary, sinusoidal), which also show good OOD generalization results.
>
> > (2) For example, in the cumulative sum task, positional attention only needs to equally weight the first i tokens onto the i-th token, which allows the output to approximately scale by the same factor as the input within a certain range.
>
> Although the reviewer proposes a plausible conceptual solution for the cumulative sum in the fixed-length regime, we argue that this explanation undermines the applicability of our method. Specifically, this reasoning does not extend to the variable-length regime (i.e., training and testing on sequences of up to n values), where a single scale factor cannot be universally applied to sequences of different lengths. In Section 6.1, we also demonstrate favorable results in this more challenging setting. Moreover, the reviewer's concern seems to primarily focus on our simplest task. Our study includes other tasks that require complex reasoning, where our method also performs competitively. We address these aspects further in the next comment.

---

> ### Author Response · Authors · 2024-11-24
> **Response to Reviewer 77EA  (Part 2/4)**
>
> > (3) However, this design introduces notable limitations: first, positional attention is unable to model relative relationships between different inputs X, limiting its applicability to tasks that require more complex relative positioning.
>
> The proposed architecture cannot adaptively model relationships between rows of X only using attention. However, this does not mean it cannot generally model relative relationships. We appreciate the reviewer’s observation and take this opportunity to clarify the architecture’s capacity to address relative reasoning tasks.
>
> Following [1], relational reasoning involves tasks where a model learns relationships between entities (e.g., numbers, text) independent of the specific entities observed. These tasks may include pattern matching, comparisons, and conditional selections. While our architecture performs well on relational reasoning tasks like sorting and finding minimums, which capture relative relationships within sequences, the reviewer seems unconvinced of its capability to handle more complex tasks. To address this, we present a more challenging setting, requiring relative reasoning over both numbers and text, which we describe below.
>
> Briefly, we present the model with an input consisting of a mixture of alphanumeric and numeric elements corresponding to categories (alphanumeric) and prices (numeric). We then ask the model to output either the minimum/maximum/sum of a random subset of categories. We measure value generalization with respect to the category names (using unseen categories when testing) as well as the category prices (testing on larger prices). We present 3 experiments: one where the prompt is exclusively to calculate the minimum, one where the prompt is exclusively to calculate the sum and finally, one where the prompt either asks for minimum or maximum (which we call multitask). This task requires pattern matching, since the categories between the train and test data are different. It requires conditional reasoning, since the queries are about a subset of the categories in the samples. Finally, it requires algorithmic reasoning to compute the output.
>
> For details on our experimental setting, please refer to Appendix C.6 of the revised manuscript. The alphanumeric part of the input is tokenized and passed through an embedding layer, while the numeric part is passed through a linear layer. Below, we present the median MSE and MAPE losses for both standard and positional Transformer on all three experiments.
>
> **Median MSE for standard transformer:**
> | OOD Scaling factor |1|2|3|4|5|6|7|8|9|10|
> | :-------- | -------: |  -------: |  -------: |  -------: |  -------: |  -------: |  -------: |  -------: |  -------: |  -------: |
> min | 0.015 | 8.702 | 31.278 | 82.631 | 122.833 | 155.723 | 155.550 | 153.188 | 149.889 | 148.727
> sum | 0.128 | 5.329 | 24.528 | 46.977 | 58.523 | 97.080 | 120.254 | 139.406 | 173.005 | 187.943
> multitask | 0.017 | 2.574 | 15.124 | 27.309 | 54.959 | 94.314 | 146.044 | 199.593 | 270.039 | 347.843
>
> ---
>
> **Median MSE for positional transformer:**
> | OOD Scaling factor |1|2|3|4|5|6|7|8|9|10|
> | :-------- | -------: |  -------: |  -------: |  -------: |  -------: |  -------: |  -------: |  -------: |  -------: |  -------: |
> min | 0.012 | 0.090 | 0.207 | 0.383 | 0.580 | 0.913 | 1.376 | 1.986 | 2.733 | 3.121
> sum | 0.095 | 0.730 | 1.532 | 2.018 | 3.397 | 4.932 | 6.407 | 7.273 | 9.425 | 11.779
> multitask | 0.014 | 0.141 | 0.466 | 1.033 | 1.640 | 2.311 | 3.045 | 4.668 | 6.075 | 8.940
>
> ---
>
> **Median MAPE for standard transformer:**
> | OOD Scaling factor |1|2|3|4|5|6|7|8|9|10|
> | :-------- | -------: |  -------: |  -------: |  -------: |  -------: |  -------: |  -------: |  -------: |  -------: |  -------: |
> min | 4.68% | 50.86% | 63.99% | 74.71% | 83.90% | 83.00% | 79.46% | 75.62% | 66.50% | 59.20%
> sum | 2.28% | 6.04% | 9.60% | 12.39% | 13.03% | 13.05% | 12.30% | 12.42% | 11.62% | 11.37%
> multitask | 4.74% | 20.50% | 36.92% | 36.26% | 39.56% | 46.51% | 49.53% | 53.05% | 55.67% | 57.56%
>
> ---
>
> **Median MAPE for positional transformer**
> | OOD Scaling factor |1|2|3|4|5|6|7|8|9|10|
> | :-------- | -------: |  -------: |  -------: |  -------: |  -------: |  -------: |  -------: |  -------: |  -------: |  -------: |
> min | 3.53% | 4.64% | 5.62% | 5.40% | 5.66% | 6.41% | 5.64% | 6.25% | 7.06% | 7.19%
> sum | 2.23% | 3.16% | 2.86% | 2.81% | 2.45% | 2.64% | 2.61% | 2.78% | 2.70% | 2.85%
> multitask | 3.26% | 4.12% | 5.56% | 6.10% | 7.60% | 8.05% | 7.58% | 10.44% | 8.31% | 8.83%
>
> We note that, even in this complex relational task with mixed-type
> inputs, the positional Transformer still significantly outperforms the standard Transformer, demonstrating the potential utility of our application for certain real-world applications.
>
> [...]

---

> ### Author Response · Authors · 2024-11-24
> **Response to Reviewer 77EA  (Part 3/4)**
>
> Furthermore, the capabilities of the architecture can also be analyzed through the lens of computational models. As outlined in our paper, the expressivity of the proposed architecture is demonstrated through its ability to simulate the PCOC model. Furthermore, [2] establishes that computations performed by transformers are achievable by the MPC computational model. In Appendix A, we note that while PCOC uses data-agnostic communication, it can simulate MPC provided sufficient memory is available. This suggests that positional transformers are as expressive as MPC and standard transformers. However, achieving this equivalence for certain tasks may require additional depth or width in the architecture.
>
> Finally, it is important to note that our arguments are not intended to claim that positional attention is fundamentally superior. We do not believe it outperforms in every aspect. Instead, our goal is to enhance the out-of-distribution (OOD) performance of transformer architectures on algorithmic tasks. Such results are particularly relevant to the community focused on Neural Algorithmic Reasoning [3, 4]. We propose positional attention as a solution to address self-attention’s shortcomings in such tasks.
>
> > (4) Second, this model appears suitable primarily for tasks involving numeric values that do not require one-hot encoding, constraining its utility in a broader range of applications.
>
> We believe the reviewer refers to “one-hot encoding” as a representation related to tokenization of the input, extending beyond the numerical data used in our previous experiments. If this interpretation is correct, we kindly refer the reviewer to response (3), where we demonstrate the effectiveness of our architecture in addressing a multimodal reasoning task.
>
> > (5) Several minor typos and formatting inconsistencies were observed, such as missing periods on lines 156, 356, and 364. Addressing these issues would improve the manuscript’s professionalism and readability.
>
> We appreciate the reviewer’s attentiveness in identifying these typos and inconsistencies. The suggested changes have been incorporated into the revised manuscript.
>
> > (6) In my view, the proposed architecture resembles a learnable adjacency matrix version of a Graph Neural Network (GNN), even with the explicit incorporation of positional information. Could the authors compare the results with those achieved by a GNN under the same sentence length? It would be interesting to see if the proposed method significantly outperforms GNNs, as such a comparison could highlight the unique advantages of this approach
>
> While the relationship between transformers and Graph Neural Networks (GNNs) has been well-established, it is important to clarify the differences between our architecture from GNNs. In short, positional Transformers “exchange messages” using a communication network that can change at every layer, whereas GNNs operate within a fixed, predefined graph structure at all layers.
>
> To address the reviewer’s curiosity, we compared the performance of our architecture against Graph Convolutional Networks (GCNs) and Graph Attention Networks (GAT). Since the tasks tested have no underlying native graph, we tested these models on complete and star graphs. Notably, the original GAT architecture on a complete graph is similar to a standard transformer but differs in that the value, key, and query weights are shared in GAT.
> Below, we present results reporting the median MSE for solving the cumulative sum and sorting tasks. We added these additional results to Appendix H of the paper.
>
> **Task: Cumulative Sum**
> | OOD Scaling factor |1|2|3|4|5|6|7|8|9|10|
> | :-------- | -------: |  -------: |  -------: |  -------: |  -------: |  -------: |  -------: |  -------: |  -------: |  -------: |
> GCN (complete graph) |3.86e+00 | 2.05e+01 | 4.25e+01 | 7.41e+01 | 1.19e+02 | 1.71e+02 | 2.31e+02 | 3.05e+02 | 3.88e+02 | 4.82e+02 |
> GCN (star graph) | 3.97e+00 | 2.01e+01 | 4.24e+01 | 7.44e+01 | 1.16e+02 | 1.68e+02 | 2.26e+02 | 3.07e+02 | 3.99e+02 | 4.62e+02 |
> GAT (complete graph) | 3.73e+00 | 1.99e+01 | 4.25e+01 | 7.47e+01 | 1.21e+02 | 1.80e+02 | 2.57e+02 | 3.32e+02 | 4.08e+02 | 5.42e+02 |
> GAT (star graph) | 3.83e+00 | 1.96e+01 | 4.13e+01 | 7.53e+01 | 1.16e+02 | 1.73e+02 | 2.35e+02 | 3.12e+02 | 4.13e+02 | 5.01e+02 |
>
> [...]

---

> ### Author Response · Authors · 2024-11-24
> **Response to Reviewer 77EA  (Part 4/4)**
>
> **Task: Sorting**
>
> | OOD Scaling factor |1|2|3|4|5|6|7|8|9|10|
> | :-------- | -------: |  -------: |  -------: |  -------: |  -------: |  -------: |  -------: |  -------: |  -------: |  -------: |
> GCN (complete graph) | 1.92e-01 | 9.79e-01 | 2.23e+00 | 3.92e+00 | 6.06e+00 | 8.93e+00 | 1.26e+01 | 1.66e+01 | 2.07e+01 | 2.67e+01 |
> GCN (star graph) | 1.92e-01 | 1.00e+00 | 2.08e+00 | 3.74e+00 | 6.08e+00 | 8.88e+00 | 1.23e+01 | 1.65e+01 | 2.12e+01 | 2.62e+01 |
> GAT (complete graph) | 1.92e-01 | 9.62e-01 | 2.10e+00 | 3.68e+00 | 5.61e+00 | 8.17e+00 | 1.09e+01 | 1.45e+01 | 1.90e+01 | 2.40e+01 |
> GAT (star graph) | 1.92e-01 | 9.93e-01 | 2.00e+00 | 3.51e+00 | 5.62e+00 | 7.83e+00 | 1.08e+01 | 1.46e+01 | 1.86e+01 | 2.31e+01 |
>
> As the results indicate, neither GCN nor GAT works very well even for in-distribution data (OOD scaling factor = 1), let alone achieving OOD generalization. We believe this is because these tasks are inherently unsuitable for standard message-passing architectures. If the reviewer has an alternative GNN architecture in mind, please let us know.
>
> **Additional theoretical comparisons:** From a computational perspective, the flexibility of attention allows positional Transformers to represent massively parallel algorithms, as there are no communication restrictions at each round. While standard and positional Transformers are aligned with parallel computational models due to their unrestricted communication, GNNs, on the other hand, are aligned with another type of computational model called distributed computing models [5], where communication is constrained by an input graph.
>
> Comparisons between parallel and distributed computational models are a popular topic [6], but are outside the scope of our paper. In the realm of neural networks and algorithms, a comparison between transformers and Graph Neural Networks has been established in Sec 5 of [2]. In theory, transformers are more efficient in the number of layers needed to simulate parallel algorithms, as their communication is more flexible than message-passing architectures. Overall, each one has its advantages and disadvantages. Since we do not deal with graph-based tasks in our paper, we specifically focus on the transformer architecture.
>
> We once again thank the reviewer and welcome any further questions.
>
> ---
> References:\
> [1] Boix-Adsera et al. When can transformers reason with abstract symbols? ICLR 2024\
> [2] Sanford et al. Transformers, parallel computation, and logarithmic depth. ICML 2024\
> [3] Bevilacqua et al. Neural Algorithmic Reasoning with Causal Regularisation. ICML 2023\
> [4] Rodionov et al. Neural Algorithmic Reasoning Without Intermediate Supervision. NeurIPS 2023\
> [5] Loukas, A. What graph neural networks cannot learn: depth vs width. ICLR 2020\
> [6] Das Sarma et al. Distributed Verification and Hardness of Distributed Approximation. STOC 2011

---

> > ### Comment · Reviewer_77EA · 2024-11-26
> >
> > Thank you for your detailed and thoughtful responses, as well as for providing additional experiments and analyses. Your efforts to address my earlier concerns are appreciated. Below, I outline two key concerns I still have after reviewing your responses and additional results:
> >
> > **OOD Generalization in Complex Reasoning Tasks:**
> >
> > While I understand your results demonstrate strong OOD performance in algorithmic tasks, the underlying mechanism appears inherently tied to problems where OOD arises from straightforward transformations (e.g., scaling of numeric values). Such tasks, while valuable, do not necessarily generalize to more intricate reasoning challenges like mathematical theorem proving or tasks requiring abstract symbolic reasoning. These types of tasks often involve OOD scenarios that extend beyond simple numeric transformations, such as logical derivations, multi-step reasoning, or structural pattern recognition. Without a clear extension to such domains, the claimed robustness of the proposed approach in OOD scenarios seems limited in scope.
> >
> > **Architectural Limitations for Position-Input Decoupling:**
> >
> > The proposed positional attention mechanism's design decouples the attention matrix from the input, which introduces constraints in modeling tasks where positional relationships between inputs and outputs are complex and dynamic. For instance, consider a task requiring the sum of all numbers preceding a special token (e.g., "0") in the output sequence. Such tasks require the architecture to dynamically integrate positional and content-based cues, which the fixed positional attention mechanism may struggle to represent effectively. While your additional experiments address some relative reasoning scenarios, the inability to handle more nuanced tasks involving dynamic input-output positional dependencies remains a concern.

---

> ### Author Response · Authors · 2024-11-26
> **Reply to Official Comment by Reviewer 77EA (Part 1/2)**
>
> We thank the reviewer for engaging in the discussion phase. We address the reviewer's additional concerns one by one below:
>
> >While I understand your results demonstrate strong OOD performance in algorithmic tasks, the underlying mechanism appears inherently tied to problems where OOD arises from straightforward transformations (e.g., scaling of numeric values).
>
> We would like to clarify that the new experiments provided in the rebuttal and Appendix C.6, perform OOD in both categories and numerical values. This is mentioned explicitly in our rebuttal and Appendix C.6 in the revised version of the paper. We beg the reviewer to read it carefully since this will alleviate a lot of confusion.
>
> The categories are alphanumeric, not numeric. Alphanumeric means that the input is a string containing both characters and numbers. For example, “Cat1” and “Cat2” correspond to Category 1 and Category 2, respectively. Therefore, one cannot simply perform straightforward transformations like scaling the input.
>
> >Such tasks, while valuable, do not necessarily generalize to more intricate reasoning challenges like mathematical theorem proving or tasks requiring abstract symbolic reasoning.
>
> We would like to emphasize that, in this work, we are not interested in theorem proving tasks, and as such, those tasks are completely out-of-scope in our paper. Nowhere in our manuscript or rebuttal did we mention that we are interested in solving such tasks. We find it a bit disproportionate and unfair to be judged on tasks that we never mentioned or cared to solve in the first place.
>
> >These types of tasks often involve OOD scenarios that extend beyond simple numeric transformations
>
> We would be grateful if the reviewer reads the rebuttal and Appendix C.6. We provided additional experiments to illustrate performance on tasks that combine alphanumeric (letters and numbers in string format) and numerical values.
>
> >such as logical derivations, multi-step reasoning, or structural pattern recognition.
>
> As we mention in our rebuttal and Appendix C.6 the new experiments do indeed require all the above. Namely, the new tasks require pattern matching, since the categories between the train and test data are different, conditional reasoning, since the queries are about a subset of the categories in the samples as well as algorithmic reasoning to compute the output.
>
> >Without a clear extension to such domains, the claimed robustness of the proposed approach in OOD scenarios seems limited in scope.
>
> We would be extremely grateful if the reviewer reads the rebuttal and Appendix C.6 for more details.
>
> >For instance, consider a task requiring the sum of all numbers preceding a special token (e.g., "0") in the output sequence.
>
> The task provided in Appendix C.6 illustrates exactly this point. The categories, which are in alphanumeric format, and thus cannot simply be scaled, are completely different in the train and test distributions. This means that the model needs to  (i) understand the appropriate symbols for categories unseen during training, (ii) isolate the unseen categories corresponding to the query, and (iii) perform the appropriate aggregation on the corresponding values.
>
> >Such tasks require the architecture to dynamically integrate positional and content-based cues, which the fixed positional attention mechanism may struggle to represent effectively.
>
> This is exactly what the task in Appendix C.6 demonstrates. As demonstrated, positional attention still outperforms standard attention in the setting of our chosen task.
>
> >While your additional experiments address some relative reasoning scenarios, the inability to handle more nuanced tasks involving dynamic input-output positional dependencies remains a concern
>
> We hope that we addressed this multiple times already. Please let us know if you have any further concerns.

---

> ### Author Response · Authors · 2024-11-26
> **Reply to Official Comment by Reviewer 77EA (Part 2/2)**
>
> For completeness, we restate the theoretical justification about why such tasks are possible for positional Transformers (repeated from the rebuttal above):
>
> The capabilities of the architecture can also be analyzed through the lens of computational models. As outlined in our paper, the expressivity of the proposed architecture is demonstrated through its ability to simulate the PCOC model. Furthermore, [1] establishes that computations performed by transformers are achievable by the MPC computational model. In Appendix A, we note that while PCOC uses data-agnostic communication, it can simulate MPC provided sufficient memory is available. This suggests that positional transformers are as expressive as MPC and standard transformers. However, achieving this equivalence for certain tasks may require additional depth or width in the architecture.
>
> Finally, it is important to note that our arguments are not intended to claim that positional attention is fundamentally superior. We do not believe it outperforms in every aspect. Instead, our goal is to enhance the out-of-distribution (OOD) performance of transformer architectures on algorithmic tasks. Such results are particularly relevant to the community focused on Neural Algorithmic Reasoning [2, 3]. We propose positional attention as a solution to address self-attention’s shortcomings in such tasks.
> ***
> References:\
> [1] Sanford et al. Transformers, parallel computation, and logarithmic depth. ICML 2024\
> [2] Bevilacqua et al. Neural Algorithmic Reasoning with Causal Regularisation. ICML 2023\
> [3] Rodionov et al. Neural Algorithmic Reasoning Without Intermediate Supervision. NeurIPS 2023

---

> > ### Comment · Reviewer_77EA · 2024-11-26
> >
> > Thanks for your reply, I apologize for the experiment I overlooked earlier and will modify the score based on your reply. Further, I am curious whether there can be a mechanism-level analysis that allows me to understand how this architecture accomplishes the tasks in Appendix C.6.

---

> > > ### Author Response · Authors · 2024-11-26
> > > **Response to Reviewer 77EA**
> > >
> > > We thank the reviewer for raising their score. We are grateful. We will get back to the reviewer's insightful question about a mechanism-level analysis tomorrow since it is currently late at night in our timezone.

---

> > > > ### Author Response · Authors · 2024-11-27
> > > > **Response to Reviewer's 77EA additional question**
> > > >
> > > > As the reviewer requested, we provide some intuition on how positional Transformers can solve the task discussed below. An interesting observation is that, while some solutions for this task depend on data for communication, others can be implemented using PCOC, ensuring communication remains independent of input values. To illustrate how positional Transformers might solve this task, we provide a sketch of a PCOC protocol:
> > > >
> > > > Matching our experimental setting, consider the case n=8 and k=4. For simplicity, assume that category-value pairs are assigned to the first eight machines, while the 9th machine holds the query prompt. In the first round, the query is distributed from the 9th machine to machines 1–8. Each machine locally computes whether its category matches the query, outputting 1 if true and 0 otherwise. For the next rounds, machines 1–8 communicate in a binary tree structure (as illustrated in Figure 6 in Appendix A). Local computations in these rounds apply the query function only to entries with an indicator value of 1, achieving the desired computation.
> > > >
> > > > Note that this is a simplified sketch of one possible approach to solving this task with communication that is data-oblivious. There could be many other ways to solve this task while maintaining this property.  Furthermore, consider that the experimental task is slightly more challenging as the input is tokenized character by character, and as such, each machine holds only a portion of the data.

---

### Official Review · Reviewer_K8dA · 2024-10-30

**Soundness:** 3
**Presentation:** 2
**Contribution:** 1
**Rating:** 3
**Confidence:** 4

**Summary:**

This paper proposes decoupling the query and key vectors from the value vectors, and making them global for every layer.
This results in a transformer structure that is learned but fixed at every layer.
The authors demonstrate on some simple synthetic tasks that the model still can perform these tasks, and that they generalise on size and values.

**Strengths:**

Generally, decoupling structure and computation might have some positive effects, depending on the task.
By having a separate key query decided at the start of the forward pass can achieve this decoupling, and depending on how they are computed, can help in generalisation.
The paper shows some tasks in which this is possible without explicit position embeddings and layer dependent attention routing, which may hold some promise for graph structured tasks and transformers (e.g. molecule-related)

**Weaknesses:**

The perspective of Transformers as a graph convolution network with a learned attention mask is [not new](https://thegradient.pub/transformers-are-graph-neural-networks/). Further, the decoupling of structure and layer-input is also [not novel](https://aclanthology.org/2022.acl-long.327.pdf).
The overall contribution of this paper is minimal, and I would suggest focusing on parameterisations of P, particularly task-specific, for this type of model.
The write-up on PCOC is also strongly reminiscent of message-passing in graphical models, so I'm not sure what this perspective contributes, though I may be missing something here.

**Questions:**

- Can you plot accuracy on the tasks you’ve benchmarked on? MSE doesn’t really seem informative here.
- Could you have more information / variants of how you compute P? This seems to be the most crucial element of the structure, now that it’s been decoupled from each layer’s input. What tasks work with just an embedding table? What tasks would require some prior processing from another Transformer model? etc.

**Details Of Ethics Concerns:**

No ethics concerns.

---

> ### Author Response · Authors · 2024-11-22
> **Response to Reviewer K8dA (Part 1/4)**
>
> We thank the reviewer for the comments. Below, we address their concerns point by point. If you have any additional questions or concerns, please let us know. We are more than happy to discuss them further.
>
> > This paper proposes decoupling the query and key vectors from the value vectors, and making them global for every layer.
>
> The reviewer’s statement suggests the impression that we learn a single attention mask and apply it universally across all layers. We would like to clarify that this is not the case, as we will explain in detail below.
>
> Upon revisiting certain text passages (lines 67 and 82), we recognize how the phrasing might have led to this interpretation. To prevent such misunderstandings, we have rectified these passages to ensure our intended meaning is clear and accurately represented.
>
> To clarify, we would like to restate our proposal briefly. We propose to detach the computation of query and key vectors from the value vectors and the MLPs, but this is an independent process for each layer. Importantly, there is no global structure that is propagated throughout the entire model. Instead, each query and key matrix, corresponding to its respective attention head, independently derives its values.
>
> The input to these query and key matrices consists of positional encodings, which are the same across all layers and attention heads. However, the outcomes of each attention head may differ because they are parameterized by distinct sets of weights.
>
> With this clarification, we address the weaknesses and questions made by the reviewer.
>
> ----
>
> > The perspective of Transformers as a graph convolution network with a learned attention mask is not new.
>
> Maybe we do not understand the implication the reviewer is trying to convey, but this widely-recognized perspective was never presented as a novelty in our work. Our paper does not involve graph neural networks or address graph-based problems.
>
> > Further, the decoupling of structure and layer-input is also not novel.
>
> Our work significantly differs from [1] in several aspects, as outlined below:
>
> - **Global Mask [1] vs. Layer-Specific Attention (ours):** In [1], a separate module computes a global mask represented as an undirected weighted graph derived from the input embeddings. In contrast, our approach does not involve a global mask or a dedicated module. Instead, attention weights are computed independently at every layer.
> - **Dependency on Input Embeddings:** In [1], attention weights rely on input embeddings to compute the global mask. In our approach, attention weights are computed solely based on positional encodings, independent of the input embeddings.
>
> Additionally, the setting of our work is fundamentally different, which could be another source of misunderstanding. Unlike [1], which focuses on dependency parsing for NLP tasks, our work investigates the ability of transformers to learn to execute algorithms, a capability with numerous potential applications [2].
>
> > The overall contribution of this paper is minimal, and I would suggest focusing on parameterisations of P, particularly task-specific, for this type of model.
>
> We believe that the reviewer is judging the contribution of our work based on the perspective that our architecture is reminiscent of that in Graph Neural Networks. However, we re-emphasize that this is not the case. While we appreciate the reviewer’s suggestion, in general, the parametrization of P is not critical for scale generalization in learning algorithms, as long as they remain expressive. In practice, full expressivity may not always be required to represent all communication patterns. To further substantiate this point, as suggested, we also provide empirical results with different positional encodings and observe no difference in performance, as discussed further below.

---

> ### Author Response · Authors · 2024-11-22
> **Response to Reviewer K8dA (Part 2/4)**
>
> > The write-up on PCOC is also strongly reminiscent of message-passing in graphical models,
>
> While both PCOC and message-passing models involve the “exchange of messages”, the mechanisms are fundamentally different. Specifically, PCOC exchanges messages using a network that can be changed at every round, while message-passing is restricted to a predefined graph structure. This flexibility allows PCOC to represent massively parallel algorithms since there are no communication restrictions at each round.
>
> Standard and Positional Transformers are aligned with parallel computational models, since their communication is not restricted. In contrast, Graph Neural Networks - which are aligned with another type of computational model called distributed computing models [3] - have their communication restricted by the input graph.
>
> Comparisons between parallel and distributed computational models are a popular topic [4], but are outside the scope of our paper. In the realm of neural networks and algorithms, a comparison between transformers and Graph Neural Networks has been established in Sec 5 of [5]. In theory, Transformers are more efficient in the number of layers needed to simulate parallel algorithms, as their communication is more flexible than the one of message-passing architectures. Overall, each one has its advantages and disadvantages. In our paper, since we do not deal with graph-based tasks, we specifically focus on the Transformer architecture.
>
> > I'm not sure what this perspective contributes, though I may be missing something here.
>
> The introduction of PCOC is motivated by the need to demonstrate the expressivity of our positional transformers. We do this because our architecture removes important components from the standard transformer architecture, namely:
> - Removing input values from the computation of attention weights;
> - Removing positional encodings from the input matrix.
>
> Section 5 investigates whether these restrictions do not compromise positional transformers’ ability to simulate parallel algorithms. To provide such expressivity guarantees, we examine how the proposed architecture can simulate some specific computational model. The fundamental differences between our proposed architecture and existing computational models
> require the development of a parallel model that better reflects the positional transformer architecture, hence the introduction of PCOC.

---

> ### Author Response · Authors · 2024-11-22
> **Response to Reviewer K8dA (Part 3/4)**
>
> > Could you have more information / variants of how you compute P? This seems to be the most crucial element of the structure, now that it’s been decoupled from each layer’s input. What tasks work with just an embedding table? What tasks would require some prior processing from another Transformer model? etc.
>
> We emphasize that the matrix P representing positional encodings is not computed, but rather provided as part of the input. This does not imply that the attention weights are fixed. Instead, P is used by the attention layer to determine how information from the input X is combined.
>
> ​​Theoretically, representing parallel algorithms requires the given P to be expressive, meaning it must enable the representation of any communication pattern among the input rows. For instance, one-hot encodings fulfill this requirement. However, other variations can be used in practice.
>
> To this end, we present results for the cumulative sum and sorting tasks, using two other alternatives: binary and sinusoidal positional encodings. Due to space constraints, we include results for only these two tasks; the full results for all tasks can be found in Appendix C.3, highlighted in purple.
>
> ---
> **Results for cumulative sum**
> | OOD /Scaling factor |1|2|3|4|5|6|7|8|9|10|
> | :-------- | -------: |  -------: |  -------: |  -------: |  -------: |  -------: |  -------: |  -------: |  -------: |  -------: |
> Standard Transformer (Binary PE) | 1.65e-05 | 9.97e-02 | 6.91e-01 | 3.23e+00 | 7.36e+00 | 1.18e+01 | 1.90e+01 | 2.72e+01 | 3.93e+01 | 5.09e+01 |
> Positional Transformer (Binary PE) | 2.09e-05 | 2.98e-04 | 1.47e-03 | 3.87e-03 | 7.79e-03 | 1.30e-02 | 1.86e-02 | 2.65e-02 | 3.54e-02 | 4.37e-02 |
> Standard Transformer (Sinusoidal PE) | 8.45e-05 | 1.52e+00 | 1.31e+01 | 3.50e+01 | 7.24e+01 | 1.22e+02 | 1.72e+02 | 2.37e+02 | 3.31e+02 | 4.13e+02 |
> Positional Transformer (Sinusoidal PE) | 1.30e-03 | 1.84e-02 | 5.25e-02 | 1.27e-01 | 2.47e-01 | 4.24e-01 | 5.61e-01 | 8.66e-01 | 1.17e+00 | 1.47e+00 |
>
> ---
>
> **Results for sorting**
> | OOD / Scaling factor |1|2|3|4|5|6|7|8|9|10|
> | :-------- | -------: |  -------: |  -------: |  -------: |  -------: |  -------: |  -------: |  -------: |  -------: |  -------: |
> Standard Transformer (Binary PE) | 9.58e-04 | 2.42e-01 | 1.84e+00 | 5.45e+00 | 1.04e+01 | 1.65e+01 | 2.36e+01 | 3.40e+01 | 4.51e+01 | 5.94e+01 |
> Positional Transformer (Binary PE) | 1.55e-04 | 1.75e-03 | 5.58e-03 | 1.19e-02 | 2.50e-02 | 4.46e-02 | 7.16e-02 | 9.73e-02 | 1.37e-01 | 1.84e-01 |
> Standard Transformer (Sinusoidal PE) | 1.34e-03 | 3.79e-01 | 2.57e+00 | 7.33e+00 | 1.41e+01 | 2.25e+01 | 3.18e+01 | 4.14e+01 | 5.03e+01 | 6.09e+01 |
> Positional Transformer (Sinusoidal PE) | 4.94e-04 | 5.68e-03 | 2.47e-02 | 8.38e-02 | 1.77e-01 | 3.03e-01 | 4.23e-01 | 6.37e-01 | 8.67e-01 | 1.07e+00 |
>
> ---
>
> As observed, less expressive positional encodings, such as sinusoidal encodings with $n/2$ dimensions and binary encodings with $\log n$ dimensions, also demonstrate consistently strong performance in positional transformers. This suggests that, in practice, learning certain algorithms may not require expressive positional encodings.

---

> > ### Comment · Reviewer_K8dA · 2024-11-27
> >
> > I understand that your paper does not involve graph neural networks or try to address those problems. However, my point was that
> > 1. GNNs can be viewed as a specific form of Transformers where the attention weights are fixed to 1 where the two nodes are connected. The connection between nodes are fixed given the graph structure, and therefore independent of the input.
> > 2. By using a layer-independent and input-independent P, you are keeping the weights at each layer fully determined by P. This would seem to be analogous to a weighted graph that is determined by P.
> >
> > In that sense, the structure you are proposing is very similar to GNNs, with the added benefit of being able to learn the graph.
> >
> > > We emphasize that the matrix P representing positional encodings is not computed, but rather provided as part of the input.
> >
> > This aspect further confuses me. Is P dependent on the input X? Or is it a set of embeddings learned per task?

---

> ### Author Response · Authors · 2024-11-22
> **Response to Reviewer K8dA (Part 4/4)**
>
> > Can you plot accuracy on the tasks you’ve benchmarked on? MSE doesn’t really seem informative here.
>
> Before presenting the additional experiments on accuracy, we would like to clarify to the reviewer that our experimental setup for all tasks is inherently regressive, as we work with real numbers and not integers. Specifically, our model processes a fixed-size list of real numbers as input and outputs another list of the same size, also consisting of real numbers. Consequently, there is no definitive “accuracy” measure one could consider. However, we consider the following two metrics which we believe closely resemble accuracy measures:
> 1. We evaluate the model on lists containing integers (while training is still done using real numbers) and round the model’s output to the nearest integer (or nearest 0.5 for the median task). A prediction is considered “correct” if the rounded list exactly matches the ground truth list.
> 2. We evaluate the model on lists of real numbers, considering a prediction "correct" if each entry in the predicted list is within an absolute precision of 0.05 and a relative precision of 5% compared to the corresponding entry of the ground truth list.
> ---
> **Accuracies of standard Transformer using (1):**
>
> | OOD / Scaling factor |1|2|3|4|5|6|7|8|9|10|
> | :-------- | -------: |  -------: |  -------: |  -------: |  -------: |  -------: |  -------: |  -------: |  -------: |  -------: |
> Cumulative sum | 1.00 | 0.90 | 0.58 | 0.37 | 0.19 | 0.10 | 0.07 | 0.02 | 0.01 | 0.01
> Cumulative min | 1.00 | 0.65 | 0.35 | 0.19 | 0.07 | 0.03 | 0.01 | 0.01 | 0.00 | 0.00
> Median | 0.94 | 0.08 | 0.00 | 0.00 | 0.00 | 0.00 | 0.00 | 0.00 | 0.00 | 0.00
> Sorting | 0.97 | 0.03 | 0.00 | 0.00 | 0.00 | 0.00 | 0.00 | 0.00 | 0.00 | 0.00
> Maximum sum subarray | 1.00 | 0.41 | 0.04 | 0.01 | 0.00 | 0.00 | 0.00 | 0.00 | 0.00 | 0.00
>
> ---
> **Accuracies of Positional Transformer using (1):**
>
> | OOD / Scaling factor |1|2|3|4|5|6|7|8|9|10|
> | :-------- | -------: |  -------: |  -------: |  -------: |  -------: |  -------: |  -------: |  -------: |  -------: |  -------: |
> Cumulative sum | 1.00 | 1.00 | 1.00 | 1.00 | 1.00 | 1.00 | 1.00 | 0.98 | 0.98 | 0.97
> Cumulative min | 1.00 | 1.00 | 1.00 | 1.00 | 0.99 | 0.98 | 0.95 | 0.93 | 0.89 | 0.85
> Median | 1.00 | 0.97 | 0.87 | 0.75 | 0.63 | 0.49 | 0.36 | 0.28 | 0.20 | 0.14
> Sorting | 1.00 | 0.99 | 0.91 | 0.78 | 0.63 | 0.46 | 0.32 | 0.18 | 0.10 | 0.05
> Maximum sum subarray | 1.00 | 0.99 | 0.96 | 0.88 | 0.77 | 0.66 | 0.54 | 0.46 | 0.35 | 0.29
>
> ---
> **Accuracies of standard Transformer using (2):**
>
> | OOD / Scaling factor |1|2|3|4|5|6|7|8|9|10|
> | :-------- | -------: |  -------: |  -------: |  -------: |  -------: |  -------: |  -------: |  -------: |  -------: |  -------: |
> Cumulative sum | 1.00 | 0.72 | 0.46 | 0.31 | 0.24 | 0.18 | 0.11 | 0.09 | 0.08 | 0.07
> Cumulative min | 1.00 | 0.83 | 0.60 | 0.45 | 0.38 | 0.27 | 0.24 | 0.15 | 0.14 | 0.12
> Median | 0.99 | 0.53 | 0.29 | 0.17 | 0.14 | 0.09 | 0.09 | 0.08 | 0.07 | 0.05
> Sorting | 0.83 | 0.18 | 0.07 | 0.05 | 0.02 | 0.02 | 0.02 | 0.01 | 0.00 | 0.01
> Maximum sum subarray | 1.00 | 0.54 | 0.26 | 0.13 | 0.11 | 0.06 | 0.04 | 0.04 | 0.04 | 0.02
>
> ---
> **Accuracies of Positional Transformer using (2):**
>
> | OOD / Scaling factor |1|2|3|4|5|6|7|8|9|10|
> | :-------- | -------: |  -------: |  -------: |  -------: |  -------: |  -------: |  -------: |  -------: |  -------: |  -------: |
> Cumulative sum | 1.00 | 1.00 | 1.00 | 0.99 | 0.99 | 0.98 | 0.97 | 0.96 | 0.95 | 0.96
> Cumulative min | 1.00 | 1.00 | 0.99 | 0.96 | 0.93 | 0.89 | 0.88 | 0.85 | 0.83 | 0.82
> Median | 1.00 | 0.97 | 0.93 | 0.89 | 0.83 | 0.82 | 0.76 | 0.74 | 0.73 | 0.70
> Sorting | 0.99 | 0.89 | 0.77 | 0.67 | 0.59 | 0.50 | 0.46 | 0.43 | 0.40 | 0.40
> Maximum sum subarray | 1.00 | 0.95 | 0.90 | 0.83 | 0.81 | 0.78 | 0.74 | 0.75 | 0.72 | 0.69
> ---
> The results are given in the table below and can also be located in Appendix C.8 of the revised manuscript (in purple). Given the unforgiving nature of those metrics (since one entry that fails to satisfy the criterion will cause the entire prediction to be labeled as “incorrect”) it is expected that as the OOD scale factor increases the accuracy will sharply decrease. However, we highlight that our architecture still grossly outperforms the standard transformer (which is consistent with the results presented when considering the MSE loss).
>
> We thank the reviewer again and invite any further questions. We hope that, if satisfied, the reviewer will consider raising their score.
>
> ---
> [1] Shen, Y., Tan, S., Sordoni, A., Li, P., Zhou, J., Courville, A. (2022). Unsupervised Dependency Graph Network
>
> [2] Veličković, P., Blundell, C. (2021) Neural Algorithmic Reasoning
>
> [3] Loukas, A. (2019). What graph neural networks cannot learn: depth vs width.
>
> [4] Das Sarma, A., et al. (2011). Distributed Verification and Hardness of Distributed Approximation
>
> [5] Sanford, C., Hsu, D., & Telgarsky, M. (2023). Representational Strengths and Limitations of Transformers

---

> > ### Author Response · Authors · 2024-11-27
> > **Reminder to Reviewer K8dA**
> >
> > Would the reviewer kindly confirm whether our response has adequately addressed all questions? If there are any remaining points of confusion, we would appreciate clarification. If the reviewer is satisfied, we would be grateful if they could consider raising the score.

---

> ### Author Response · Authors · 2024-11-27
> **Reply to Official Comment by Reviewer K8dA (Part 1/2)**
>
> > I understand that your paper does not involve graph neural networks or try to address those problems.
>
> Great! Thank you for clarifying, and replying to our rebuttal as well.
>
> >GNNs can be viewed as a specific form of Transformers where the attention weights are fixed to 1 where the two nodes are connected. The connection between nodes are fixed given the graph structure, and therefore independent of the input.
>
> Right! We agree with this.
>
> >By using a layer-independent and input-independent P, you are keeping the weights at each layer fully determined by P.
>
> This is correct. The attention weights in our architecture are fully determined by P. This represents the matrix of positional encodings, which can be one-hot, sinusoidal or binary, for example. In particular, the attention weights are a function of P. For the reviewer's convenience, we posted the definition of positional attention below, but for more information please see Figure 1 and Section 4 in our paper.
>
> >This would seem to be analogous to a weighted graph that is determined by P.
>
> If the reviewer implies that the attention weights are a parameterized function of P, then this statement is correct. If the reviewer means that the adjacency matrix is equal to P, then this is not what we do in this paper, since the algorithmic tasks we tackle (sorting, median, etc…) are not naturally related to graphs.
>
> >In that sense, the structure you are proposing is very similar to GNNs, with the added benefit of being able to learn the graph.
>
> This is a very important observation of the reviewer. Indeed, from a GNN perspective, our architecture is able to learn *a new weighted graph at every layer*. This is not possible for GNNs, where the graph structure is fixed at each layer. This is crucial and really boosts the performance of our method. Although the tasks that we are interested in are not graph-based tasks, we took the experimental results a step further by comparing our method against GNNs as well in Appendix H of the revised manuscript. We provide a summary of the results below.
>
> We compared the performance of our architecture against Graph Convolutional Networks (GCNs) and Graph Attention Networks (GAT). Since the tasks tested have no underlying native graph, we tested these models on complete and star graphs. Notably, the original GAT architecture on a complete graph is similar to a standard transformer but differs in that the value, key, and query weights are shared in GAT.
>
> Below, we present results reporting the median MSE for solving the cumulative sum and sorting tasks.
>
> Task: Cumulative Sum
> | OOD Scaling factor |1|2|3|4|5|6|7|8|9|10|
> | :-------- | -------: |  -------: |  -------: |  -------: |  -------: |  -------: |  -------: |  -------: |  -------: |  -------: |
> GCN (complete graph) |3.86e+00 | 2.05e+01 | 4.25e+01 | 7.41e+01 | 1.19e+02 | 1.71e+02 | 2.31e+02 | 3.05e+02 | 3.88e+02 | 4.82e+02 |
> GCN (star graph) | 3.97e+00 | 2.01e+01 | 4.24e+01 | 7.44e+01 | 1.16e+02 | 1.68e+02 | 2.26e+02 | 3.07e+02 | 3.99e+02 | 4.62e+02 |
> GAT (complete graph) | 3.73e+00 | 1.99e+01 | 4.25e+01 | 7.47e+01 | 1.21e+02 | 1.80e+02 | 2.57e+02 | 3.32e+02 | 4.08e+02 | 5.42e+02 |
> GAT (star graph) | 3.83e+00 | 1.96e+01 | 4.13e+01 | 7.53e+01 | 1.16e+02 | 1.73e+02 | 2.35e+02 | 3.12e+02 | 4.13e+02 | 5.01e+02 |
>
> Task: Sorting
> | OOD Scaling factor |1|2|3|4|5|6|7|8|9|10|
> | :-------- | -------: |  -------: |  -------: |  -------: |  -------: |  -------: |  -------: |  -------: |  -------: |  -------: |
> GCN (complete graph) | 1.92e-01 | 9.79e-01 | 2.23e+00 | 3.92e+00 | 6.06e+00 | 8.93e+00 | 1.26e+01 | 1.66e+01 | 2.07e+01 | 2.67e+01 |
> GCN (star graph) | 1.92e-01 | 1.00e+00 | 2.08e+00 | 3.74e+00 | 6.08e+00 | 8.88e+00 | 1.23e+01 | 1.65e+01 | 2.12e+01 | 2.62e+01 |
> GAT (complete graph) | 1.92e-01 | 9.62e-01 | 2.10e+00 | 3.68e+00 | 5.61e+00 | 8.17e+00 | 1.09e+01 | 1.45e+01 | 1.90e+01 | 2.40e+01 |
> GAT (star graph) | 1.92e-01 | 9.93e-01 | 2.00e+00 | 3.51e+00 | 5.62e+00 | 7.83e+00 | 1.08e+01 | 1.46e+01 | 1.86e+01 | 2.31e+01 |
>
> As the results indicate, neither GCN nor GAT work very well even for in-distribution data (OOD scaling factor = 1), let alone achieving OOD generalization. We believe this is because these tasks are inherently unsuitable for standard message-passing architectures. If the reviewer has an alternative GNN architecture in mind, please let us know.

---

> > ### Author Response · Authors · 2024-11-27
> > **Reply to Official Comment by Reviewer K8dA (Part 2/2)**
> >
> > >This aspect further confuses me. Is P dependent on the input X? Or is it a set of embeddings learned per task?
> >
> > P neither depends on the input X nor is it a set of embeddings learned per task. This represents the matrix of positional encodings, which can be one-hot, sinusoidal or binary, for example. This matrix is fixed for a certain input length and it is used to determine the attention weights at every layer/attention head.
> >
> > In our architecture, the attention weights are defined as follows:
> > \begin{equation}
> >     A^{(\ell, h)} = \text{softmax}\left(\left(PW^{(\ell, h)}_Q\right)\cdot \left(PW^{(\ell, h)}_K\right)^\top\right).
> > \end{equation}
> > Where $W^{(\ell, h)}_Q$ and $W^{(\ell, h)}_K$ are learnable matrices. Note that the parameters are superscripted by the layer index $\ell$ and the attention head index $h$, so they are different for every layer/attention head.

---

> > > ### Author Response · Authors · 2024-11-28
> > > **Additional Response to Reviewer K8dA**
> > >
> > > Based on the reviewer’s last question, we believe the reviewer believes that we must be using some task-specific graph information to determine the attention weights, either directly or through P. We would like to clarify that our paper demonstrates that generic, yet expressive, positional encodings are sufficient to determine good attention weights, resulting in strong OOD performance across multiple tasks. As we mentioned above, we set P to be one-hot, sinusoidal, or binary in our experiments, and the additional experiments that the reviewer requested. No graph- or task-specific relational information is used in our attention mechanism, explicitly or through P. This is what sets our architecture apart and enables robust OOD performance compared to the Transformer for the tasks that we care about.
> > >
> > > We would also be grateful if the reviewer could let us know if they have any more concerns, and if satisfied, consider raising their score.

---

> > > > ### Author Response · Authors · 2024-11-29
> > > >
> > > > Does the reviewer plan to address the remaining points? The authors have addressed the reviewer’s concerns, yet the response has not been discussed further. It is unclear whether the reviewer finds the clarifications satisfactory or if other issues remain. If the clarifications are satisfactory, could the reviewer kindly consider updating their evaluation score?

---

### Official Review · Reviewer_DxTU · 2024-11-03

**Soundness:** 3
**Presentation:** 3
**Contribution:** 2
**Rating:** 5
**Confidence:** 2

**Summary:**

This paper proposes that using fixed attention patterns with only positional information (positional transformers) can enhance transformer capabilities for tasks such as sorting and minsum. The authors explain this phenomenon by demonstrating that positional transformers can effectively implement any algorithm defined in a parallel computation model (PCOC), and use experiments to valid their findings.

**Strengths:**

The paper demonstrates both experimentally and theoretically that attention mechanisms utilizing only positional information possess significant expressive power, presenting interesting implications for future research directions.

**Weaknesses:**

While the paper effectively demonstrates the expressive power of positional transformers, I have two main concerns:

1. Regarding the theoretical analysis of positional transformers' expressive power (Section 5):
The argument that positional transformers can simulate PCOC seems somewhat not surprising, given that MPC can simulate PCOC (with more rounds), PCOC can be considered a special case of MPC. While Lines 123-129 argue that "MPC requires input data to specify destinations for communication," these destinations are not truly "specified by input," as each destination is computed from the source with detail-designed mlp[1]. Since (one-hot) positional information can easily implement any fixed attention pattern, this theoretical contribution seems limited, especially considering that [1] achieved similar results without utilizing positional information.

2. Regarding the experimental validation (Section 6):
The experimental setup may not adequately demonstrate positional transformers' superiority over vanilla transformers in these tasks. The input data for vanilla transformers are sequences of scalar numbers, and adding absolute positional information to these scalars significantly impacts the original data. This naturally disadvantages vanilla transformers, as they must simultaneously process both the original data (for value/output) and positional information (for attention) by their sum. Maybe some additional experiments, such as use *nonparallel* vectors to denote each (integer) number and let model predict target label y one by one, would be more convincing.

[1] Sanford, C., Hsu, D., & Telgarsky, M. (2024). Transformers, parallel computation, and logarithmic depth.

**Questions:**

Please refer to the weaknesses section.

**Details Of Ethics Concerns:**

As highlighted in the weaknesses section, it may necessary to present more robust and well-supported experiments to support their findings. If there are any misunderstandings on my part, please point them out, and I will reconsider my evaluation of this work.

---

> ### Author Response · Authors · 2024-11-22
> **Response to Reviewer DxTU (Part 1/5)**
>
> We sincerely thank the reviewer for their time and insights on our manuscript. Below, we address the reviewer’s questions and concerns one by one. Please let us know if you have more questions or concerns. We are happy to discuss them.
>
>
> > The argument that positional transformers can simulate PCOC seems somewhat not surprising, given that MPC can simulate PCOC (with more rounds), PCOC can be considered a special case of MPC. [...]
>
>
> The fact that MPC could simulate PCOC only establishes a relation between two distinct parallel computational models. However, this connection does not provide insights into the expressive power of positional transformers, which remains our primary focus in our theoretical analysis.
> The authors find it challenging to discern the specific implications of the reviewer’s comment. We propose addressing the reviewer’s concern from two potential perspectives:
>
> **Why is our theoretical contribution necessary?**
>
> We would like to begin by clarifying the motivation behind our analysis. The positional transformer architecture differs from the standard transformer architecture by:
>
> 1. Removing input values from the computation of attention weights;
>
> 2. Removing positional encodings from the input matrix.
>
> Section 5 investigates whether these restrictions do not compromise positional transformers’ ability to simulate parallel algorithms. Due to these differences between the architectures, we find this theoretical assessment necessary. We kindly encourage the reviewer to consider how the proposed architecture would lack substantiation without a formalized expressivity result.

---

> ### Author Response · Authors · 2024-11-22
> **Response to Reviewer DxTU (Part 2/5)**
>
> **Is our theoretical contribution a corollary of [1]?**
>
> The results in [1] do not imply our theoretical results. In this context, to provide expressivity guarantees, we examine if our proposed architecture can simulate some computational model. While MPC could be a candidate (which could simulate PCOC), its assumptions do not fully align with the proposed architecture, namely because:
> - MPC allows input values to determine communication across machines.
>
> - MPC imposes sublinear memory constraints w.r.t. the number of machines.
>
> These fundamental differences require the development of a parallel model that better reflects the positional transformer architecture, hence the introduction of PCOC. Regarding the specific simulation proof technique utilized in our work, we emphasize that the results in [1] do not apply to our setting, and our theoretical results differ in several ways with important practical implications. We discuss this in detail below.
>
> The primary difference between the two approaches lies in how communication is modeled. In our setting, communication is executed exclusively by the attention layer. The parameters in $W_Q$ and $W_K$ determine how information is propagated across machines, which aligns with the oracle assumption of our computational model. In contrast, as the reviewer mentions, the communication in MPC can be determined by local functions. Consequently, in the proof of [1], this is a two-stage process in which row-wise MLPs are first required to encode the destinations. In the second stage, $W_Q$ and $W_K$ transform this information into appropriate encodings (Proposition B.1 in [1]).
>
> This approach would only directly translate to our setting with additional modifications to the architecture. In [1], MLPs generate destinations, while in our architecture, they do not influence the computation of attention weights. Adapting this strategy would involve transforming $W_Q$ and $W_K$ into MLPs and using them as look-up tables to map sources and destinations, as in [1]. This deviates from our intended design and the standard transformer architecture, which uses linear transformations inside attention and often relies on explicit positional information. Moreover, incorporating MLPs into the computation of attention weights unnecessarily increases model complexity.
>
> Considering all these aspects, we believe that there are too many critical differences among the papers that justify a separate expressivity result for our work.

---

> ### Author Response · Authors · 2024-11-22
> **Response to Reviewer DxTU (Part 3/5)**
>
> > […] While Lines 123-129 argue that "MPC requires input data to specify destinations for communication," these destinations are not truly "specified by input," as each destination is computed from the source with detail-designed mlp[1].
>
> We appreciate the reviewer’s attention to this inconsistency in our text. To clarify, MPC does not require the destination to be explicitly specified in the input, as it can be inferred through local functions. A more precise way to convey our argument is that communication in MPC may not be agnostic to input values. We updated our manuscript to rectify the passage outlined by the reviewer. We point to passages 126-131, 231-251, and 730-740 in the revised manuscript encompassing these changes, highlighted in blue.
>
> > [...] Since (one-hot) positional information can easily implement any fixed attention pattern, this theoretical contribution seems limited, especially considering that [1] achieved similar results without utilizing positional information.
>
> While we respect the reviewer’s perspective, we believe the comparison requires further elaboration. Although we acknowledge [1] as an excellent contribution, certain assumptions made in [1] may have been overlooked by the reviewer, as outlined below.
>
> First, the reviewer states that no positional information is utilized in [1], but this is not entirely accurate. Their proof strategy requires row-specific MLPs (e.g., MLP_1, MLP_2, etc.) during the communication phase. While this approach is valid (see [2, 3]), it inherently encodes positional information through indices. In a more conventional setting, replacing these row-specific MLPs with a single, larger MLP capable of representing all local MLPs would necessitate distinct positional identifiers for each row input. Without this positional information, rows with identical inputs would produce identical outputs, contradicting the requirements of their proof.
>
> The reviewer might further argue that relying on one-hot positional encodings could trivialize the communication aspect of our proof, as works such as [1] may not require such expressive positional representations. While the reviewer may view this as a limiting assumption, we argue that the proof technique of [1] also makes limiting assumptions that are subject to debate. For example, in [1] and [3], local MLPs effectively function as a look-up table that maps source and destinations. However, the complexity of this operation and its dependence on the number of inputs is not explicitly characterized in [1] nor [3]. Thus, despite avoiding positional encodings, these works depend on potentially complex operations within their MLPs.
>
> Ultimately, the two different approaches are equally valid. In our proof, we leverage explicit positional information, which we consider more conventional in practice due to its empirical success and the avoidance of row-specific MLPs.
>
> Furthermore, in practice, full expressivity may not always be required to represent all communication patterns. In our experiments, we do not observe changes in performance utilizing other types of positional encodings (see Appendix C.3 in the revised manuscript for additional results).

---

> ### Author Response · Authors · 2024-11-22
> **Response to Reviewer DxTU (Part 4/5)**
>
> > Regarding the experimental validation (Section 6): [...]
>
> We agree with the reviewer that the observed results may stem from differences in input formats, but this aligns with our intended assumptions and architectural design. Specifically, our architecture separates input values and positional information into two distinct streams: (a) one dedicated to the value matrix and MLPs, and (b) another for the query and key matrices—an approach not feasible in standard transformer architectures.
>
> Furthermore, we emphasize that the standard transformer architecture’s expressiveness depends on positional encodings [4], which are explicitly added to the input. Without these encodings, the standard transformer architecture becomes permutation-equivariant [5], which is unsuited for algorithmic tasks such as sorting (permutation invariant).
>
> To investigate this further, we have conducted additional experiments exploring variations in the data fed into these inputs to examine whether performance improvements can be attributed solely to any specific factor. Before detailing these experiments, we wish to correct a crucial detail in the reviewer’s comment: our work does not exclusively use integer data and thus cannot adopt the tokenization scheme proposed. Nonetheless, we attempted to accommodate this feedback in variation (2). The variations include the following:
>
> 1. **Standard Transformers:** Numbers and positional encodings for the MLPs, value, query, and key matrices.
> 2. **Standard Transformers with non-parallel input:** Input numbers are placed in one-hot positions, that is, the input is Diag(X) where X is the input list to other models. No additional positional encodings are used.
> 3. **Positional Transformers:** Numbers for the MLPs and value matrix, and positional encodings for the query and key matrices.
> 4. **Misaligned Positional Transformers:** Numbers for the MLPs and value matrix; Positional encodings for the MLPs, value, query, and key matrices (adding positional encodings to the input).
> 5. **Input-regularized Standard Transformers:** Numbers for the MLPs, value, query, and key matrices; Positional encodings only in query and key matrices.
> 6. **No Positional Encodings:** Numbers for the MLPs, value, query, and key matrices.
> 7. **Using RoPE Only:** Numbers in MLPs, value, query, and key matrices, removing absolute positional encodings, and using only RoPE in standard transformers.
>
> Below, we show the results for two tasks: cumulative sum and sorting. Their respective plots are also illustrated in Appendix G, highlighted in blue.
>
> ---
> **Results for cumulative sum:**
> | OOD / Scaling factor |1|2|3|4|5|6|7|8|9|10|
> | :-------- | -------: |  -------: |  -------: |  -------: |  -------: |  -------: |  -------: |  -------: |  -------: |  -------: |
> Standard Transformer | 1.39e-05 | 2.86e-01 | 2.41e+00 | 6.77e+00 | 1.52e+01 | 2.74e+01 | 4.39e+01 | 6.32e+01 | 8.18e+01 | 1.04e+02 |
> Standard Transformer (non-parallel input) | 1.13e-03 | 4.94e-01 | 1.07e+01 | 5.44e+01 | 1.37e+02 | 2.11e+02 | 3.94e+02 | 6.75e+02 | 9.80e+02 | 1.39e+03 |
> Positional Transformer | 4.53e-06 | 1.49e-04 | 7.22e-04 | 1.92e-03 | 3.83e-03 | 6.49e-03 | 1.01e-02 | 1.51e-02 | 2.06e-02 | 2.59e-02 |
> Misaligned Positional Transformer | 1.51e-05 | 5.50e-01 | 4.52e+00 | 1.42e+01 | 3.10e+01 | 5.29e+01 | 8.20e+01 | 1.24e+02 | 1.61e+02 | 1.99e+02 |
> Input-regularized Standard Transformer | 2.71e-05 | 4.14e-02 | 7.57e-01 | 4.95e+00 | 1.74e+01 | 4.02e+01 | 8.80e+01 | 1.54e+02 | 2.36e+02 | 3.59e+02 |
> No positional encoding | 3.98e+00 | 3.73e+01 | 1.21e+02 | 3.07e+02 | 6.03e+02 | 9.11e+02 | 1.40e+03 | 1.88e+03 | 2.46e+03 | 3.35e+03 |
> RoPE only | 3.99e+00 | 3.73e+01 | 1.57e+02 | 3.43e+02 | 5.69e+02 | 9.12e+02 | 1.31e+03 | 1.91e+03 | 2.41e+03 | 2.93e+03 |
>
> ---
> **Results for sorting:**
> | OOD / Scaling factor |1|2|3|4|5|6|7|8|9|10|
> | :-------- | -------: |  -------: |  -------: |  -------: |  -------: |  -------: |  -------: |  -------: |  -------: |  -------: |
> Standard Transformer | 8.67e-04 | 3.10e-01 | 1.40e+00 | 3.74e+00 | 7.96e+00 | 1.25e+01 | 1.81e+01 | 2.57e+01 | 3.16e+01 | 4.02e+01 |
> Standard Transformer (non-parallel input) | 4.73e-04 | 1.33e-01 | 9.19e-01 | 2.41e+00 | 4.90e+00 | 9.57e+00 | 1.35e+01 | 2.04e+01 | 2.65e+01 | 3.57e+01 |
> Positional Transformer | 1.57e-04 | 1.02e-03 | 2.77e-03 | 6.23e-03 | 1.18e-02 | 1.93e-02 | 2.96e-02 | 4.15e-02 | 5.67e-02 | 7.24e-02 |
> Misaligned Positional Transformer | 2.95e-04 | 1.25e-02 | 1.24e-01 | 4.79e-01 | 1.17e+00 | 2.20e+00 | 3.79e+00 | 5.66e+00 | 8.94e+00 | 1.16e+01 |
> Input-regularized Standard Transformer | 1.85e-05 | 1.28e-01 | 8.05e-01 | 2.68e+00 | 5.34e+00 | 9.67e+00 | 1.43e+01 | 2.12e+01 | 2.72e+01 | 3.65e+01 |
> No positional encoding | 2.51e-01 | 2.79e+01 | 1.43e+02 | 3.75e+02 | 9.30e+02 | 1.56e+03 | 2.28e+03 | 3.46e+03 | 4.37e+03 | 5.43e+03 |
> RoPE only | 2.59e-01 | 3.11e+01 | 2.05e+02 | 5.69e+02 | 1.32e+03 | 2.13e+03 | 2.99e+03 | 4.52e+03 | 5.81e+03 | 7.44e+03 |
>
> ---
>
> [...]

---

> ### Author Response · Authors · 2024-11-22
> **Response to Reviewer DxTU (Part 5/5)**
>
> As demonstrated, none of these variations can even achieve reasonably low test loss over OOD data. For example, the losses for all other variations are at least 100x higher than those of the Positional Transformer. This highlights the importance of the following 3 unique features of Positional Transformer, which enabled its good OOD generalization performance: (1) use positional encodings, (2) do not use positional encodings in the value matrix of the attention, (3) use only fixed positional encodings for computing the attention weights. Note that these align with algorithms that are typically used to solve algorithmic tasks.
>
> Regarding the second suggestion (“predict target label y one by one”), we believe the reviewer may refer to an autoregressive approach, where each cumulative prediction is sequential (e.g., appending the i-th element at step i). While common in NLP, such approaches inject greater supervision than our algorithmic setting for cumulative tasks due to the incremental addition of elements. Also, this approach is inherently sequential, diverging from the parallel models explored here and in related works such as [1]. If we have misunderstood or if the reviewer has suggestions on how this setting could improve results, we welcome further discussion.
>
> Once again, we sincerely thank the reviewer and welcome any further comments. We hope the clarifications and additional results provided are sufficient for them to consider improving their score.
>
> ---
> [1] Sanford, C., Hsu, D., & Telgarsky, M. (2024). Transformers, parallel computation, and logarithmic depth
>
> [2] Loukas, A. (2019). What graph neural networks cannot learn: depth vs width
>
> [3] Sanford, C., Hsu, D., & Telgarsky, M. (2023). Representational Strengths and Limitations of Transformers
>
> [4] Peréz, J., Barceló, P., & Marinkovic, J. (2022) Attention is Turing Complete
>
> [5] Tsai, Y., Bai, S., Yamada, M., Morency, L., Salakhutdinov, R. (2019) Transformer Dissection: A Unified Understanding of Transformer's Attention via the Lens of Kernel

---

> ### Author Response · Authors · 2024-11-24
> **Additional response to Reviewer DxTU**
>
> Responding to the reviewer's concern about the encoding of the input we present complementary results on a more challenging setting, requiring relative reasoning over both numbers and text, which we describe below. Briefly, we present the model with an input consisting of a mixture of alphanumeric and numeric elements corresponding to categories (alphanumeric) and prices (numeric). We then ask the model to output either the minimum/maximum/sum of a random subset of categories. We measure value generalization with respect to the category names (using unseen categories when testing) as well as the category prices (testing on larger prices). We present 3 experiments: one where the prompt is exclusively to calculate the minimum, one where the prompt is exclusively to calculate the sum and finally, one where the prompt either asks for minimum or maximum (which we call multitask). This task requires pattern matching, since the categories between the train and test data are different. It requires conditional reasoning, since the queries are about a subset of the categories in the samples. Finally, it requires algorithmic reasoning to compute the output.
>
> For details on our experimental setting, please refer to Appendix C.6 of the revised manuscript. The alphanumeric part of the input is tokenized and passed through an embedding layer, while the numeric part is passed through a linear layer. Below, we present the median MSE and MAPE losses for both standard and positional Transformer on all three experiments.
>
> **Median MSE for standard transformer:**
> | OOD Scaling factor |1|2|3|4|5|6|7|8|9|10|
> | :-------- | -------: |  -------: |  -------: |  -------: |  -------: |  -------: |  -------: |  -------: |  -------: |  -------: |
> min | 0.015 | 8.702 | 31.278 | 82.631 | 122.833 | 155.723 | 155.550 | 153.188 | 149.889 | 148.727
> sum | 0.128 | 5.329 | 24.528 | 46.977 | 58.523 | 97.080 | 120.254 | 139.406 | 173.005 | 187.943
> multitask | 0.017 | 2.574 | 15.124 | 27.309 | 54.959 | 94.314 | 146.044 | 199.593 | 270.039 | 347.843
>
> ---
>
> **Median MSE for positional transformer:**
> | OOD Scaling factor |1|2|3|4|5|6|7|8|9|10|
> | :-------- | -------: |  -------: |  -------: |  -------: |  -------: |  -------: |  -------: |  -------: |  -------: |  -------: |
> min | 0.012 | 0.090 | 0.207 | 0.383 | 0.580 | 0.913 | 1.376 | 1.986 | 2.733 | 3.121
> sum | 0.095 | 0.730 | 1.532 | 2.018 | 3.397 | 4.932 | 6.407 | 7.273 | 9.425 | 11.779
> multitask | 0.014 | 0.141 | 0.466 | 1.033 | 1.640 | 2.311 | 3.045 | 4.668 | 6.075 | 8.940
>
> ---
>
> **Median MAPE for standard transformer:**
> | OOD Scaling factor |1|2|3|4|5|6|7|8|9|10|
> | :-------- | -------: |  -------: |  -------: |  -------: |  -------: |  -------: |  -------: |  -------: |  -------: |  -------: |
> min | 4.68% | 50.86% | 63.99% | 74.71% | 83.90% | 83.00% | 79.46% | 75.62% | 66.50% | 59.20%
> sum | 2.28% | 6.04% | 9.60% | 12.39% | 13.03% | 13.05% | 12.30% | 12.42% | 11.62% | 11.37%
> multitask | 4.74% | 20.50% | 36.92% | 36.26% | 39.56% | 46.51% | 49.53% | 53.05% | 55.67% | 57.56%
>
> ---
>
> **Median MAPE for positional transformer**
> | OOD Scaling factor |1|2|3|4|5|6|7|8|9|10|
> | :-------- | -------: |  -------: |  -------: |  -------: |  -------: |  -------: |  -------: |  -------: |  -------: |  -------: |
> min | 3.53% | 4.64% | 5.62% | 5.40% | 5.66% | 6.41% | 5.64% | 6.25% | 7.06% | 7.19%
> sum | 2.23% | 3.16% | 2.86% | 2.81% | 2.45% | 2.64% | 2.61% | 2.78% | 2.70% | 2.85%
> multitask | 3.26% | 4.12% | 5.56% | 6.10% | 7.60% | 8.05% | 7.58% | 10.44% | 8.31% | 8.83%
>
> We note that, even in this complex relational task with mixed-type
> inputs, the positional Transformer still significantly outperforms the standard Transformer, demonstrating the potential utility of our application for certain real-world applications.

---

> > ### Author Response · Authors · 2024-11-27
> > **Kind reminder to Reviewer DxTU**
> >
> > Would the reviewer kindly confirm whether our response has adequately addressed all questions? If there are any remaining points of confusion, we would appreciate clarification. If the reviewer is satisfied, we would be grateful if they could consider raising the score.

---

> > > ### Comment · Reviewer_DxTU · 2024-11-27
> > >
> > > I appreciate the authors' response and detailed explanation regarding my misunderstanding of the theoretical aspects of this work. However, regarding the experimental part, the embedding method for numbers differs significantly from real-world transformer implementations. In practice, numbers like '1.32' are typically tokenized as separate characters ('1', '.', '3', '2'), with each integer assigned a non-parallel vector across architectures. While I acknowledge the authors' efforts to explore alternative methods, such as using non-parallel inputs by diagonalizing each number into a one-hot vector corresponding to its position, I think these experiments do not effectively demonstrate the superiority of positional attention in these tasks. Particularly for OOD data, the standard tokenization method used in real-world applications doesn't cause severe representation shifts (In the current implementation, numbers like 1.32 and 13200 share parallel representations 1.32v and 13200v). Adding positional information through linear transformation disrupts the original data structure, making it difficult to utilize positional information effectively. Therefore, I align with other reviewers' opinions that this work requires substantial revision and is not suitable for the current conference in its present form.

---

> ### Author Response · Authors · 2024-11-28
> **Response to Official Comment by Reviewer DxTU (Part 1/3)**
>
> We appreciate the reviewer’s reply, and respect their decision to maintain their score. In any case, since the rebuttal is public, we would like to provide more evidence that the reviewer’s assumption about representation of numbers is incorrect. All results are provided in Appendix C.7, and we also summarize them in the tables below. We would also like to thank the reviewer for providing a clearer description of the experiment that they would like to observe in their second reply.
>
> We understand that the reviewer would like to see a comparison to the Transformer architecture, where a standard tokenization and digit representation is used in the Transformer architecture.  Below we perform the **exact tokenization and non-parallel digit representation which the reviewer requested.** For example: a number like “1.32” is first tokenized to (‘1’, ‘.’, ‘3’, ‘2’), and then a learnable embedding layer is used to obtain non-parallel representations of each digit of the tokenized input. Then learnable positional encodings are created through another embedding layer, which has the same dimension as the non-parallel representation of the digits, and these positional encodings are added to the non-parallel representation of the digits in a standard fashion.
>
> In the new experiment, we measure the performance of the same Transformer architecture setting as in previous experiments. Briefly, **the Transformer architecture has very poor OOD performance, even for the simplest “min” task. We took the experiment a step further, and we fine-tuned the pre-trained large version of GPT2 (774 million parameters) from Hugging Face. This experiment also failed to beat our much smaller positional Transformer.**
>
> This is to be expected. Tokenization of numbers makes the OOD task even more difficult. That’s because, in order to achieve OOD generalization, the model needs to learn to combine digits that represent a number into an appropriate representation, and then perform the reasoning and algorithmic tasks. It is almost impossible for the training procedure to converge to such a solution just by observing the training dataset, and without seeing anything about the test dataset. The numbers between the train and test datasets are largely different. Therefore, this approach naturally fails in the OOD setting. It only works in-distribution.
>
> On the contrary, representing numbers as actual numbers, and not tokens, removes the issue of having to re-combine tokens into an appropriate representation. This “task” is embedded into the simple fact of representing numbers as numbers. The learnable parallel representation of numbers is also not an issue for our architecture, since our architecture is clearly able to OOD generalize on all tasks that we experimented with. This is true even for the newer tasks, which include textual information, as we presented above and in Appendices C.6 and C.7.
>
> We hope this clarifies the concern of the reviewer that our comparison with the Transformer was not sufficient. In fact, with the proposed tokenization approach, the Transformer performed worse than what we initially reported. The worse performance of the fine-tuned GPT2 model further reinforces the argument that tokenization of numbers for algorithmic reasoning tasks is not the best approach one can implement.
>
> [...]

---

> ### Author Response · Authors · 2024-11-28
> **Response to Official Comment by Reviewer DxTU (Part 2/3)**
>
> We present here the results of the experiments described earlier.
>
> ## Results for the first setting of Appendix C.7, character-only tokenization ##
>
>
>
> **Median MSE for positional Transformer (first setting of Appendix C.7, character-only tokenization):**
>
> | OOD Scaling factor |1|2|3|4|5|6|7|8|9|10|
> | :-------- | -------: |  -------: |  -------: |  -------: |  -------: |  -------: |  -------: |  -------: |  -------: |  -------: |
> min | 0.010 | 0.265 | 0.251 | 0.440 | 0.802 | 1.298 | 1.983 | 2.970 | 4.206 | 4.817
> sum | 0.094 | 0.872 | 1.428 | 2.305 | 3.463 | 5.178 | 6.355 | 10.623 | 13.432 | 16.923
> multitask | 0.016 | 0.140 | 0.561 | 1.026 | 1.679 | 3.096 | 4.211 | 7.230 | 7.070 | 9.997
> ***
>
> **Median MSE for standard Transformer (first setting of Appendix C.7, character-only tokenization):**
>
> | OOD Scaling factor |1|2|3|4|5|6|7|8|9|10|
> | :-------- | -------: |  -------: |  -------: |  -------: |  -------: |  -------: |  -------: |  -------: |  -------: |  -------: |
> min | 0.016 | 5.125 | 27.519 | 60.279 | 96.470 | 137.361 | 159.923 | 180.919 | 209.009 | 252.358
> sum | 0.109 | 46.082 | 169.309 | 237.623 | 236.518 | 307.131 | 496.052 | 768.734 | 954.127 | 1092.671
> multitask | 0.018 | 1.094 | 5.785 | 11.105 | 12.456 | 14.069 | 20.393 | 31.993 | 48.419 | 67.631
> ***
>
> **Median MAPE for positional Transformer (first setting of Appendix C.7, character-only tokenization):**
>
> | OOD Scaling factor |1|2|3|4|5|6|7|8|9|10|
> | :-------- | -------: |  -------: |  -------: |  -------: |  -------: |  -------: |  -------: |  -------: |  -------: |  -------: |
> min | 4.06% | 9.81% | 6.95% | 6.10% | 6.30% | 6.57% | 7.76% | 7.63% | 8.04% | 7.48%
> sum | 2.24% | 3.40% | 3.10% | 2.98% | 2.89% | 3.03% | 2.95% | 3.05% | 3.17% | 3.07%
> multitask | 4.76% | 5.79% | 6.60% | 6.91% | 7.78% | 7.64% | 7.75% | 8.94% | 8.85% | 8.15%
> ***
>
> **Median MAPE for standard Transformer (first setting of Appendix C.7, character-only tokenization):**
>
>
> | OOD Scaling factor |1|2|3|4|5|6|7|8|9|10|
> | :-------- | -------: |  -------: |  -------: |  -------: |  -------: |  -------: |  -------: |  -------: |  -------: |  -------: |
> min | 4.83% | 26.30% | 44.61% | 53.68% | 62.15% | 64.70% | 64.20% | 62.54% | 61.95% | 60.23%
> sum | 2.48% | 23.80% | 30.09% | 35.61% | 32.64% | 27.62% | 24.23% | 26.94% | 27.83% | 30.56%
> multitask | 3.90% | 12.46% | 23.93% | 24.13% | 23.05% | 23.76% | 23.36% | 24.61% | 24.36% | 23.80%
> ***
>
> ## Results for the second setting of Appendix C.7, full tokenization ##
>
>
> **Median MSE for standard Transformer (second setting of Appendix C.7, full tokenization):**
>
> | OOD Scaling factor |1|2|3|4|5|6|7|8|9|10|
> | :-------- | -------: |  -------: |  -------: |  -------: |  -------: |  -------: |  -------: |  -------: |  -------: |  -------: |
> min | 0.016 | 10.933 | 41.297 | 73.948 | 130.342 | 185.914 | 256.781 | 360.054 | 454.650 | 569.807
> sum | 0.120 | 194.902 | 598.695 | 1233.994 | 2210.533 | 3177.370 | 4256.874 | 5924.806 | 7579.528 | 9857.966
> multitask | 0.018 | 14.030 | 43.701 | 83.704 | 138.093 | 195.776 | 298.741 | 375.595 | 486.065 | 629.310
> ***
>
> **Median MSE for positional Transformer (second setting of Appendix C.7, full tokenization):**
>
> | OOD Scaling factor |1|2|3|4|5|6|7|8|9|10|
> | :-------- | -------: |  -------: |  -------: |  -------: |  -------: |  -------: |  -------: |  -------: |  -------: |  -------: |
> min | 0.016 | 7.472 | 34.322 | 66.397 | 108.438 | 170.878 | 245.226 | 320.794 | 439.192 | 546.993
> sum | 0.125 | 92.457 | 512.782 | 1054.894 | 1835.501 | 2958.749 | 4212.169 | 5535.744 | 7195.396 | 8943.418
> multitask | 0.018 | 10.139 | 38.796 | 72.566 | 124.182 | 189.541 | 272.537 | 357.093 | 479.905 | 606.741
> ***
>
> **Median MAPE for standard Transformer (second setting of Appendix C.7, full tokenization):**
>
> | OOD Scaling factor |1|2|3|4|5|6|7|8|9|10|
> | :-------- | -------: |  -------: |  -------: |  -------: |  -------: |  -------: |  -------: |  -------: |  -------: |  -------: |
> min | 5.48% | 38.53% | 57.57% | 69.01% | 75.63% | 79.37% | 82.07% | 83.19% | 84.64% | 86.62%
> sum | 2.69% | 41.03% | 59.16% | 66.86% | 72.57% | 76.13% | 77.89% | 79.92% | 82.15% | 83.36%
> multitask | 4.07% | 42.84% | 60.22% | 70.40% | 73.92% | 77.68% | 80.67% | 83.00% | 83.87% | 85.60%
> ***
>
> **Median MAPE for positional Transformer (second setting of Appendix C.7, full tokenization):**
>
> | OOD Scaling factor |1|2|3|4|5|6|7|8|9|10|
> | :-------- | -------: |  -------: |  -------: |  -------: |  -------: |  -------: |  -------: |  -------: |  -------: |  -------: |
> min | 4.93% | 34.43% | 51.22% | 62.44% | 68.76% | 72.94% | 76.00% | 78.13% | 80.61% | 82.07%
> sum | 2.82% | 27.88% | 48.38% | 57.52% | 64.98% | 68.76% | 73.63% | 75.40% | 78.34% | 79.41%
> multitask | 4.47% | 34.95% | 55.74% | 64.00% | 71.15% | 74.25% | 77.82% | 79.94% | 80.78% | 82.97%
> ***
>
> [...]

---

> ### Author Response · Authors · 2024-11-28
> **Response to Official Comment by Reviewer DxTU (Part 3/3)**
>
> ## Results for the third setting of Appendix C.7, fined-tuned GPT2 ##
>
> **Median MSE for fine-tuned GPT2 (third setting of Appendix C.7, fined-tuned GPT2):**
>
> | OOD Scaling factor |1|2|3|4|5|6|7|8|9|10|
> | :-------- | -------: |  -------: |  -------: |  -------: |  -------: |  -------: |  -------: |  -------: |  -------: |  -------: |
> min | 0.078 | 0.391 | 1.285 | 2.989 | 5.539 | 8.361 | 12.811 | 13.824 | 18.318 | 18.053
> sum | 1.026 | 81.531 | 392.995 | 928.482 | 1795.709 | 2755.613 | 4056.421 | 5551.319 | 7082.871 | 9060.282
> multitask | 0.107 | 1.326 | 3.551 | 8.850 | 11.614 | 15.932 | 22.069 | 28.775 | 31.080 | 35.961
> ***
>
> **Median MAPE for fine-tuned GPT2  (third setting of Appendix C.7, fined-tuned GPT2):**
> | OOD Scaling factor |1|2|3|4|5|6|7|8|9|10|
> | :-------- | -------: |  -------: |  -------: |  -------: |  -------: |  -------: |  -------: |  -------: |  -------: |  -------: |
> min | 6.29% | 7.78% | 10.08% | 12.39% | 13.02% | 13.91% | 16.60% | 16.60% | 17.50% | 16.55%
> sum | 9.60% | 26.71% | 44.69% | 56.53% | 65.23% | 69.04% | 73.99% | 76.58% | 79.09% | 81.36%
> multitask | 9.39% | 14.01% | 13.78% | 15.05% | 14.64% | 15.21% | 16.02% | 18.31% | 15.17% | 16.37%

---

> > ### Author Response · Authors · 2024-11-29
> >
> > Would the reviewer confirm if our response addresses the tokenization issue? If satisfied, we would greatly appreciate consideration for a score increase.

---

### Meta-Review · Area_Chair_oJ8u · 2024-12-21

**Metareview:**

The paper proposes "positional attention" for improving out-of-distribution (OOD) generalization in Transformers on algorithmic reasoning tasks. The key idea is to compute attention weights using only fixed positional encodings, rather than input values.
While reviewers appreciated the motivation and experiments in this paper, they had several critical concerns that were not fully addressed. Most importantly, the core claim - that positional attention enhances OOD performance - lacks adequate precision and evidence. Many reviewers pointed out that restricting attention in this way is likely to hurt performance on many other tasks (beyond those studied in this paper) — but this important aspect was not explored adequately in the paper. The authors performed additional experiments in their rebuttal, but the experiments did not address the concerns of the reviewers.
I personally read the paper to confirm these concerns, and thus I must recommend rejection at this time.
I encourage the authors to consider the reviewer feedback in revisions.

**Additional Comments On Reviewer Discussion:**

See above.

---

### Decision · Program_Chairs · 2025-01-22

Reject